# MULTI-TASK NEURAL PROCESSES

**Donggyun Kim, Seongwoong Cho, Wonkwang Lee, Seunghoon Hong**
School of Computing, KAIST
{kdgyun425, seongwoongjo, wonkwang.lee, seunghoon.hong}@kaist.ac.kr

## ABSTRACT

Neural Processes (NPs) consider a task as a function realized from a stochastic process and flexibly adapt to unseen tasks through inference on functions. However, naive NPs can model data from only a *single* stochastic process and are designed to infer each task independently. Since many real-world data represent a set of correlated tasks from multiple sources (*e.g.,* multiple attributes and multi-sensor data), it is beneficial to infer them jointly and exploit the underlying correlation to improve the predictive performance. To this end, we propose Multi-Task Neural Processes (MTNPs), an extension of NPs designed to jointly infer tasks realized from multiple stochastic processes. We build MTNPs in a hierarchical way such that inter-task correlation is considered by conditioning all per-task latent variables on a single global latent variable. In addition, we further design our MTNPs so that they can address multi-task settings with incomplete data (*i.e.,* not all tasks share the same set of input points), which has high practical demands in various applications. Experiments demonstrate that MTNPs can successfully model multiple tasks jointly by discovering and exploiting their correlations in various real-world data such as time series of weather attributes and pixel-aligned visual modalities. We release our code at `https://github.com/GitGyun/multi_task_neural_processes`.

## 1 INTRODUCTION

Neural Processes (NPs) (Garnelo et al., 2018b) are a class of meta-learning methods that model a distribution of functions (*i.e.* a stochastic process). By considering a task as a function realized from the underlying stochastic process, they can flexibly adapt to various unseen tasks through inference on functions. The adaptation requires only one forward step of a trained neural network without any costly retraining or fine-tuning, and has linear complexity to the data size. NPs can also quantify their prediction uncertainty, which is essential in risk-sensitive applications (Gal & Ghahramani, 2016). Thanks to such appealing properties, there have been increasing attempts to improve NPs in various domains, such as image regression (Kim et al., 2019; Gordon et al., 2020), image classification (Requeima et al., 2019; Wang & Van Hoof, 2020), time series regression (Qin et al., 2019; Norcliffe et al., 2021), and spatio-temporal regression (Singh et al., 2019).

In this paper, we explore extending NPs to a multi-task setting where correlated tasks are realized simultaneously from multiple stochastic processes. Many real-world data represent multiple correlated functions, such as different attributes or modalities. For instance, medical data (Johnson et al., 2016; Harutyunyan et al., 2019) or climate data (Wang et al., 2016) contain various correlated attributes on a patient or a region that need to be inferred simultaneously. Similarly, in multi-task vision data (Lin et al., 2014; Zhou et al., 2017; Zamir et al., 2018), multiple labels of different visual modalities are associated with an image. In such scenarios, it is beneficial to exploit functional correlation by modeling the functions jointly rather than independently, in terms of performance and efficiency (Caruana, 1997). Unfortunately, naive NPs lack mechanisms to jointly handle a set of multiple functions and cannot capture their correlations either. This motivates us to extend NPs to model multiple tasks jointly by exploiting the inter-task correlation.

In addition to extending NPs to multi-task settings, we note that handling multi-task data often faces a practical challenge where observations can be incomplete (*i.e.* not all the functions share the common sample locations). For example, when we collect multi-modal signals from different sensors, the sensors may have asynchronous sampling rates, in which case we can observe signals from only an arbitrary subset of sensors at a time. To fully utilize such incomplete observations, the model should be able to associate functions observed in different inputs such that it can improve the predictive performance of all functions using their correlation. A multivariate extension of Gaussian Processes (GPs) (Álvarez et al., 2012) can handle incomplete observations to infer multiple functions jointly. However, naive GPs suffer from cubic complexity to the data size and needs approximations to reduce the complexity. Also, their behaviour depend heavily on the kernel choice (Kim et al., 2019).

To address these challenges, we introduce Multi-Task Neural Processes (MTNPs), a new family of stochastic processes that jointly models multiple tasks given possibly incomplete data. We first design a combined space of multiple functions, which allows not only joint inference on the functions but also handling incomplete data. Then we define a Latent Variable Model (LVM) of MTNP that theoretically induces a stochastic process over the combined function space. To exploit the inter-task correlation, we introduce a hierarchical LVM consists of (1) a global latent variable that captures knowledge about all tasks and (2) task-specific latent variables that additionally capture knowledge specific to each task conditioned on the global latent variable. Inducing each task conditioned on the global latent, the hierarchical LVM allows MTNP to effectively learn and exploit functional correlation in multi-task inference. MTNP also inherits advantages of NP, such as flexible adaptation, scalable inference, and uncertainty-aware prediction. Experiments in synthetic and real-world datasets show that MTNPs effectively utilize incomplete observations from multiple tasks and outperform several NP variants in terms of accuracy, uncertainty estimation, and prediction coherency.

## 2 PRELIMINARY

### 2.1 BACKGROUND: NEURAL PROCESSES

We consider a task $f^t : \mathcal{X} \to \mathcal{Y}^t$ as a realization of a stochastic process over a function space $(\mathcal{Y}^t)^{\mathcal{X}}$ that generates a data $D^t = (X_D, Y_D^t) = \{(x_i, y_i^t)\}_{i \in \mathcal{I}(D^t)}$, where $\mathcal{I}(D^t)$ denotes a set of data index. Neural Processes (NPs) use a conditional latent variable model to learn the stochastic process. Given a set of observations $C^t = (X_C, Y_C^t) = \{(x_i, y_i^t)\}_{i \in \mathcal{I}(C^t)}$, NP infers the target task $f^t$ through a latent variable $z$ and models the data $D^t$ by a factorized conditional distribution $p(Y_D^t|X_D, z)$:

$$p(Y_D^t|X_D, C^t) = \int p(Y_D^t|X_D, z)p(z|C^t)dz = \int \prod_{i \in \mathcal{I}(D^t)} p(y_i^t|x_i, z)p(z|C^t)dz. \qquad (1)$$

We refer to the set of observations $C^t$ as a *context* data and the modeling data $D^t$ as a *target* data.

NP models the generative model $p(Y_D^t|X_D, z)$ and the conditional prior $p(z|C^t)$ by two neural networks, a decoder $p_\theta$ and an encoder $q_\phi$, respectively. Since the direct optimization of Eq.1 is intractable, the networks are trained by maximizing the following variational lower-bound.

$$\log p_\theta(Y_D^t|X_D, C^t) \geq \mathbb{E}_{q_\phi(z|D^t)}[\log p_\theta(Y_D^t|X_D, z)] - D_{KL}(q_\phi(z|D^t)||q_\phi(z|C^t)). \qquad (2)$$

Note that the decoder network $q_\phi$ is also used as a variational posterior $q_\phi(z|D)$. The parameter sharing between model prior and variational posterior gives us an intuitive interpretation of the loss function: the KL term acts as a regularizer for the encoder $q_\phi$ such that the summary of the context is close to the summary of the target. This reflects the assumption that the context and target are generated by the same underlying data-generating process and aids effective test-time adaptation. After training, NP infers the target function according to the latent variable model (Eq.1).

### 2.2 EXTENDING TO MULTIPLE TARGET FUNCTIONS

Now we extend the setting to multi-task learning problems where multiple tasks $f^1, \cdots, f^T$ are realized from $T$ stochastic processes simultaneously, each of which has its own function space $(\mathcal{Y}^t)^{\mathcal{X}}, \forall t \in \mathcal{T} = \{1, 2, \cdots, T\}$. Let $D = (X_D, Y_D^{1:T}) = \bigcup_{t \in \mathcal{T}} D^t$ be a multi-task target data, where each $D^t$ corresponds to the data of task $f^t$. Then the learning objective for the set of $T$ realized tasks is to model the conditional probability $p(Y_D^{1:T}|X_D, C)$ given the multi-task context $C = (X_C, Y_C^{1:T}) = \bigcup_{t \in \mathcal{T}} C^t$, where each $C^t$ is a set of observations of task $f^t$. The sets $C$ and $D$ can be arbitrarily chosen, but we assume $C \subset D$ for simplicity.

However, assuming the complete context $C$ for all tasks is often challenged by many practical issues, such as asynchronous sampling across multiple sensors or missing labels in multi-attribute data. To address such challenges, we relax the assumptions on context $C$ and let $\mathcal{I}(C^t)$ be different across $t \in \mathcal{T}$. In this case, an input point $x_i$ can be associated with a partial set of output values $\{y_i^t\}_{t \in \mathcal{T}_i}, \mathcal{T}_i \subsetneq \mathcal{T}$, which is referred *incomplete* observation. Next, we present two ways to use NPs to model the multi-task data and discuss their limitations.

**Single-Task Neural Processes (STNPs)** A straightforward application of NPs to the multi-task setting is assuming independence across tasks and define independent NPs over the function spaces $(\mathcal{Y}^1)^{\mathcal{X}}, \cdots, (\mathcal{Y}^T)^{\mathcal{X}}$. We refer to this approach as Single-Task Neural Processes (STNPs). Specifically, a STNP has $T$ independent latent variables $v^1, \cdots, v^T$, where each $v^t$ implicitly represents a task $f^t$.

$$p(Y_D^{1:T}|X_D, C) = \prod_{t=1}^{T} \int p(Y_D^t|X_D, v^t)p(v^t|C^t)dv^t. \qquad (3)$$

Figure 1: Graphical models of three different stochastic processes for multiple functions. Gray and white circles represent observable and latent variables, respectively.

Thanks to the independence assumption, STNPs can handle incomplete context by conditioning on each task-specific data $C^t$ independently. However, this approach can only model the marginal distributions for each task, ignoring complex inter-task correlation within the joint distribution of the tasks. Note that this is especially impractical for multi-task settings under the incomplete data since each task $f^t$ can be learned only from $C^t$, ignoring rich contexts available in other data $C^{t'}$, $\forall t' \neq t$.

**Joint-Task Neural Process (JTNP)**   An alternative approach is to combine output spaces to a product space $\mathcal{Y}^{1:T} = \prod_{t \in \mathcal{T}} \mathcal{Y}^t$ and define a single NP over the function space $(\mathcal{Y}^{1:T})^{\mathcal{X}}$. We refer to this approach as Joint-Task Neural Processes (JTNPs). In this case, a single latent variable $z$ governs all $T$ tasks jointly.

$$p(Y_D^{1:T}|X_D, C) = \int p(Y_D^{1:T}|X_D, z)p(z|C)dz. \tag{4}$$

JTNPs are amenable to incorporate correlation across tasks through the shared variable $z$. However, by definition, they require complete context and target for both training and inference. This is because any incomplete set of output values $\{y_i^t\}_{t \in \mathcal{T}_i}$ for an input point $x_i$ such that $\mathcal{T}_i \neq \mathcal{T}$ is not a valid element of the product space $\mathcal{Y}^{1:T}$. In addition, it relies solely on a single latent variable to explain all tasks, ignoring per-task stochastic factors in each function $f^t$.

In what follows, we propose an alternative formulation for jointly handling multiple tasks on incomplete data, which (1) enables a probabilistic inference on the incomplete data and (2) is more amenable for learning both task-specific and task-agnostic functional representations.

## 3   MULTI-TASK NEURAL PROCESSES

In this section, we describe *Multi-Task Neural Processes* (MTNPs), a family of stochastic processes to model multiple functions jointly and handle incomplete data. We first formulate MTNPs using a hierarchical LVM. Then we propose the training objective and a neural network model.

### 3.1   FORMULATION

Our objective is to extend NPs to jointly infer multiple tasks from incomplete context. Discussions in Section 2.2 suggest that direct modeling of a distribution over functions of form $f : \mathcal{X} \rightarrow \prod_{t \in \mathcal{T}} \mathcal{Y}^t$ is achievable via JTNP (Eq. 4), yet it requires complete data in both training and inference. To circumvent this problem, we reformulate the functional form by $h : \mathcal{X} \times \mathcal{T} \rightarrow \bigcup_{t \in \mathcal{T}} \mathcal{Y}^t$. Note that this functional form allows us to model the same set of functions as JTNP by $f(x_i) = (h(x_i, 1), \cdots, h(x_i, T))$. However, by using the union form we can exploit incomplete data since any partial set of output values $\{y_i^t\}_{t \in \mathcal{T}_i}$ now becomes a set of valid output values at different input points $(x_i, t), t \in \mathcal{T}_i$. For notational convenience, we denote $x_i^t = (x_i, t)$ and assume input points in the context $C$ and the target $D$ are embedded by the task indices, *i.e.*, $C = (X_C^{1:T}, Y_C^{1:T}) = \bigcup_{t \in \mathcal{T}} C^t$ where $C^t = (X_C^t, Y_C^t) = \{(x_i^t, y_i^t)\}_{i \in \mathcal{I}(C^t)}$ and the same for $D$.

Next, we present a latent variable model that induces a stochastic process over functions of form $h$. To make use of both task-agnostic and task-specific knowledge, we define a hierarchical latent variable model (Figure 1(c)). In this model, the global latent variable $z$ captures shared stochastic factors across tasks using the whole context $C$, while per-task stochastic factors are captured by the task-specific latent variable $v^t$ using $C^t$ and $z$. It induces the predictive distribution on the target by:

$$p(Y_D^{1:T}|X_D^{1:T}, C) = \int \int \left[ \prod_{t=1}^{T} p(Y_D^t|X_D^t, v^t)p(v^t|z, C^t) \right] p(z|C)dv^{1:T}dz, \tag{5}$$

where $v^{1:T} := (v^1, \cdots, v^T)$. Similar to Eq. 1, we assume the conditional independence on $p(Y_D^t|X_D^t, v^t)$. Note that this hierarchical model can capture and leverage the inter-task correlation by sharing the same $z$ across $v^{1:T}$. Also, it is amenable to fully utilize the incomplete data:

since the global variable $z$ is inferred from the entire context data $C = \bigcup_{t \in \mathcal{T}} C^t$ and is conditioned to infer task-specific latent variable $v^t$, each function $f^t$ induced by $v^t$ exploits the observations available for not only itself $C^t$, but also for other tasks $C^{t'}, \forall t' \neq t$. Next, we show that Eq. 5 induces a stochastic process over the functions of form $h : \mathcal{X} \times \mathcal{T} \rightarrow \bigcup_{t \in \mathcal{T}} \mathcal{Y}^t$.

**Proposition 1.** *Consider the following generative process on data $D$ and context $C$, which is a generalized form of Eq. 5.*

$$z \sim p(z|C), \ v^t \sim p(v^t|z, t, C), \ y_i^t \sim p(y_i^t|x_i^t, v^t), \ \forall t \in \mathcal{T}, \ \forall i \in \mathcal{I}(D). \tag{6}$$

*Then under some mild assumptions, there exists a stochastic process over functions of form $h : \mathcal{X} \times \mathcal{T} \rightarrow \bigcup_{t \in \mathcal{T}} \mathcal{Y}^t$, where the data $D$ is generated.*

*Proof.* We leave the proof in Appendix A.2. $\qquad\square$

We refer to the resulting stochastic processes as Multi-Task Neural Processes (MTNPs). In the perspective of stochastic process, Eq. 5 allows us to learn functional posterior not only on each task via $v^t$, but also across the tasks via $z$. Then optimizing Eq. 5 can be interpreted as learning to learn each task captured by $v^t$ together with the functional correlation captured by $z$.

### 3.2 LEARNING AND INFERENCE

We use an encoder network $q_\phi$ and a decoder network $p_\theta$ to approximate the conditional prior and generative model in Eq. 5, respectively. Since the direct optimization of Eq. 5 is intracable, we train the networks via the following variational lower bound, where we use the same network $q_\phi$ for both conditional prior and variational posterior as in NP:

$$\log p_\theta(Y_D^{1:T}|X_D^{1:T}, C)$$

$$\geq \mathbb{E}_{q_\phi(z|D)} \Big[ \sum_{t=1}^{T} \mathbb{E}_{q_\phi(v^t|z, D^t)}[\log p_\theta(Y_D^t|X_D^t, v^t)]] - D_{\mathrm{KL}}\Big(q_\phi(v^t|z, D^t) \,||\, q_\phi(v^t|z, C^t)\Big) \Big]$$

$$- D_{KL}\Big(q_\phi(z|D) \,||\, q_\phi(z|C)\Big), \tag{7}$$

We leave the derivation in Appendix A.3. The above objective reflects several desirable behaviors for our model. Similar to NP, the KL divergences encourage that both latent variables $z$ and $v^t$ inferred from the context data are consistent with those inferred from the entire target data. On the other hand, we observe that minimizing the KL divergence on task-specific variables forces the global latent $z$ to be informative across all tasks, such that it can induce the task-specific factors $v^t$ from the limited context $C^t$. This makes the model encode *correlated* information across tasks in $z$ and use it for inferring each task with $v^t$, which is critically important for joint inference with incomplete context data. After training, MTNP infers the target functions according to the latent variable model (Eq. 5).

### 3.3 NEURAL NETWORK MODEL FOR MTNP

This section presents an implementation of MTNPs composed of an encoder $q_\phi$ and a decoder $p_\theta$ (Eq. 7). While our MTNP formulation is not restricted to a specific architecture, we adopt ANP (Kim et al., 2019) as our backbone, which implements the encoder by attention layers (Vaswani et al., 2017) and the decoder by a MLP. Figure 2 illustrates the overall architecture.

In the following, we denote a stacked multi-head attention block (Parmar et al., 2018) by $Attn(Q, K, V)$ and a MLP by $\psi(x)$. Also, we denote $e^t$ by a learnable task embedding for $t \in \mathcal{T}$ which is used to condition on the task index $t$.

**Latent Encoder** The latent encoder samples global and per-task latent variables by aggregating the context $C$. For each context example $(x_i^t, y_i^t) \in C^t$, we first project it to a hidden representation $s_i^t = \psi_s(x_i, y_i) + e^t$. Then we aggregate them to a task-specific representation $s^t$ via self-attention followed by a pooling operation, which is further aggregated to a global representation $s$.

$$s^t = pool(Attn(\{s_i^t\}_{i \in \mathcal{I}(C^t)}, \{s_i^t\}_{i \in \mathcal{I}(C^t)}, \{s_i^t\}_{i \in \mathcal{I}(C^t)})), \quad \forall t \in \mathcal{T}, \tag{8}$$

$$s = pool(Attn(\{s^t\}_{t \in \mathcal{T}}, \{s^t\}_{t \in \mathcal{T}}, \{s^t\}_{t \in \mathcal{T}})). \tag{9}$$

Note that the first attention is applied along the example axis (per-task) to encode information of each task, while the second one is applied along the task axis (across-task) to aggregate the information across the tasks. Then, we get the global and task-specific latent variables via ancestral sampling.

$$z \sim q_\phi(z|C) = \mathcal{N}(\psi_{(z,1)}(s), \psi_{(z,2)}(s)), \tag{10}$$

$$v^t \sim q_\phi(v^t|z, C^t) = \mathcal{N}(\psi_{(v^t,1)}(s^t, z), \psi_{(v^t,2)}(s^t, z)), \quad \forall t \in \mathcal{T}. \tag{11}$$

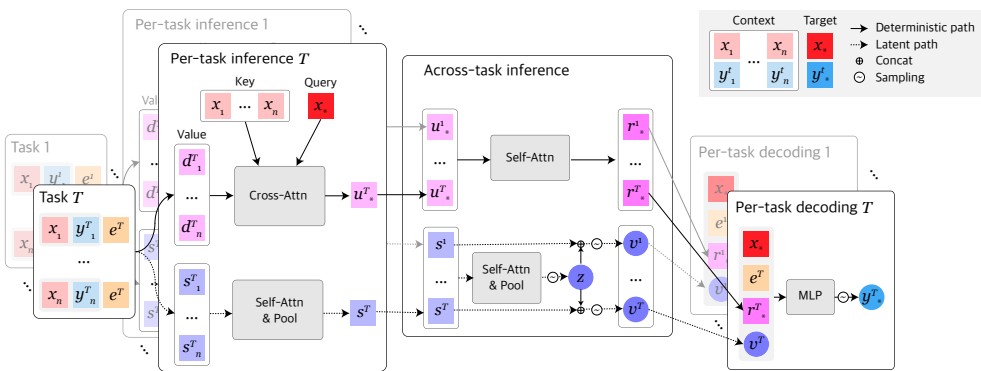

Figure 2: Architecture of the neural network model for MTNP.

**Deterministic Encoder**   To further improve the expressiveness of model, we extend the deterministic encoder of Kim et al. (2019) that produces local representation specific to both target example and task via attention mechanism. As in the latent encoder, we first project each context example $(x_i^t, y_i^t) \in C^t$ to a hidden representation $d_i^t = \psi_d(x_i, y_i^t) + e^t$ that serves as *value* embedding in cross-attention. Also, we use context and target input $x_i^t$ as *key* and *query* embeddings for the cross-attention, respectively. Then we apply cross-attention along the example axis (per-task) followed by self-attention along the task axis (across-task).

$$\{u_i^t\}_{i \in \mathcal{I}(D^t)} = Attn(\{x_i^t\}_{i \in \mathcal{I}(C^t)}, \{x_i^t\}_{i \in \mathcal{I}(C^t)}, \{d_i^t\}_{i \in \mathcal{I}(D^t)}), \quad \forall t \in \mathcal{T}, \tag{12}$$

$$\{r_i^t\}_{t \in \mathcal{T}} = Attn(\{u_i^t\}_{t \in \mathcal{T}}, \{u_i^t\}_{t \in \mathcal{T}}, \{u_i^t\}_{t \in \mathcal{T}}), \quad \forall i \in \mathcal{I}(D). \tag{13}$$

**Decoder**   Finally, the decoder produces predictive distributions for the target output $y_i^t \in Y_D^t$ for each target input $x_i^t$. We first project the input to $w_i^t = \psi_w(x_i) + e^t$, then concatenate it with the corresponding latent variable $v^t$ and deterministic representation $r_i^t$. The output distribution is computed by MLPs, whose output depends on the type of the task.

$$y_i^t \sim \begin{cases} \mathcal{N}(\psi_{(y,1)}(w_i^t, v^t, r_i^t), \psi_{(y,2)}(w_i^t, v^t, r_i^t)), & \text{if } y_i^t \text{ is continuous,} \\ \text{Categorical}(\psi_{(y,1)}(w_i^t, v^t, r_i^t)), & \text{if } y_i^t \text{ is discrete,} \end{cases} \quad \forall i \in \mathcal{I}(D^t), \forall t \in \mathcal{T}. \tag{14}$$

## 4   RELATED WORK

**Stochastic Processes for Multi-Task Learning**   There exist several stochastic processes related to MTNPs that consider learning multiple tasks. Multi-Output Gaussian Processes (MOGPs) (Álvarez et al., 2012) extend Gaussian Processes (GPs) to infer multiple tasks together, and also handle incomplete data. However, MOGPs are usually trained on a single set of tasks, thus require a lot of observations to produce accurate predictions. Recently there have been some attempts to combine meta-learning and GPs (Fortuin et al., 2019; Titsias et al., 2020), while they do not consider multi-task settings. Sequential Neural Processes (SNPs) (Singh et al., 2019; Yoon et al., 2020) treat a sequence of tasks using a sequence of NPs. While SNPs are designed to model temporal dynamics underlying a sequence of homogeneous tasks, MTNPs can capture arbitrary correlation across a set of heterogeneous tasks. Finally, Conditional Neural Adaptive Processes (CNAPs) (Requeima et al., 2019; Bateni et al., 2020) consider a general classification model for different sets of classes. However, like NPs, they infer each task independently and do not explicitly consider inter-task correlation during inference. Also, CNAPs are designed specifically to classification tasks, while MTNPs are generally applicable to various tasks including classification and regression.

**Hierarchical Models in Neural Process Family**   Since the pioneering work by Garnelo et al. (2018b), several variants of NP introduced the concept of hierarchical modeling. Attentive Neural Processes (ANPs) (Kim et al., 2019) incorporate attention mechanism to a deterministic variable, which is additional context information for each target example to improve the expressive power of the model and prevent the underfitting issues of vanilla NPs. Similarly, Wang & Van Hoof (2020) introduce local latent variables to incorporate example-specific stochasticity, which extends the graphical model of NPs to the hierarchical one. Our MTNP formulation also involves a hierarchical latent variable model but has a different structure orthogonal to the prior works. MTNPs use a global latent variable to jointly model multiple task functions while using per-task latent variables to capture task-specific stochasticity. Although extending the model to contain example-level local latent variables is possible, we adopt the deterministic local representation as in ANPs for simplicity.

## 5 EXPERIMENTS

We evaluate MTNP on three datasets, including both synthetic and real-world tasks. In all experiments, we construct incomplete context data by selecting a complete subset $C \subset D$ of size $m = |\mathcal{I}(C)|$ from the target, then randomly drop the output points independently according to the *missing rate* $\gamma \in [0, 1]$ ($\gamma = 0$ means complete data). We repeat the procedure with five different random seeds and report the mean values of each evaluation metric.

**Baselines** In each experiment, we compare MTNP with two NP variants, STNP and JTNP. We adopt ANP (Kim et al., 2019) as a backbone architecture for STNP and JTNP, which is a strong NP baseline. Since JTNP cannot handle incomplete data, we build a stronger baseline by the combination of STNP and JTNP (S+JTNP), where missing labels are imputed by STNP and then used to jointly infer the tasks by JTNP. In 1D regression tasks, we additionally compare two Multi-Output Gaussian Processes baselines, CSM (Ulrich et al., 2015) and MOSM (Parra & Tobar, 2017), and two meta-learning baselines, MAML (Finn et al., 2017) and Reptile (Nichol et al., 2018), where we slightly modify the meta-learning baselines to learn multiple tasks jointly from incomplete data. At training time, we set $\gamma = 0.5$ for all models but keeping $\gamma = 0$ for JTNP. At test time, we evaluate the models in various missing rates $\gamma \in \{0, 0.25, 0.5, 0.75\}$. We provide architectural and training details in Appendix B. We also provide ablation studies on architectural designs such as self-attentions and pooling, parameter sharing and task embedding, latent and deterministic encoders in Appendix H.

### 5.1 1D CURVE REGRESSION ON SYNTHETIC DATA

**Dataset and Metric** We begin with 1D synthetic regression tasks where the target functions are correlated by shared parameters (*e.g.,* scale, bias, phase) but have different shapes. Inspired by Guo et al. (2020b), we first randomly sample global parameters $a, b, c, w \in \mathbb{R}$ shared across the tasks, then generate four correlated tasks using different activation functions as follows.

$$y_i^t = a \cdot \text{act}_t(wx_i + b) + c, \quad \text{act}_t \in \{\text{Sine}, \text{Tanh}, \text{Sigmoid}, \text{Gaussian}\}, \; x_i \sim \mathcal{U}(-5, 5). \quad (15)$$

To simulate task-specific stochasticity, we perturb the parameters $(a, b, c, w)$ with small *i.i.d.* Gaussian noises per task. In this setting, the model has to learn per-task functional characteristics imposed by different activation functions and per-task noises, as well as how to share the underlying parameters unseen during training among the tasks. For evaluation, we generate training and testing sets via non-overlapping splits of the parameters, then measure mean squared error (MSE) normalized by the scale parameter $a$ to aggregate results on functions with different scales. See Appendix C for details.

**Results** Table 1 shows the quantitative results with $\gamma = 0.5$. More comprehensive results with different missing rates and standard deviations for the metrics are provided in Appendix D. As it shows, MTNP outperforms all baselines in all tasks and context sizes. This can be attributed to the ability of MTNP to (1) exploit all available context examples to infer inter-task general knowledge (*i.e.* $a, b, c, w$) and (2) translate it back to functional representations for each task. In contrast, STNP fails to predict multiple tasks accurately due to the independent task assumption. Although JTNP is designed to discover and utilize inter-task correlations, its performances do not show dramatic improvement over STNP since its observations are largely based on noisy imputations from STNP. We also observe that GP baselines (MOSM, CSM) perform even worse than STNP when the context size is small, despite their inherent ability to joint inference on incomplete data. We conjecture that it is because GPs lack a meta-training mechanism that allows NPs (and MTNPs) to quickly learn the tasks using a few examples. Gradient-based meta-learning baselines (MAML, Reptile) are also comparable to STNP and JTNP but perform worse than MTNP. This could be due to the lack of global inference on function space, which leads them to overfit the context points. As an illustrating example, we also plot predicted distributions from the models in a highly incomplete scenario ($m = 10$ and $\gamma = 0.5$) in Figure 3 (a). We observe that STNP generally suffers from inaccurate predictions due to limited context, while MTNP successfully exploits incomplete observations from different tasks to improve the predictive performance. The qualitative results for all baselines are provided in Appendix D.

We also perform an ablation study on the latent variable model to justify the effectiveness of our hierarchical formulation. We consider two variants of MTNP that consist of the global latent variable only (MTNP-G) and the task-specific latent variables only (MTNP-T). Then we evaluate the models in three synthetic datasets generated with different levels of inter-task correlation. Specifically, we construct partially correlated tasks as described before, totally correlated tasks by removing the task-specific noises, and independent tasks by sampling the parameters $a, b, c, w$ independently for

Table 1: Average normalized MSE on synthetic tasks, with varying context size ($m$) and $\gamma = 0.5$.

| task | Sine | | | Tanh | | | Sigmoid | | | Gaussian | | |
|---|---|---|---|---|---|---|---|---|---|---|---|---|
| $m$ | 5 | 10 | 20 | 5 | 10 | 20 | 5 | 10 | 20 | 5 | 10 | 20 |
| MAML | 0.2962 | 0.1582 | 0.0701 | 0.0991 | 0.0342 | 0.0131 | 0.0321 | 0.0119 | 0.0069 | 0.0696 | 0.0353 | 0.0174 |
| Reptile | 0.5164 | 0.2886 | 0.1414 | 0.1656 | 0.0557 | 0.0291 | 0.0619 | 0.0220 | 0.0181 | 0.1371 | 0.0679 | 0.0374 |
| MOSM | 0.7852 | 0.4410 | **0.0298** | 0.4912 | 0.1444 | 0.1618 | 0.0720 | 0.0127 | 0.0013 | 0.3329 | 0.0857 | 0.0190 |
| CSM | 0.8529 | 0.3587 | 0.1537 | 0.6884 | 0.3669 | 0.0726 | 0.2437 | 0.0730 | 0.0137 | 0.1525 | 0.0961 | 0.0407 |
| STNP | 0.5212 | 0.2609 | 0.0993 | 0.1307 | 0.0468 | 0.0159 | 0.0203 | 0.0067 | 0.0025 | 0.0799 | 0.0409 | 0.0222 |
| S+JTNP | 0.3848 | 0.2340 | 0.1114 | 0.1015 | 0.0418 | 0.0168 | 0.0163 | 0.0065 | 0.0032 | 0.0613 | 0.0318 | 0.0161 |
| MTNP | **0.2636** | **0.1137** | 0.0485 | **0.0435** | **0.0115** | **0.0040** | **0.0066** | **0.0014** | **0.0006** | **0.0360** | **0.0132** | **0.0069** |

Figure 3: (a) Predictions from NP baselines and MTNP. Black line: true function. Black dots: context points. Black crosses: imputed points from STNP. Lighter colored lines: posterior predictive samples where different colors used for different tasks. Darker colored line: mean of the samples. (b) Relative performance of MTNP variants on synthetic tasks with different levels of inter-task correlation. Top: on totally correlated tasks. Middle: on partially correlated tasks. Bottom: on independent tasks.

each task. Figure 3 (b) shows the result in $m = 10$ and $\gamma = 0.5$. When the tasks are correlated (first and second rows of the figure), we can see introducing global latent improves the overall performance, which is further improved by the hierarchical formulation. When the tasks are independent (third row of the figure), sharing all knowledge through a single global latent degrades the performance (MTNP-G). On the other hand, MTNP and MTNP-T do not suffer from such a negative transfer since each of the independent tasks can be addressed by per-task latent variables separately. The overall results demonstrate that incorporating both global and task-specific information is the most effective and robust against various levels of inter-task correlation.

## 5.2 1D TIME-SERIES REGRESSION ON WEATHER DATA

**Dataset and Metric** To demonstrate our method in a practical, real-world domain, we perform an experiment on weather data. Weather attributes are physically correlated with each other, and the observations are often incomplete due to different sensor configurations or coverage per station. Also, the observed attributes are highly stochastic, making MTNP's stochastic process formulation fits it well. We use a dataset gathered by Dark Sky API [1], consisting of 12 daily weather attributes collected at 266 cities for 258 days. We choose six attributes, namely low and high temperatures (TempMin, TempMax), humidity (Humidity), precipitation probability (Precip), cloud cover (Cloud), and dew point (Dew), which forms six correlated tasks. We normalize each attribute to be standard Gaussian and the time to be in $[0, 1]$. We divide the data into 200 training, 30 valid, and 33 test sets of time series, where each set corresponds to a unique city. We evaluate the prediction performance by MSE. Since the data is noisy, we also report negative log-likelihood as a metric of uncertainty estimation.

**Results** Table 2 summarizes quantitative results. More comprehensive results with different missing rates, context sizes, and standard deviations for the metrics are provided in Appendix E. MTNP outperforms all baselines in both accuracy and uncertainty estimation, which demonstrates that it generalizes well to real-world stochastic data. More interestingly, Figure 4 illustrates how MTNP

---

[1]https://github.com/imantsm/COVID-19

Table 2: Average MSE and NLL on weather tasks, with $m = 10$ and $\gamma = 0.5$.

| task | TempMin | | TempMax | | Humidity | | Precip | | Cloud | | Dew | |
|---|---|---|---|---|---|---|---|---|---|---|---|---|
| metric | MSE | NLL | MSE | NLL | MSE | NLL | MSE | NLL | MSE | NLL | MSE | NLL |
| MAML | 0.0067 | - | 0.0094 | - | 0.0705 | - | 0.3041 | - | 0.2987 | - | 0.0106 | - |
| Reptile | 0.0060 | - | 0.0078 | - | 0.0691 | - | 0.3160 | - | 0.3047 | - | 0.0096 | - |
| MOSM | 0.0091 | -0.0194 | 0.0124 | -0.0259 | 0.0827 | 1.3831 | 0.3021 | 4.1009 | 0.3170 | 2.0663 | 0.0128 | -0.0255 |
| CSM | 0.0069 | -0.8839 | 0.0123 | -0.8522 | 0.0906 | 0.6640 | 0.2895 | 3.1897 | 0.2983 | 1.2655 | 0.0118 | -0.7243 |
| STNP | 0.0046 | -1.1514 | 0.0069 | -1.0390 | 0.0632 | 0.1273 | 0.2607 | 1.1242 | 0.2631 | 0.8563 | 0.0086 | -0.9815 |
| S+JTNP | 0.0045 | -1.1703 | 0.0068 | -1.0681 | 0.0607 | 0.0169 | 0.2348 | 0.6792 | 0.2376 | 0.6812 | 0.0084 | -0.9946 |
| MTNP | **0.0037** | **-1.1832** | **0.0054** | **-1.1049** | **0.0546** | **-0.1006** | **0.2276** | **0.6557** | **0.2215** | **0.6660** | **0.0073** | **-1.0331** |

Figure 4: Visualization of MTNP's internal knowledge transfer. By observing additional data from Cloud task (at red triangles) given upon a few context points (at blue dots), the predicted mean and variance of Precip task improve at the additionally observed region.

transfer its knowledge from one task (Cloud) to another (Precip) given the incomplete observations. When the observation is sparse (Figure 4(a)), the model produces an inaccurate prediction with high uncertainty for unobserved input domains. However, when the additional observations are available for the other attribute (Figure 4(b),(c)), MTNP successfully transfers the knowledge to improve the prediction. It shows that MTNP can effectively learn to exploit the incomplete observation by transferring knowledge across tasks.

## 5.3   2D IMAGE COMPLETION ON FACE DATA

**Dataset and Metric**   We further demonstrate our approach to more challenging 2D structured function regression tasks. Following Garnelo et al. (2018a), we interpret an RGB image as a function that maps a 2D pixel location $x_i \in [0, 1]^2$ to its RGB values $y_i \in [0, 1]^3$, and extend its concept to pixel-aligned 2D spatial data for the multi-task setting. Specifically, we consider four pixel-aligned visual modalities with a resolution of $32 \times 32$ on celebrity faces as a set of tasks, namely RGB image (RGB) (Liu et al., 2015), semantic segmentation map (Segment) (Lee et al., 2020), Sobel edge (Edge) (Kanopoulos et al., 1988), and Projected Normalized Coordinate Code (PNCC) (Zhu et al., 2016). We then construct training and testing sets with non-overlapping splits of face images. To evaluate the Segment task, we report mean Intersection-over-Union (mIoU). For the other tasks, we report MSE. We also measure prediction coherency across tasks to evaluate the task correlation captured by models. To measure the coherency between the predictions, we generate pseudo-labels by translating the RGB prediction into the other three modalities using image-to-image translation methods (Kanopoulos et al., 1988; Guo et al., 2020a; Chen et al., 2018), then measure errors (MSE or 1 - mIoU) between the pseudo-labels and predictions. Additional details are provided in Appendix F.

**Results**   Table 3 summarizes the quantitative comparison results. More comprehensive results with different missing rates are provided in Appendix G. Overall, we observe similar results with the 1D regression experiments where MTNP generates more accurate predictions over STNP and S+JTNP by effectively exploiting the incomplete data. We also observe that the MTNP produces more coherent predictions over the baselines, which shows that it indeed learns to exploit the correlation across tasks effectively. To further validate the results, we present qualitative comparison results in Figure 5. We observe that STNP and S+JTNP produce inaccurate (red boxes) or incoherent (green box) outputs when the number of contexts is extremely small. On the other hand, MTNP (1) consistently regresses coherent functions regardless of the number of observable contexts, and (2) its predictions are more accurate than the baselines given the same number of contexts (green box).

Finally, we investigate the discovery and exploitation of task correlations achieved by MTNP. We first partition tasks into *source* and *target* tasks. Then, we measure relative performance improvement on the target tasks before and after the model observes data from source tasks. We summarize the results in Table 4, where we average performance gains coming from all possible combinations of source tasks for each target task. By observing which task is the most beneficial to each of the other tasks, we observe that there are two groups of highly correlated tasks (RGB-Edge) and (Segment-PNCC). These

Table 3: Quantitative results on 2D function regression ($\gamma = 0.5$). Upper rows show prediction performance and lower rows show prediction coherency, reported by MSE and (1-mIoU) for continuous and categorical data, respectively (lower-the-better).

| Tasks | RGB | | | Edge | | | Segment | | | PNCC | | |
|---|---|---|---|---|---|---|---|---|---|---|---|---|
| $m$ | 10 | 100 | 512 | 10 | 100 | 512 | 10 | 100 | 512 | 10 | 100 | 512 |
| STNP | 0.0440 | 0.0154 | 0.0054 | 0.0359 | 0.0256 | 0.0116 | 0.6637 | 0.4669 | 0.2958 | 0.0102 | 0.0015 | 0.00061 |
| S+JTNP | 0.0421 | 0.0129 | 0.0046 | 0.0338 | 0.0190 | 0.0090 | 0.6316 | 0.4341 | 0.3171 | 0.0105 | 0.0021 | 0.00088 |
| MTNP | **0.0400** | **0.0114** | **0.0032** | **0.0323** | **0.0166** | **0.0060** | **0.6073** | **0.4013** | **0.2882** | **0.0082** | **0.0012** | **0.00058** |
| STNP | - | - | - | 0.0314 | 0.0261 | 0.0174 | 0.6632 | 0.5863 | 0.5361 | 0.0362 | 0.0267 | 0.0231 |
| S+JTNP | - | - | - | 0.0187 | 0.0124 | 0.0143 | 0.5298 | 0.5110 | 0.5140 | 0.0106 | 0.0139 | 0.0194 |
| MTNP | - | - | - | **0.0184** | **0.0089** | **0.0053** | **0.5161** | **0.4923** | **0.4963** | **0.0104** | **0.0115** | **0.0134** |

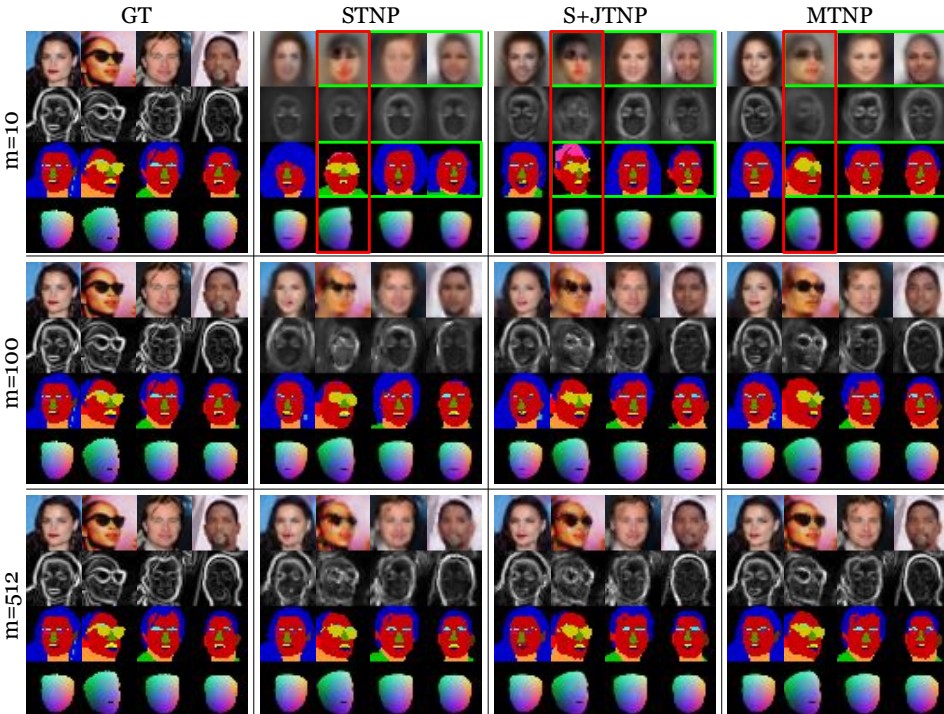

Figure 5: Qualitative results on 2D function regression. Performances of all models improve as the number of observable contexts ($m$) increases. However, under the limited number of observable contexts (*e.g.* $m = 10$), STNP and S+JTNP produce inaccurate outputs (*e.g.* mis-predicting hairs and poses as in the green box) or incoherent outputs (*e.g.* different head poses as in the red box).

results demonstrate that MTNP successfully captured dependence among tasks considering that (1) RGB and Edge are composed of two correlated low-level signals (*e.g.* color intensity and its gradients) (2) while both Segment and PNCC contain high-level semantic information on facial landmarks. Note that discovering inter-task correlations is one of the actively studied topics in the machine learning literature, where the efforts often come at the cost of extra computations and resources due to hard-coded (Zamir et al., 2018; Standley et al., 2020) or hand-crafted (Pal & Balasubramanian, 2019) algorithms.

Table 4: Relative performance gain (%).

| Source \ Target | RGB | Edge | Segment | PNCC |
|---|---|---|---|---|
| RGB | - | **53.02** | 8.73 | 18.57 |
| Edge | **6.35** | - | 8.18 | 15.70 |
| Segment | 5.13 | 33.30 | - | **29.24** |
| PNCC | 5.58 | 31.88 | **15.88** | - |

## 6 CONCLUSION

We propose Multi-Task Neural Processes (MTNPs), a new family of stochastic processes designed to infer multiple functions jointly from incomplete data, along with a hierarchical latent variable model. Through extensive experiments, we demonstrate that the proposed MTNPs can leverage incomplete data to solve multiple heterogenous tasks by learning to discover and exploit task-agnostic and task-specific knowledge. Scaling up our method to large-scale datasets will be a promising research direction. To this end, our method can be improved in several aspects by (1) generalizing to unseen task space $\mathcal{T}$ and (2) allowing empty context data for some tasks such that we can generalize MTNPs in more diverse real-world scenarios such as zero-shot inference and semi-supervised learning.

**Acknowledgements**    This work was supported by the Institute of Information & communications Technology Planning & Evaluation (IITP) (No. 2021-0-00537 and 2019-0-00075) and the National Research Foundation of Korea (NRF) (No. 2021R1C1C1012540) funded by the Korea government (MSIT).

**Ethics Statement**    Recently, detecting and removing data bias have become essential problems towards producing fair machine learning models. We believe that our work can contribute to detect unintentional data bias present in multi-attribute data. MTNP can be seen as a universal correlation learner who learns arbitrary correlation across tasks purely data-driven way. Therefore, given potentially biased multi-attribute data (e.g., multiple personal attributes), MTNP may detect any biased relationship by learning the correlation between them. For example, we may perform the task-to-task transfer analysis on a trained MTNP as discussed in Section 5.2 and Section 5.3, then see which task (or attribute) has a high correlation with another task (or attribute).

**Reproducibiltiy Statement**    In this work, we present two major theoretical results (Proposition 1 and Eq. 7), a neural network model (Section 3.3), and experiments on three datasets (Section 5). We give a complete proof of Proposition 1 in Appendix A.2 and the ELBO derivation for Eq. 7 in Appendix A.3. We provide architectural details and training hyper-parameters of the models used in the experiments in Appendix B. Finally, details on the experimental settings and datasets are provided in Appendix C and Appendix F.

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

# APPENDIX

## A    THEORETICAL JUSTIFICATIONS

In this section, we give a proof of Proposition 1 with a brief introduction of the Kolmogorov Extension Theorem (Itô et al., 1984), and derive training objectives of STNP, JTNP, and MTNP.

### A.1    STOCHASTIC PROCESSES AND THE KOLMOGOROV EXTENSION THEOREM

A stochastic process $F : \mathcal{X} \times \Omega \to \mathcal{Y}$ is a collection of random variables $\{\mathbb{Y}_x : \Omega \to \mathcal{Y}\}_{x \in \mathcal{X}}$ which is indexed by an index set $\mathcal{X}$. Also, all the random variables are defined on a single probability space $(\Omega, \mathcal{F}, \mathbb{P})$ and a value space $\mathcal{Y}$. This can be interpreted as a distribution over a function space $\mathcal{Y}^{\mathcal{X}}$, such that sampling a function $f$ corresponds to $f(\cdot) = F(\cdot, \omega)$, $\omega \in \Omega$. Another interpretation of $F$ is a random function, since $F(x, \cdot) = \mathbb{Y}_x$ is a random variable.

Suppose we have observed input and output sequences $X = (x_1, x_2, \cdots, x_n)$ and $Y = (y_{x_1}, y_{x_2}, \cdots, y_{x_n})$ of a function $f : \mathcal{X} \to \mathcal{Y}$. With a slight abuse of a notation, let $p(Y|X) = \rho_X(Y)$ be the marginal distribution of $Y$ on a product space $\prod_{i=1}^{n} \mathcal{Y}$, where each $i$-th space of the product is the output space of $\mathbb{Y}_{x_i}$, $1 \leq i \leq n$. Then by the Kolmogorov Extension Theorem, the data $(X, Y)$ induces a stochastic process $F$ such that $\exists\, \omega \in \Omega$ s.t. $y_{x_i} = F(x_i, \omega)$ for all $i = 1, 2, \cdots, n$, if the distribution $p(Y|X)$ satisfies two conditions: consistency and exchangability.

1. (Consistency) For any $m$ such that $1 \leq m < n$,

$$\int p(Y|X)dY_{m+1:n} = p(Y_{1:m}|X_{1:m}), \tag{16}$$

   where $X_{i_1:i_2} = (x_{i_1}, x_{i_1+1}, \cdots, x_{i_2})$ and $Y_{i_1:i_2} = (y_{i_1}, y_{i_1+1}, \cdots, y_{i_2})$ for all $i_1 \leq i_2$.

2. (Exchangability) For any permutation $\pi$ on $\mathcal{X}_{1:n}$ (a permutation $\pi$ on set $S$ is a bijection $\pi : S \to S$),

$$p(\pi \circ Y | \pi \circ X) = p(Y|X), \tag{17}$$

   where $\mathcal{X}_{1:n} = \{x_1, \cdots, x_n\}$, $\pi \circ X = (\pi(x_1), \pi(x_2), \cdots, \pi(x_n))$, and $\pi \circ Y = (y_{\pi(x_1)}, y_{\pi(x_2)}, \cdots, y_{\pi(x_n)})$.

### A.2    MTNP IS A STOCHASTIC PROCESS

In the case of MTNP, we observe input and output sequences $X = ((x_1, t_1), (x_2, t_2), \cdots, (x_n, t_n))$ and $Y = (y_{(x_1, t_1)}, y_{(x_2, t_2)}, \cdots, y_{(x_n, t_n)})$ of a function $h : \mathcal{X} \times \mathcal{T} \to \bigcup_{t \in \mathcal{T}} \mathcal{Y}^t$. Note that in the main text, we abbreviate $x_i^t = (x_i, t)$ and $y_i^t = y_{(x_i, t)}$ for visibility. Now we want to show the existence of a stochastic process $H : \mathcal{X} \times \mathcal{T} \times \Omega \to \bigcup_{t \in \mathcal{T}} \mathcal{Y}^t$, where the data $D = (X, Y)$ is generated. This can be done by showing the following conditions.

1. (Consistency) For any $m$ such that $1 \leq m < n$,

$$\int p(Y|X, C)dY_{m+1:n} = p(Y_{1:m}|X_{1:m}, C), \tag{18}$$

   where $X_{i_1:i_2} = ((x_{i_1}, t_{i_1}), \cdots, (x_{i_2}, t_{i_2}))$ and $Y_{i_1:i_2} = (y_{(x_{i_1}, t_{i_1})}, \cdots, y_{(x_{i_2}, t_{i_2})})$.

2. (Exchangability) For any permutation $\pi$ on $\mathcal{X}_{1:n} \times \mathcal{T}$,

$$p(\pi \circ Y | \pi \circ X, C) = p(Y|X, C), \tag{19}$$

   where $\mathcal{X}_{1:n} = \{x_1, \cdots, x_n\}$, $\pi \circ X = (\pi((x_1, t_1)), \cdots, \pi((x_n, t_n)))$, and $\pi \circ Y = (y_{\pi((x_1, t_1))}, \cdots, y_{\pi((x_n, t_n))})$.

Here $p(Y|X, C) = \rho_X(Y|C)$ is the conditional distribution of $Y$ given any context $C$. Note that $C$ is conditioned since we are modeling functional *posterior* of $h$, rather than *prior*.

Now we provide the proof of Proposition 1, which states that the following generative model defines a stochastic process.

$$z \sim p(z|C),\ v^t \sim p(v^t|z, t, C),\ y_{(x_i, t)} \sim p(y_{(x_i, t)}|x_i, t, v^t)\ \forall t \in \mathcal{T},\ \forall x_i \in \mathcal{X}_{1:n}, \tag{20}$$

To show the conditions of Kolmogorov Extension Theorem, we need two assumptions on the data generating process (Eq. 20). First, we assume the distribution defined by the data generating process is finite so that the order of integral can be swapped. Also, we assume that the conditional distribution $p(y_{(x_i,t)}|x_i, t, v^t)$ can implicitly *select* the per-task latent variable $v^t$ among $v^{1:T}$ using the given task index $t$, *i.e.,* there exists a distribution $\tilde{p}$ such that $\tilde{p}(y_{(x_i,t)}|x_i, t, v^{1:T}) = p(y_{(x_i,t)}|x_i, t, v^t)$. This means no more than that the latent variables $v^{1:T}$ are indeed *task-specific*, such that each $v^t$ corresponds to task $f^t$. Note that our neural-network model of MTNP (Figure 2) indeed satisfies the second assumption, since the decoder selects the corresponding per-task latent variable $v^t$ given the task index $t \in \mathcal{T}$.

*Proof.* We first show the consistency condition. From the data generating process (Eq. 20),

$$\int p(Y|X, C)dY_{m+1:n} \tag{21}$$

$$= \int\int\int \left(\prod_{i=1}^{n} p(y_{(x_i,t_i)}|x_i, t_i, v^{t_i})\right)\left(\prod_{t=1}^{T} p(v^t|z, t, C)\right)p(z|C)dv^{1:T}dzdY_{m+1:n} \tag{22}$$

$$= \int\int \left(\prod_{i=1}^{m} p(y_{(x_i,t_i)}|x_i, t_i, v^{t_i})\right)$$
$$\left(\int \prod_{i=m+1}^{n} p(y_{(x_i,t_i)}|x_i, t_i, v^{t_i})dY_{m+1:n}\right)\left(\prod_{t=1}^{T} p(v^t|z, t, C)\right)p(z|C)dv^{1:T}dz \tag{23}$$

$$= \int\int \left(\prod_{i=1}^{m} p(y_{(x_i,t_i)}|x_i, t_i, v^{t_i})\right)\left(\prod_{t=1}^{T} p(v^t|z, t, C)\right)p(z|C)dv^{1:T}dz \tag{24}$$

$$= p(Y_{1:m}|X_{1:m}, C). \tag{25}$$

Next, we show the exchangability condition. Let $\pi_1, \pi_2$ be the values of first and second coordinate of $\pi$, such that $\pi((x_i, t_i)) = (\pi_1((x_i, t_i)), \pi_2((x_i, t_i)))$. Then

$$p(\pi \circ Y|\pi \circ X, C) \tag{26}$$

$$= \int\int \left(\prod_{i=1}^{n} \tilde{p}(y_{\pi((x_i,t_i))}|\pi((x_i, t_i)), v^{1:T})\right)\left(\prod_{t=1}^{T} p(v^t|z, t, C)\right)p(z|C)dv^{1:T}dz \tag{27}$$

$$= \int\int \left(\prod_{i=1}^{n} p(y_{\pi((x_i,t_i))}|\pi((x_i, t_i)), v^{\pi_2((x_i,t_i))})\right)\left(\prod_{t=1}^{T} p(v^t|z, t, C)\right)p(z|C)dv^{1:T}dz \tag{28}$$

$$= \int\int \left(\prod_{i=1}^{n} p(y_{(x_i,t_i)}|x_i, t_i, v^{t_i})\right)\left(\prod_{t=1}^{T} p(v^t|z, t, C)\right)p(z|C)dv^{1:T}dz \tag{29}$$

$$= p(Y|X, C). \tag{30}$$

Here we used the assumption about $p(y_{(x_i,t)}|x_i, t, v^t)$ such that $\tilde{p}(y_{\pi((x_i,t_i))}|\pi((x_i, t_i)), v^{1:T}) = p(y_{\pi((x_i,t_i))}|\pi((x_i, t_i)), v^{\pi_2((x_i,t_i))})$. Since $1 \leq m < n$ and $\pi$ are arbitrarily chosen, the data generating process (Eq. 20) satisfies the conditions of the Kolmogorov Extension Theorem. Thus there exists a stochastic process $H : \mathcal{X} \times \mathcal{T} \times \Omega \rightarrow \bigcup_{t \in \mathcal{T}} \mathcal{Y}^t$, whose realizations are functions of the form $h : \mathcal{X} \times \mathcal{T} \rightarrow \bigcup_{t \in \mathcal{T}} \mathcal{Y}^t$. $\square$

Note that the latent variable model of MTNP (Eq. 5) is a special case of the data generating process (Eq. 20), where $p(v^t|z, t, C) = p(v^t|z, C^t)$. Thus MTNP is a stochastic process over the functions of form $h : \mathcal{X} \times \mathcal{T} \rightarrow \bigcup_{t \in \mathcal{T}} \mathcal{Y}^t$.

### A.3 ELBO DERIVATION FOR MTNP

We derive the evidence lower bound (ELBO) for $\log p_\theta(Y_D^{1:T}|X_D^{1:T}, C)$. Recall that $C = (X_C^{1:T}, Y_C^{1:T}) = \bigcup_{t \in \mathcal{T}} C^t$ where $C^t = (X_C^t, Y_C^t) = \{(x_i^t, y_i^t)\}_{i \in \mathcal{I}(C^t)}$ and $D = (X_D^{1:T}, Y_D^{1:T}) = \bigcup_{t \in \mathcal{T}} D^t$ where $D^t = (X_D^t, Y_D^t) = \{(x_i^t, y_i^t)\}_{i \in \mathcal{I}(D^t)}$. For simplicity, we assume $C^t \subset D^t$ for all $t$ so that $C \subset D$ as well. Also, to avoid confusion, for this derivation we denote the conditional prior networks as $p_\theta(z|C^t)$ and $p_\theta(v^t|z, C^t)$ and then replace them with $q_\phi(z|C^t)$ and $q_\phi(v^t|z, C^t)$ respectively, when we introduce parameter-sharing between prior and variational posterior networks.

First, the conditional log-likelihood has a lower bound

$$\log p_\theta(Y_D^{1:T}|X_D^{1:T}, C) \tag{31}$$

$$= \mathbb{E}_{q_\phi(z|D)}\left[\log \frac{p_\theta(Y_D^{1:T}, z|X_D^{1:T}, C)}{p_\theta(z|X_D^{1:T}, Y_D^{1:T}, C)}\right] \tag{32}$$

$$= \mathbb{E}_{q_\phi(z|D)}\left[\log \frac{p_\theta(Y_D^{1:T}|X_D^{1:T}, C, z)p_\theta(z|X_D^{1:T}, C)}{p_\theta(z|D)}\right] \tag{33}$$

$$= \mathbb{E}_{q_\phi(z|D)}\left[\log \frac{p_\theta(Y_D^{1:T}|X_D^{1:T}, C, z)p_\theta(z|C)}{p_\theta(z|D)}\right] \tag{34}$$

$$= \mathbb{E}_{q_\phi(z|D)}\left[\log p_\theta(Y_D^{1:T}|X_D^{1:T}, C, z)\right] + D_{\text{KL}}\Big(q_\phi(z|D) \,||\, p_\theta(z|D)\Big)$$
$$- D_{\text{KL}}\Big(q_\phi(z|D) \,||\, p_\theta(z|C)\Big) \tag{35}$$

$$\geq \mathbb{E}_{q_\phi(z|D)}\left[\log p_\theta(Y_D^{1:T}|X_D^{1:T}, C, z)\right] - D_{\text{KL}}\Big(q_\phi(z|D) \,||\, p_\theta(z|C)\Big), \tag{36}$$

where $p_\theta(Y_D^{1:T}|X_D^{1:T}, C, z)$ in Eq. 36 can be further expanded by

$$\log p_\theta(Y_D^{1:T}|X_D^{1:T}, C, z) \tag{37}$$

$$= \mathbb{E}_{\prod_{t=1}^T q_\phi(v^t|z, D^t)}\left[\log \frac{p_\theta(Y_D^{1:T}, v^{1:T}|X_D^{1:T}, C, z)}{p_\theta(v^{1:T}|X_D^{1:T}, Y_D^{1:T}, C, z)}\right] \tag{38}$$

$$= \mathbb{E}_{\prod_{t=1}^T q_\phi(v^t|z, D^t)}\left[\log \frac{p_\theta(Y_D^{1:T}|X_D^{1:T}, C, z, v^{1:T})p_\theta(v^{1:T}|X_D^{1:T}, C, z)}{p_\theta(v^{1:T}|z, D)}\right] \tag{39}$$

$$= \mathbb{E}_{\prod_{t=1}^T q_\phi(v^t|z, D^t)}\left[\sum_{t=1}^T \log \frac{p_\theta(Y_D^t|X_D^t, v^t)p_\theta(v^t|z, C^t)}{p_\theta(v^t|z, D^t)}\right] \tag{40}$$

$$= \sum_{t=1}^T \mathbb{E}_{q_\phi}\left[\log \frac{p_\theta(Y_D^t|X_D^t, v^t)p_\theta(v^t|z, C^t)}{p_\theta(v^t|z, D^t)}\right] \tag{41}$$

$$= \sum_{t=1}^T \mathbb{E}_{q_\phi}\left[\log p_\theta(Y_D^t|X_D^t, v^t)\right] + D_{\text{KL}}\Big(q_\phi(v^t|z, D^t) \,||\, p_\theta(v^t|z, D^t)\Big)$$
$$- D_{\text{KL}}\Big(q_\phi(v^t|z, D^t) \,||\, p_\theta(v^t|z, C^t)\Big) \tag{42}$$

$$\geq \sum_{t=1}^T \mathbb{E}_{q_\phi}\left[\log p_\theta(Y_D^t|X_D^t, v^t)\right] - D_{\text{KL}}\Big(q_\phi(v^t|z, D^t) \,||\, p_\theta(v^t|z, C^t)\Big). \tag{43}$$

On Eq. 34 and Eq. 40, we use the conditional independence relation follows from the latent variable model (Eq. 5) By combining Eq. 36 and Eq. 43, and also by sharing the parameters of conditional priors $p_\theta(z|C)$ and $p_\theta(v^t|z, C^t)$ with variational posteriors $q_\phi(z|C)$ $q_\phi(v^t|z, C^t)$, we get

the following lower bound.

$$\log p_\theta(Y_D^{1:T}|X_D^{1:T}, C) \tag{44}$$

$$\geq \mathbb{E}_{q_\phi}\left[\sum_{t=1}^{T}\mathbb{E}_{q_\phi}\left[\log p_\theta(Y_D^t|X_D^t, v^t)\right] - D_{\text{KL}}\left(q_\phi(v^t|z, D^t) \,||\, p_\theta(v^t|z, C^t)\right)\right]$$

$$- D_{\text{KL}}\left(q_\phi(z|D) \,||\, p_\theta(z|C)\right) \tag{45}$$

$$= \mathbb{E}_{q_\phi}\left[\sum_{t=1}^{T}\mathbb{E}_{q_\phi}\left[\log p_\theta(Y_D^t|X_D^t, v^t)\right] - D_{\text{KL}}\left(q_\phi(v^t|z, D^t) \,||\, q_\phi(v^t|z, C^t)\right)\right]$$

$$- D_{\text{KL}}\left(q_\phi(z|D) \,||\, q_\phi(z|C)\right). \tag{46}$$

## A.4 ELBO FOR STNP AND JTNP

STNP is no more than a collection of independent Neural Processes (NPs), where each NP corresponds to each task. Using $T$ encoders and decoders $\{(p_{\theta_t}, q_{\phi_t})\}_{t=1}^T$, the objective for STNP can be derived by summing up the NP objectives (Eq. 2). We omit the ELBO derivation for NP. Note that the parameter sharing is used for the conditional prior network $p_{\theta_t}(v^t|C^t)$ and the variational posterior network $q_{\phi_t}(v^t|D^t)$.

$$\log p_\theta(Y_D^{1:T}|X_D, C) \tag{47}$$

$$\geq \sum_{t=1}^T \mathbb{E}_{q_{\phi_t}(v^t|D^t)}\Big[\log p_{\theta_t}(Y_D^t|X_D, v^t)\Big] - D_{\text{KL}}\Big(q_{\phi_t}(v^t|D^t) \,||\, p_{\theta_t}(v^t|C^t)\Big) \tag{48}$$

$$= \sum_{t=1}^T \mathbb{E}_{q_{\phi_t}(v^t|D^t)}\Big[\log p_{\theta_t}(Y_D^t|X_D, v^t)\Big] - D_{\text{KL}}\Big(q_{\phi_t}(v^t|D^t) \,||\, q_{\phi_t}(v^t|C^t)\Big). \tag{49}$$

On the other hand, JTNP is a single NP that models all tasks jointly, by concatenating the output variables into a single vector. Using an encoder $q_\phi$ and a decoder $p_\theta$, the objective for JTNP is the same as the NP objective. Again, the encoder $q_\phi$ serves as both conditional prior and variational posterior.

$$\log p_\theta(Y_D^{1:T}|X_D, C) \tag{50}$$

$$\geq \mathbb{E}_{q_\phi(z|D)}\Big[\log p_\theta(Y_D^{1:T}|X_D, z)\Big] - D_{\text{KL}}\Big(q_\phi(z|D) \,||\, p_\theta(z|C)\Big) \tag{51}$$

$$= \mathbb{E}_{q_\phi(z|D)}\Big[\log p_\theta(Y_D^{1:T}|X_D, z)\Big] - D_{\text{KL}}\Big(q_\phi(z|D) \,||\, q_\phi(z|C)\Big). \tag{52}$$

Note that STNP and JTNP model functions with input space $\mathcal{X}$, so there is no superscript $t$ in the input variables.

# B ARCHITECTURAL AND TRAINING DETAILS

In this section, we provide architectural and training details about models used in the experiments (Section 5).

## B.1 ATTENTIVE NEURAL PROCESS

As a strong NP baseline, we adopt Attentive Neural Processes (ANPs) (Kim et al., 2019) architecture for STNP and JTNP. The encoder of ANP consists of a latent path and a deterministic path, each computes a latent variable $z$ and a deterministic representation $r_i$ specific to each target example $x_i$. Then the decoder produces a distribution for the target output $y_i$, which is assumed to be Normal.

The general architecture of ANP follows Kim et al. (2019), while two major modifications have made as follows. First, we replace the average pooling operation in stochastic path by a Pooling by Multihead Attention (PMA) layer which is introduced in Lee et al. (2019). Next, since we have a categorical task (Segment) in 2D experiment, we modify the decoder for each Segment task to output logits for the Categorical distribution it models.

## B.2 ANP MODEL FOR JTNP

This section presents a detailed description of the JTNP architecture used in the experiments. The JTNP consists of a single ANP, which consists of a latent encoder, a deterministic encoder, and a decoder. Then JTNP produces target output distribution by conditioning on the context set $C = \{(x_i, y_i^{1:T})\}_{i \in \mathcal{I}(C)}$ where $y_i^{1:T} = (y_i^1, \cdots, y_i^T)$.

**Latent Encoder** The latent encoder samples a global latent $z$. For each context example $(x_i, y_i^{1:T}) \in C$, we first project it to a hidden representation $s_i^{1:T} = \psi_s(x_i, y_i^{1:T})$ using a single MLP $\psi_s$. Then we aggregate them to a global representation $s$ via self-attention followed by a pooling operation.

$$s = pool(Attn(\{s_i^{1:T}\}_{i \in \mathcal{I}(C)}, \{s_i^{1:T}\}_{i \in \mathcal{I}(C)}, \{s_i^{1:T}\}_{i \in \mathcal{I}(C)}). \tag{53}$$

Then the global latent is sampled via two MLPs.

$$z \sim q_\phi(z|C) = \mathcal{N}(\psi_{(z,1)}(s), \psi_{(z,2)}(s)). \tag{54}$$

**Deterministic Encoder** The deterministic encoder produces local representation $r_i$ for each $i \in \mathcal{D}$. We first project each context example $(x_i, y_i^{1:T}) \in C^t$ to a hidden representation $d_i^{1:T} = \psi_d(x_i, y_i^{1:T})$ that serves as value embedding in cross-attention. Then by using the context and target input $x_i$ as key and query embeddings, we apply a cross-attention along the example axis (per-task).

$$\{r_i^{1:T}\}_{i \in \mathcal{I}(D)} = Attn(\{x_i\}_{i \in \mathcal{I}(D)}, \{x_i\}_{i \in \mathcal{I}(C)}, \{d_i^{1:T}\}_{i \in \mathcal{I}(D)}). \tag{55}$$

**Decoders** Finally, the decoder produces predictive distribution for each joint target output $y_i^{1:T}$. We first project the input to $w_i = \psi_w(x_i)$, then concatenate it with the global latent variable $z$ and deterministic representation $r_i^{1:T}$. To compute the output distribution, we first apply two MLPs on the triple $(w_i, z, r_i^{1:T})$.

$$\mu_i = \psi_{(y,1)}(w_i, z, r_i^{1:T}) \tag{56}$$
$$\sigma_i^2 = \psi_{(y,2)}(w_i, z, r_i^{1:T})). \tag{57}$$

Then for each dimension, we construct the predictive distributions as Normal or Categorical, depending on the corresponding task type.

$$y_i^t \sim \begin{cases} \mathcal{N}((\mu_i)_{\mathcal{Y}^t}, (\sigma_i^2)_{\mathcal{Y}^t}), \text{if } y_i^t \text{ is continuous,} \\ \text{Categorical}((\mu_i)_{\mathcal{Y}^t}), \text{if } y_i^t \text{ is discrete,} \end{cases} \tag{58}$$

where $(\mu_i)_{\mathcal{Y}^t}$ (or $(\sigma_i^2)_{\mathcal{Y}^t}$)) denotes the projection of $\mu_i$ (or $\sigma_i^2$) into the task-specific output space $\mathcal{Y}^t$, by indexing the corresponding dimension from $\mu_i$. For example, if all tasks are one-dimensional, then this corresponds to selecting $t$-th coordinate of $\mu_i$.

### B.3 ANP Model for STNP

This section presents a detailed description of the STNP architecture used in the experiments. The STNP consists of $T$ independent ANPs, which consists of $T$ latent encoders, $T$ deterministic encoders, and $T$ decoders. Then STNP produces target output distribution by conditioning on the context set $C = \bigcup_{t \in \mathcal{T}} C^t$ where $C^t = \{(x_i, y_i^t)\}_{t \in \mathcal{T}}$. In the following, we deonte a stacked multi-head attention block (Parmar et al., 2018) by $Attn(Q, K, V)$ and a MLP by $\psi(x)$, as in Section 3.3.

**Latent Encoders** The latent encoders sample per-task latents $v^{1:T} = (v^1, \cdots, v^T)$. For each context example $(x_i, y_i^t) \in C^t$, we first project it to a hidden representation $s_i^t = \psi_s^t(x_i, y_i^t)$ using a MLP $\psi_s^t$ specific to task $f^t$. Then we aggregate them to a task-specific representation $s^t$ via self-attention followed by a pooling operation.

$$s^t = pool(Attn(\{s_i^t\}_{i \in \mathcal{I}(C^t)}, \{s_i^t\}_{i \in \mathcal{I}(C^t)}, \{s_i^t\}_{i \in \mathcal{I}(C^t)})), \quad \forall t \in \mathcal{T}. \tag{59}$$

Note that each attention is applied along the example axis (per-task) and independent to each task. Then the per-task latent variables are sampled independently, via MLPs.

$$v^t \sim q_\phi(v^t|C^t) = \mathcal{N}(\psi_{(v^t,1)}(s^t), \psi_{(v^t,2)}(s^t)), \quad \forall t \in \mathcal{T}. \tag{60}$$

**Deterministic Encoders** The deterministic encoders produce local representation $r_i^t$ for each $i \in \mathcal{D}$ and $t \in \mathcal{T}$. We first project each context example $(x_i, y_i^t) \in C^t$ to a hidden representation $d_i^t = \psi_d^t(x_i, y_i^t)$ that serves as value embedding in cross-attention. Then by using the context and target input $x_i$ as key and query embeddings, we apply $T$ independent cross-attention along the example axis (per-task).

$$\{r_i^t\}_{i \in \mathcal{I}(D^t)} = Attn(\{x_i^t\}_{i \in \mathcal{I}(D^t)}, \{x_i^t\}_{i \in \mathcal{I}(C^t)}, \{d_i^t\}_{i \in \mathcal{I}(D^t)}), \quad \forall t \in \mathcal{T}. \tag{61}$$

**Decoders** Finally, the decoders produce predictive distributions for each target output $y_i^t$. We first project the input to $w_i^t = \psi_w^t(x_i)$, then concatenate it with the corresponding latent variable $v^t$ and deterministic representation $r_i^t$. The output distribution is computed similar to MTNP described in Section 3.3.

$$y_i^t \sim \begin{cases} \mathcal{N}(\psi_{(y^t,1)}(w_i^t, v^t, r_i^t), \psi_{(y^t,2)}(w_i^t, v^t, r_i^t)), & \text{if } y_i^t \text{ is continuous,} \\ \text{Categorical}(\psi_{(y^t,1)}(w_i^t, v^t, r_i^t)), & \text{if } y_i^t \text{ is discrete,} \end{cases} \quad \forall i \in \mathcal{I}(D^t), \forall t \in \mathcal{T}. \tag{62}$$

We use the hidden dimension $d = 128$ for all models in synthetic and CelebA experiments and $d = 64$ and in weather experiments.

### B.4 Architectural Hyper-Parameters

Table 5 summarizes the number of layers for each module in three models (STNP, JTNP, MTNP) used in the experiments. We use the same number of layers in all experiments.

| Module | STNP | JTNP | MTNP |
|---|---|---|---|
| $\psi_s$ (or $\psi_s^t$) | 3 | 3 | 3 |
| $\psi_d$ (or $\psi_d^t$) | 3 | 3 | 3 |
| $\psi_w$ (or $\psi_w^t$) | 1 | 1 | 1 |
| $\psi_y$ (or $\psi_y^t$) | 5 | 5 | 5 |
| per-task *Attn* (in STNP, MTNP) | 3 | - | 3 |
| global *Attn* (in JTNP) | - | 3 | - |
| across-task *Attn* (in MTNP) | - | - | 2 |

Table 5: Number of layers of each module in STNP, JTNP, and MTNP used in the experiments.

Table 6 summarizes the hidden dimension $dim_{\text{hidden}}$ of all models used in each experiment. In our implementation, all layers (including the positional embedding) except the input and output layers have the dimension $dim_{\text{hidden}}$.

| Dataset | STNP | JTNP | MTNP |
|---------|------|------|------|
| Synthetic | 128 | 128 | 128 |
| Weather | 64 | 64 | 64 |
| Face | 128 | 128 | 128 |

Table 6: Hidden dimensions of the models used in the experiments.

## B.5 TRAINING DETAILS

For all three models, we schedule learning rate lr by

$$\text{lr} = \text{base\_lr} \times 1000^{0.5} \times \min(\text{n\_iters} \times 1,000^{-1.5}, \text{n\_iters}^{-0.5}), \tag{63}$$

where n_iters is the number of total iterations and base_lr is the base learning rate. We also introduce *beta* coefficient on the ELBO objective following Higgins et al. (2017), which is multiplied by each KL term. The beta coefficient is scheduled to be linearly increased from 0 to 1 during the first 10000 iters, then fixed to 1. We summarize the training hyper-parameters of models used in the experiments in Table 7.

| Dataset | n_iters | base_lr | batch size |
|---------|---------|---------|------------|
| Synthetic | 300000 | 0.00025 | 24 |
| Weather | 50000 | 0.00025 | 16 |
| Face | 300000 | 0.0005 | 16 |

Table 7: Training hyper-parameters used in the experiments.

## B.6 PARAMETER SHARING IN MTNP

The overall description of our neural network model for MTNP is provided in Section 3.3. We use different parameter sharing techniques in the datasets, depending on whether the tasks are homogeneous or not. In synthetic and weather tasks, all output values are one-dimensional. Thus we tie the parameters of the per-task paths in encoder and decoder, which makes more efficient parametrization compared to per-task encoders and decoders. In visual tasks, however, the tasks have different output dimensionalities. Thus in this case, we separate the parameters of all per-task paths. As task identity is implicitly encoded by the separation of task-specific paths, we do not employ task embeddings

## B.7 OTHER BASELINES

We include two Multi-Output Gaussian Process (MOGP) baselines, MOSM (Parra & Tobar, 2017) and CSM (Ulrich et al., 2015). To make use of training set of tasks, we consider pretraining MOGPs with respect to the kernel parameters using the same meta-training dataset with MTNP, and transfer the learned kernel parameters as prior in meta-testing. This allows both MOGPs and MTNPs to be trained and evaluated under the same setting. To prevent overfitting, we early-stopped the pretraining based on NLL. We observe that such pretraining is effective in synthetic tasks but not in weather tasks, thus we report the pretrained version for results on synthetic tasks and non-pretrained version for results on weather tasks.

We also include two gradient-based meta-learning baselines, MAML (Finn et al., 2017) and Reptile (Nichol et al., 2018) that use the same meta-train/meta-test data with our method. We chose these models as they are model-agnostic meta-learning methods that can be applied to our multi-task regression setting with incomplete data. Applied to our problem, the meta-training involves bi-level optimization where the inner loop optimizes the loss for context data and the outer loop optimizes the loss for target data. We employ a similar architecture to MTNP for the baselines that consists of a 4-layer MLP encoder network shared by all tasks and task-specific 4-layer MLP decoder networks. For fair comparisons, we controlled the total number of parameters of the models similar to NP baselines (STNP, JTNP, MTNP).

## C  EXPERIMENTAL DETAILS OF 1D SYNTHETIC FUNCTION REGRESSION

In this section, we describe details of the data generating process and experimental settings of 1D function regression on synthetic tasks.

### C.1  SYNTHETIC DATASET

As discussed in the paper, we simulate synthetic tasks which are correlated by a set of parameters $a, b, c, w \in \mathbb{R}$ as follow:

$$f^t(x_i) = a \cdot \text{act}_t(wx_i + b) + c, \quad \text{act}_t \in \{\text{Sine}, \text{Tanh}, \text{Sigmoid}, \text{Gaussian}\}, \tag{64}$$

where Sine, Tanh, Sigmoid are sine, hyperbolic tangent, logistic sigmoid function, respectively, and Gaussian$(x)$ is defined as $\exp(-x^2)$. Rather than sharing the exactly same parameters $a, b, c, w$ across tasks, we add a task-specific noise to each parameter, to control the amount of correlation across tasks as follow:

$$\alpha^t = \alpha + \epsilon, \ \epsilon \sim \mathcal{N}(0, 0.1), \ \forall t \in \mathcal{T}, \ \forall \alpha \in \{a, b, c, w\}. \tag{65}$$

Thus in fact the input-output pairs of each task is generated as follow:

$$f^t(x_i) = a^t \cdot \text{act}_t(w^t x_i + b^t) + c^t, \quad \text{act}_t \in \{\text{Sine}, \text{Tanh}, \text{Sigmoid}, \text{Gaussian}\}, \tag{66}$$

We split the 1,000 functions into 800 training, 100 validation, and 100 test sets of four correlated tasks. Then we construct a training dataset, a validataion dataset, and a test dataset using the corresponding set of generated tasks. For each training and validation data, we sample 200 input points uniformly within the interval $[-5, 5]$, and applied the corresponding tasks to generate multi-task output values. For each test data, we choose 1000 input points in the uniform grid of the interval $[-5, 5]$, and generate the multi-task output values similarly. Finally, simulating the incomplete data is achieved by randomly dropping each output value $y_i^t$ with probability $\gamma \in [0, 1]$.

### C.2  EVALUATION PROTOCOL

For evaluation, we average the normalized MSE $MSE = \frac{1}{n}\sum_{i=1}^n (y_i^t - \hat{y}_i^t)^2 / a^2$ on test dataset. [2] For prediction $\hat{Y}^{1:4}$, we approximate the predictive posterior mean with Monte Carlo sampling. For example in MTNP,

$$p(y_i^t | x_i, C) = \int\int p(y_i^t | x_i, v^t) p(v^t | z, C^t) p(z | C) dv^t dz \tag{67}$$

$$\approx \frac{1}{NM} \sum_{k=1}^N \sum_{l=1}^M p(y_i^t | x_i, v_{k,l}^t), \text{ where } v_{k,l}^t \overset{\text{i.i.d.}}{\sim} p(v^t | z_k, C^t), \ z_k \overset{\text{i.i.d.}}{\sim} p(z | C). \tag{68}$$

We use $N = M = 5$, resulting total 25 samples. For STNP (or JTNP), we sample each latent $v^t$ (or $z$) 5 times, since there is no hierarchy. Since all the output distributions are Gaussian, the posterior predictive mean can be computed by averaging the means of each sample distribution $p(y_i^t | x_i, v_{k,l}^t)$. To plot the predictions in Figure 3 (a), we use the posterior means for both $z$ and $v^t$ (which corresponds to the Maximum A Posteriori estimation) and plot the mean and variance of resulting $p(y_i^t | x_i, v^t)$.

---

[2] We normalize the MSE for fair consideration of difference in amplitude $a$ across different functions.

# D ADDITIONAL RESULTS ON SYNTHETIC 1D FUNCTION REGRESSION

## D.1 ADDITIONAL RESULTS ON COMPLETE AND PARTIALLY INCOMPLETE DATA

In this section, we provide additional results on the synthetic experiment, with various missing rates $\gamma$ and also with standard deviation from 5 different random seeds. When the data is incomplete and missing some task labels (i.e., $\gamma = 0.25, 0.5, 0.75$), we can see that MTNP clearly outperforms the baselines in almost all cases. When the complete data ($\gamma = 0$) is given, MTNP still outperforms almost all baselines while achieves at least competitive performance to JTNP. Figure 6 and 7 shows that MTNP is the most robust against both context size and quality (incompleteness).

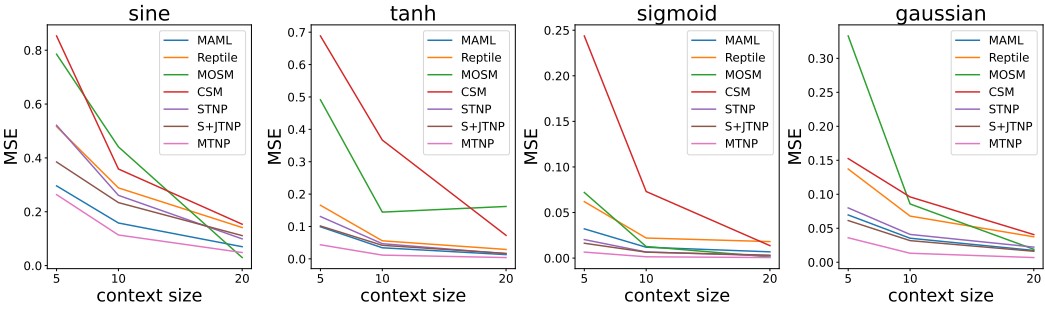

Figure 6: Performance (normalized MSE) of models against various context sizes ($m$). Missing rate ($\gamma$) is fixed to 0.5.

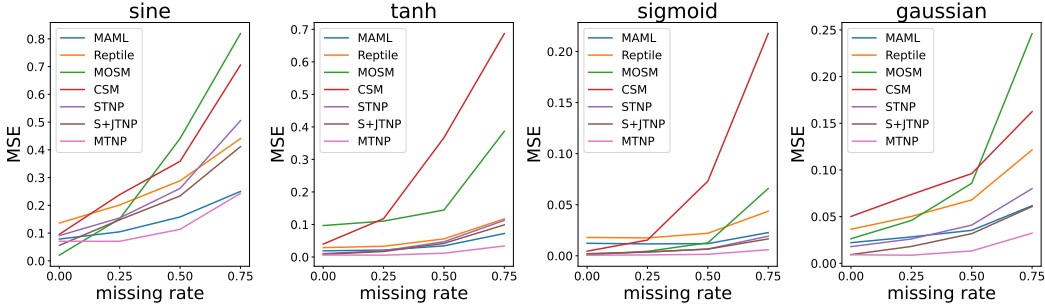

Figure 7: Performance (normalized MSE) of models against various missing rates ($\gamma$). Context size ($m$) is fixed to 10.

Table 8: Average normalized MSE on synthetic tasks, with varying context size ($m$) and $\gamma = 0$.

| task | Sine | | | Tanh | | |
|------|------|------|------|------|------|------|
| $m$ | 5 | 10 | 20 | 5 | 10 | 20 |
| MAML | $0.1943 \pm 0.0194$ | $0.0786 \pm 0.0049$ | $0.0342 \pm 0.0034$ | $0.0569 \pm 0.0063$ | $0.0190 \pm 0.0007$ | $0.0117 \pm 0.0008$ |
| Reptile | $0.3443 \pm 0.0161$ | $0.1362 \pm 0.0172$ | $0.0534 \pm 0.0043$ | $0.0918 \pm 0.0074$ | $0.0288 \pm 0.0020$ | $0.0163 \pm 0.0019$ |
| MOSM | $0.4609 \pm 0.0571$ | $\mathbf{0.0204} \pm 0.0125$ | $\mathbf{0.0002} \pm 0.0001$ | $0.1509 \pm 0.0362$ | $0.0966 \pm 0.0460$ | $0.0212 \pm 0.0113$ |
| CSM | $0.3124 \pm 0.0233$ | $0.0956 \pm 0.0149$ | $0.0095 \pm 0.0029$ | $0.1678 \pm 0.0263$ | $0.0396 \pm 0.0107$ | $0.0068 \pm 0.0034$ |
| STNP | $0.2562 \pm 0.0114$ | $0.0910 \pm 0.0104$ | $0.0200 \pm 0.0020$ | $0.0502 \pm 0.0071$ | $0.0104 \pm 0.0023$ | $0.0024 \pm 0.0004$ |
| JTNP | $\mathbf{0.1213} \pm 0.0095$ | $0.0560 \pm 0.0039$ | $0.0291 \pm 0.0011$ | $\mathbf{0.0210} \pm 0.0038$ | $0.0079 \pm 0.0004$ | $0.0057 \pm 0.0004$ |
| MTNP | $0.1793 \pm 0.0191$ | $0.0705 \pm 0.0063$ | $0.0186 \pm 0.0027$ | $0.0287 \pm 0.0027$ | $\mathbf{0.0060} \pm 0.0017$ | $\mathbf{0.0015} \pm 0.0002$ |

| task | Sigmoid | | | Gaussian | | |
|------|------|------|------|------|------|------|
| $m$ | 5 | 10 | 20 | 5 | 10 | 20 |
| MAML | $0.0239 \pm 0.0029$ | $0.0122 \pm 0.0010$ | $0.0102 \pm 0.0006$ | $0.0473 \pm 0.0035$ | $0.0221 \pm 0.0029$ | $0.0112 \pm 0.0007$ |
| Reptile | $0.0391 \pm 0.0043$ | $0.0179 \pm 0.0019$ | $0.0125 \pm 0.0013$ | $0.0879 \pm 0.0039$ | $0.0365 \pm 0.0034$ | $0.0171 \pm 0.0016$ |
| MOSM | $0.0090 \pm 0.0022$ | $0.0008 \pm 0.0005$ | $\mathbf{0.0001} \pm 0.0001$ | $0.1200 \pm 0.0098$ | $0.0263 \pm 0.0097$ | $0.0012 \pm 0.0005$ |
| CSM | $0.0366 \pm 0.0114$ | $0.0045 \pm 0.0020$ | $0.0006 \pm 0.0007$ | $0.0948 \pm 0.0114$ | $0.0502 \pm 0.0145$ | $0.0064 \pm 0.0008$ |
| STNP | $0.0064 \pm 0.0029$ | $0.0017 \pm 0.0002$ | $0.0008 \pm 0.0000$ | $0.0393 \pm 0.0048$ | $0.0179 \pm 0.0030$ | $0.0071 \pm 0.0006$ |
| JTNP | $0.0041 \pm 0.0008$ | $0.0020 \pm 0.0002$ | $0.0016 \pm 0.0001$ | $\mathbf{0.0170} \pm 0.0015$ | $0.0093 \pm 0.0007$ | $0.0072 \pm 0.0004$ |
| MTNP | $\mathbf{0.0030} \pm 0.0010$ | $\mathbf{0.0006} \pm 0.0001$ | $0.0003 \pm 0.0000$ | $0.0197 \pm 0.0008$ | $\mathbf{0.0092} \pm 0.0015$ | $\mathbf{0.0043} \pm 0.0009$ |

Table 9: Average normalized MSE on synthetic tasks, with varying context size ($m$) and $\gamma = 0.25$.

| task | Sine | | | Tanh | | |
|------|------|------|------|------|------|------|
| $m$ | 5 | 10 | 20 | 5 | 10 | 20 |
| MAML | $0.2448 \pm 0.0167$ | $0.1047 \pm 0.0069$ | $0.0438 \pm 0.0006$ | $0.0746 \pm 0.0095$ | $0.0213 \pm 0.0007$ | $0.0109 \pm 0.0009$ |
| Reptile | $0.4222 \pm 0.0304$ | $0.2015 \pm 0.0347$ | $0.0737 \pm 0.0030$ | $0.1244 \pm 0.0096$ | $0.0329 \pm 0.0012$ | $0.0153 \pm 0.0012$ |
| MOSM | $0.6531 \pm 0.0973$ | $0.1514 \pm 0.0562$ | $\mathbf{0.0038} \pm 0.0028$ | $0.2490 \pm 0.0505$ | $0.1104 \pm 0.0251$ | $0.0701 \pm 0.0766$ |
| CSM | $0.5096 \pm 0.0688$ | $0.2387 \pm 0.0524$ | $0.0516 \pm 0.0159$ | $0.4401 \pm 0.1387$ | $0.1179 \pm 0.0265$ | $0.0163 \pm 0.0033$ |
| STNP | $0.3768 \pm 0.0152$ | $0.1547 \pm 0.0145$ | $0.0492 \pm 0.0060$ | $0.0711 \pm 0.0077$ | $0.0191 \pm 0.0033$ | $0.0070 \pm 0.0013$ |
| S+JTNP | $0.2906 \pm 0.0241$ | $0.1481 \pm 0.0049$ | $0.0738 \pm 0.0066$ | $0.0531 \pm 0.0073$ | $0.0169 \pm 0.0008$ | $0.0098 \pm 0.0004$ |
| MTNP | $\mathbf{0.1871} \pm 0.0211$ | $\mathbf{0.0705} \pm 0.0027$ | $0.0297 \pm 0.0026$ | $\mathbf{0.0300} \pm 0.0029$ | $\mathbf{0.0055} \pm 0.0011$ | $\mathbf{0.0023} \pm 0.0002$ |

| task | Sigmoid | | | Gaussian | | |
|------|------|------|------|------|------|------|
| $m$ | 5 | 10 | 20 | 5 | 10 | 20 |
| MAML | $0.0255 \pm 0.0033$ | $0.0116 \pm 0.0012$ | $0.0087 \pm 0.0011$ | $0.0563 \pm 0.0018$ | $0.0283 \pm 0.0033$ | $0.0134 \pm 0.0007$ |
| Reptile | $0.0498 \pm 0.0076$ | $0.0175 \pm 0.0019$ | $0.0116 \pm 0.0004$ | $0.1112 \pm 0.0072$ | $0.0502 \pm 0.0044$ | $0.0217 \pm 0.0022$ |
| MOSM | $0.0243 \pm 0.0056$ | $0.0044 \pm 0.0016$ | $\mathbf{0.0002} \pm 0.0002$ | $0.1566 \pm 0.0310$ | $0.0459 \pm 0.0147$ | $0.0059 \pm 0.0036$ |
| CSM | $0.1315 \pm 0.0736$ | $0.0154 \pm 0.0011$ | $0.0010 \pm 0.0004$ | $0.1213 \pm 0.0228$ | $0.0737 \pm 0.0071$ | $0.0180 \pm 0.0063$ |
| STNP | $0.0121 \pm 0.0038$ | $0.0036 \pm 0.0004$ | $0.0013 \pm 0.0001$ | $0.0532 \pm 0.0072$ | $0.0260 \pm 0.0037$ | $0.0150 \pm 0.0021$ |
| S+JTNP | $0.0097 \pm 0.0017$ | $0.0038 \pm 0.0002$ | $0.0022 \pm 0.0001$ | $0.0403 \pm 0.0055$ | $0.0181 \pm 0.0005$ | $0.0112 \pm 0.0006$ |
| MTNP | $\mathbf{0.0040} \pm 0.0017$ | $\mathbf{0.0008} \pm 0.0001$ | $0.0004 \pm 0.0000$ | $\mathbf{0.0234} \pm 0.0025$ | $\mathbf{0.0087} \pm 0.0012$ | $\mathbf{0.0048} \pm 0.0003$ |

Table 10: Average normalized MSE on synthetic tasks, with varying context size ($m$) and $\gamma = 0.5$.

| task | Sine | | | Tanh | | |
|---|---|---|---|---|---|---|
| $m$ | 5 | 10 | 20 | 5 | 10 | 20 |
| MAML | $0.2962 \pm 0.0140$ | $0.1582 \pm 0.0052$ | $0.0701 \pm 0.0055$ | $0.0991 \pm 0.0085$ | $0.0342 \pm 0.0032$ | $0.0131 \pm 0.0023$ |
| Reptile | $0.5164 \pm 0.0167$ | $0.2886 \pm 0.0254$ | $0.1414 \pm 0.0431$ | $0.1656 \pm 0.0142$ | $0.0557 \pm 0.0033$ | $0.0291 \pm 0.0191$ |
| MOSM | $0.7852 \pm 0.1127$ | $0.4410 \pm 0.1269$ | $\mathbf{0.0298} \pm 0.0172$ | $0.4912 \pm 0.0706$ | $0.1444 \pm 0.0386$ | $0.1618 \pm 0.1999$ |
| CSM | $0.8529 \pm 0.2216$ | $0.3587 \pm 0.0395$ | $0.1537 \pm 0.0310$ | $0.6884 \pm 0.0841$ | $0.3669 \pm 0.0799$ | $0.0726 \pm 0.0251$ |
| STNP | $0.5212 \pm 0.0157$ | $0.2609 \pm 0.0382$ | $0.0993 \pm 0.0182$ | $0.1307 \pm 0.0134$ | $0.0468 \pm 0.0074$ | $0.0159 \pm 0.0028$ |
| S+JTNP | $0.3848 \pm 0.0203$ | $0.2340 \pm 0.0169$ | $0.1114 \pm 0.0084$ | $0.1015 \pm 0.0160$ | $0.0418 \pm 0.0066$ | $0.0168 \pm 0.0026$ |
| MTNP | $\mathbf{0.2636} \pm 0.0105$ | $\mathbf{0.1137} \pm 0.0078$ | $0.0485 \pm 0.0034$ | $\mathbf{0.0435} \pm 0.0047$ | $\mathbf{0.0115} \pm 0.0021$ | $\mathbf{0.0040} \pm 0.0002$ |

| task | Sigmoid | | | Gaussian | | |
|---|---|---|---|---|---|---|
| $m$ | 5 | 10 | 20 | 5 | 10 | 20 |
| MAML | $0.0321 \pm 0.0053$ | $0.0119 \pm 0.0014$ | $0.0069 \pm 0.0006$ | $0.0696 \pm 0.0033$ | $0.0353 \pm 0.0013$ | $0.0174 \pm 0.0024$ |
| Reptile | $0.0619 \pm 0.0089$ | $0.0220 \pm 0.0016$ | $0.0181 \pm 0.0148$ | $0.1371 \pm 0.0087$ | $0.0679 \pm 0.0039$ | $0.0374 \pm 0.0202$ |
| MOSM | $0.0720 \pm 0.0160$ | $0.0127 \pm 0.0049$ | $0.0013 \pm 0.0005$ | $0.3329 \pm 0.1578$ | $0.0857 \pm 0.0105$ | $0.0190 \pm 0.0064$ |
| CSM | $0.2437 \pm 0.0753$ | $0.0730 \pm 0.0413$ | $0.0137 \pm 0.0167$ | $0.1525 \pm 0.0402$ | $0.0961 \pm 0.0151$ | $0.0407 \pm 0.0079$ |
| STNP | $0.0203 \pm 0.0034$ | $0.0067 \pm 0.0013$ | $0.0025 \pm 0.0005$ | $0.0799 \pm 0.0098$ | $0.0409 \pm 0.0041$ | $0.0222 \pm 0.0042$ |
| S+JTNP | $0.0163 \pm 0.0024$ | $0.0065 \pm 0.0015$ | $0.0032 \pm 0.0004$ | $0.0613 \pm 0.0045$ | $0.0318 \pm 0.0021$ | $0.0161 \pm 0.0019$ |
| MTNP | $\mathbf{0.0066} \pm 0.0019$ | $\mathbf{0.0014} \pm 0.0001$ | $\mathbf{0.0006} \pm 0.0001$ | $\mathbf{0.0360} \pm 0.0018$ | $\mathbf{0.0132} \pm 0.0008$ | $\mathbf{0.0069} \pm 0.0012$ |

Table 11: Average normalized MSE on synthetic tasks, with varying context size ($m$) and $\gamma = 0.75$.

| task | Sine | | | Tanh | | |
|---|---|---|---|---|---|---|
| $m$ | 5 | 10 | 20 | 5 | 10 | 20 |
| MAML | $0.3972 \pm 0.0122$ | $0.2501 \pm 0.0133$ | $0.1401 \pm 0.0104$ | $0.1544 \pm 0.0255$ | $0.0722 \pm 0.0054$ | $0.0258 \pm 0.0047$ |
| Reptile | $0.6289 \pm 0.0200$ | $0.4404 \pm 0.0175$ | $0.2672 \pm 0.0283$ | $0.2200 \pm 0.0391$ | $0.1172 \pm 0.0167$ | $0.0426 \pm 0.0053$ |
| MOSM | $0.9726 \pm 0.0788$ | $0.8189 \pm 0.0861$ | $0.4288 \pm 0.1540$ | $0.7753 \pm 0.1060$ | $0.3867 \pm 0.0620$ | $0.1464 \pm 0.0373$ |
| CSM | $0.8747 \pm 0.1166$ | $0.7057 \pm 0.0750$ | $0.4091 \pm 0.0384$ | $0.9140 \pm 0.1262$ | $0.6870 \pm 0.0831$ | $0.3036 \pm 0.0649$ |
| STNP | $0.7329 \pm 0.0581$ | $0.5053 \pm 0.0289$ | $0.2770 \pm 0.0286$ | $0.1975 \pm 0.0256$ | $0.1128 \pm 0.0111$ | $0.0443 \pm 0.0116$ |
| S+JTNP | $0.5807 \pm 0.0573$ | $0.4115 \pm 0.0348$ | $0.2521 \pm 0.0218$ | $0.1654 \pm 0.0195$ | $0.0989 \pm 0.0127$ | $0.0426 \pm 0.0115$ |
| MTNP | $\mathbf{0.3784} \pm 0.0395$ | $\mathbf{0.2432} \pm 0.0230$ | $\mathbf{0.1295} \pm 0.0172$ | $\mathbf{0.0838} \pm 0.0085$ | $\mathbf{0.0340} \pm 0.0020$ | $\mathbf{0.0118} \pm 0.0027$ |

| task | Sigmoid | | | Gaussian | | |
|---|---|---|---|---|---|---|
| $m$ | 5 | 10 | 20 | 5 | 10 | 20 |
| MAML | $0.0520 \pm 0.0045$ | $0.0227 \pm 0.0031$ | $0.0088 \pm 0.0022$ | $0.1045 \pm 0.0050$ | $0.0617 \pm 0.0036$ | $0.0332 \pm 0.0031$ |
| Reptile | $0.0857 \pm 0.0078$ | $0.0436 \pm 0.0126$ | $0.0191 \pm 0.0037$ | $0.1777 \pm 0.0154$ | $0.1215 \pm 0.0119$ | $0.0695 \pm 0.0045$ |
| MOSM | $0.1704 \pm 0.0362$ | $0.0658 \pm 0.0132$ | $0.0131 \pm 0.0037$ | $0.2663 \pm 0.0440$ | $0.2461 \pm 0.0512$ | $0.1005 \pm 0.0116$ |
| CSM | $0.3252 \pm 0.0741$ | $0.2176 \pm 0.0392$ | $0.0832 \pm 0.0384$ | $0.2413 \pm 0.0461$ | $0.1624 \pm 0.0074$ | $0.1066 \pm 0.0176$ |
| STNP | $0.0303 \pm 0.0038$ | $0.0191 \pm 0.0030$ | $0.0067 \pm 0.0008$ | $0.1232 \pm 0.0037$ | $0.0800 \pm 0.0094$ | $0.0469 \pm 0.0037$ |
| S+JTNP | $0.0260 \pm 0.0044$ | $0.0165 \pm 0.0021$ | $0.0070 \pm 0.0005$ | $0.0970 \pm 0.0101$ | $0.0608 \pm 0.0035$ | $0.0334 \pm 0.0023$ |
| MTNP | $\mathbf{0.0136} \pm 0.0024$ | $\mathbf{0.0059} \pm 0.0004$ | $\mathbf{0.0019} \pm 0.0004$ | $\mathbf{0.0643} \pm 0.0048$ | $\mathbf{0.0323} \pm 0.0016$ | $\mathbf{0.0155} \pm 0.0018$ |

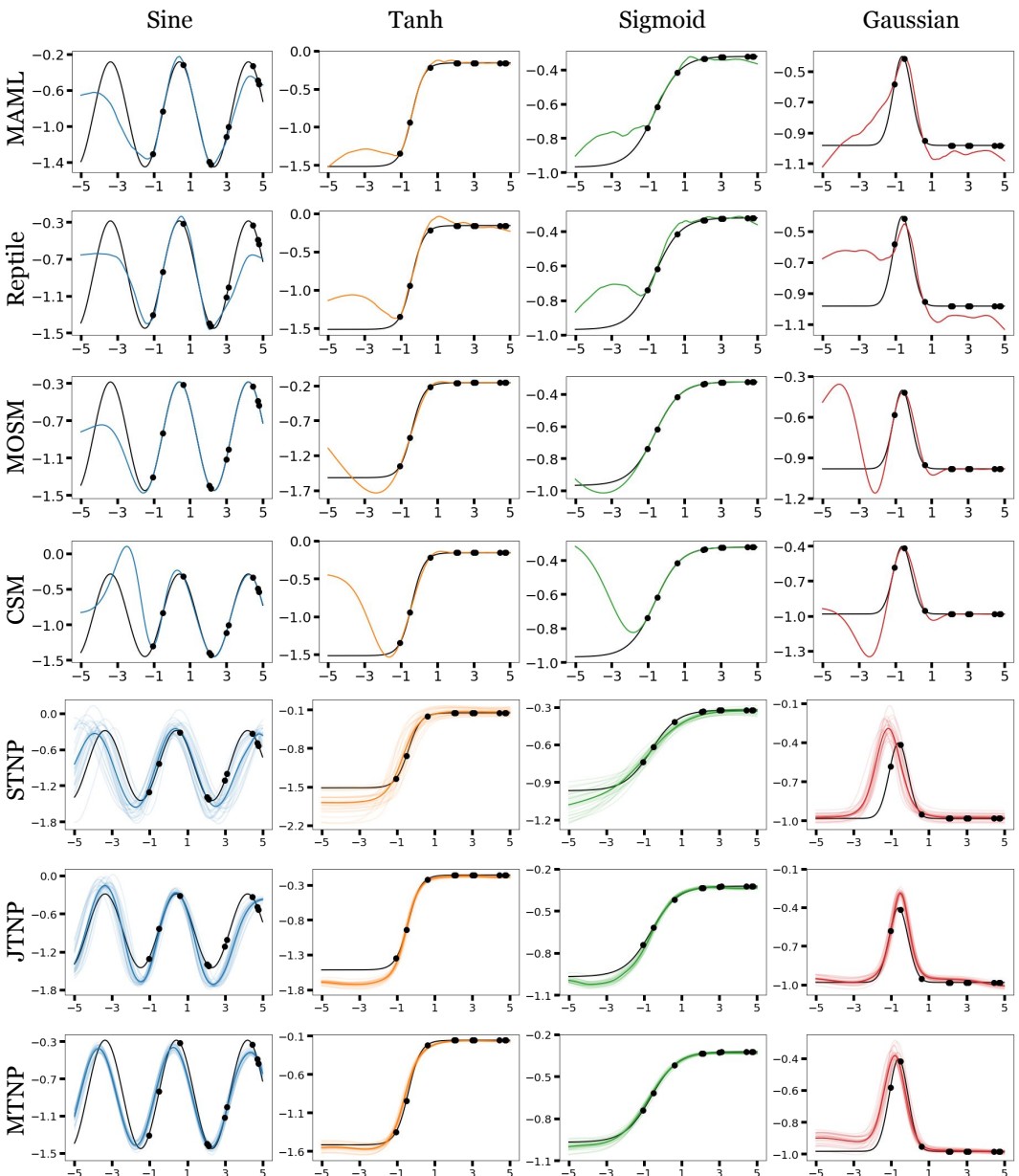

Figure 8: Qualitative results on synthetic task with $\gamma = 0$, $m = 10$. For latent variable models (STNP, JTNP, MTNP), we sample the latents 25 times and plot the mean prediction from each sample. For the other models, we plot the mean prediction.

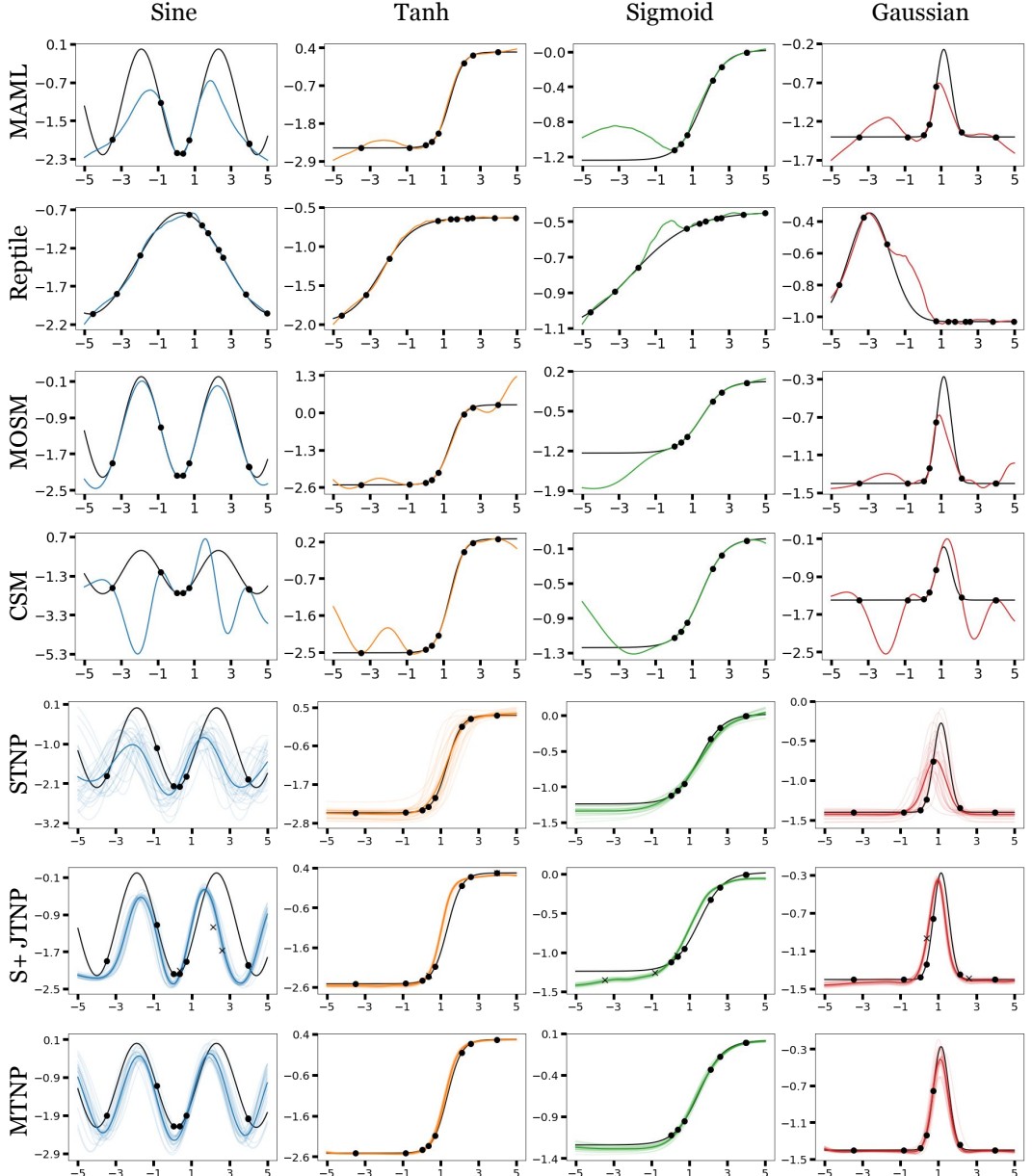

Figure 9: Qualitative results on synthetic task with $\gamma = 0.25$, $m = 10$. For latent variable models (STNP, JTNP, MTNP), we sample the latents 25 times and plot the mean prediction from each sample. For the other models, we plot the mean prediction.

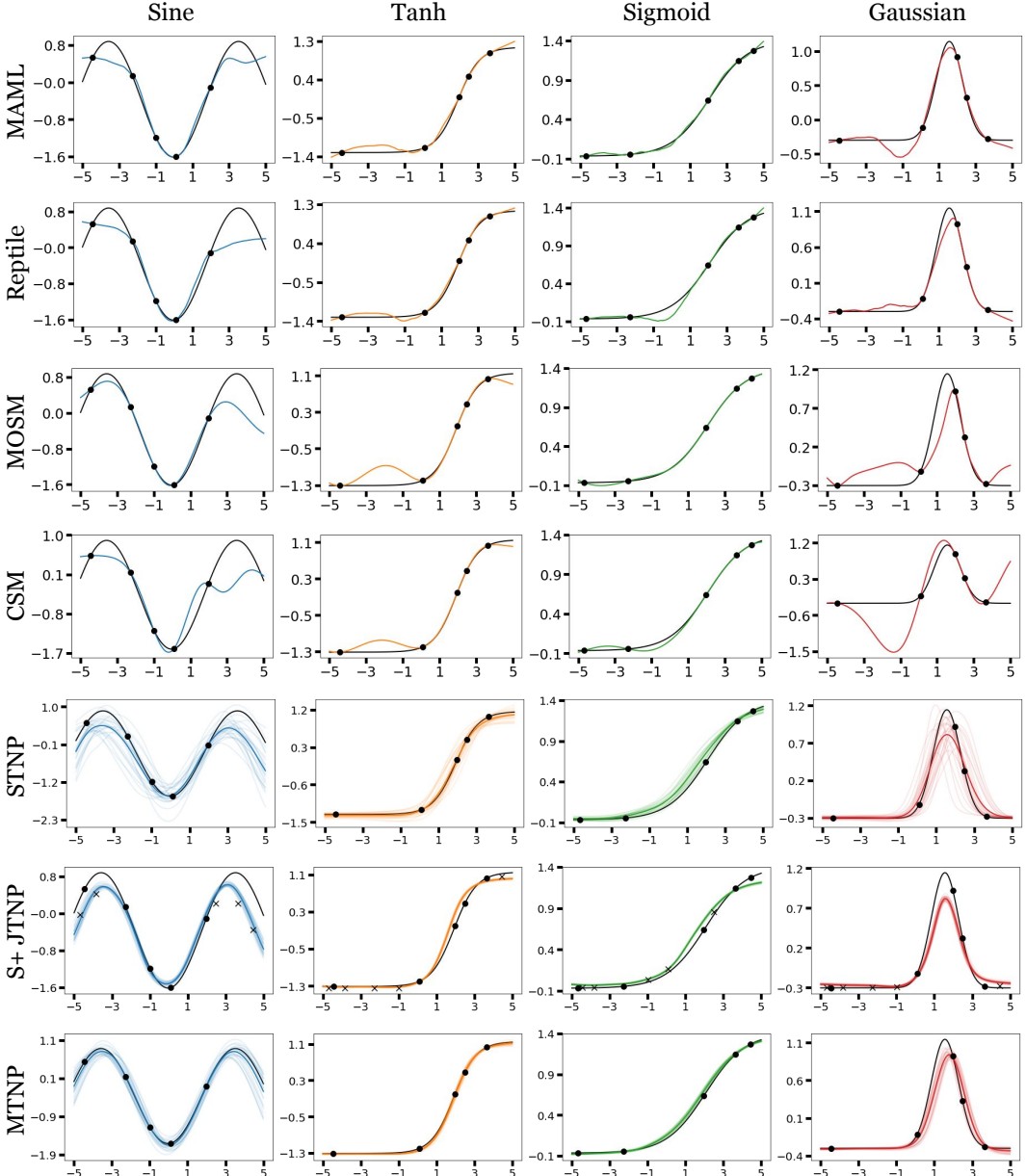

Figure 10: Qualitative results on synthetic task with $\gamma = 0.5$, $m = 10$. For latent variable models (STNP, JTNP, MTNP), we sample the latents 25 times and plot the mean prediction from each sample. For the other models, we plot the mean prediction.

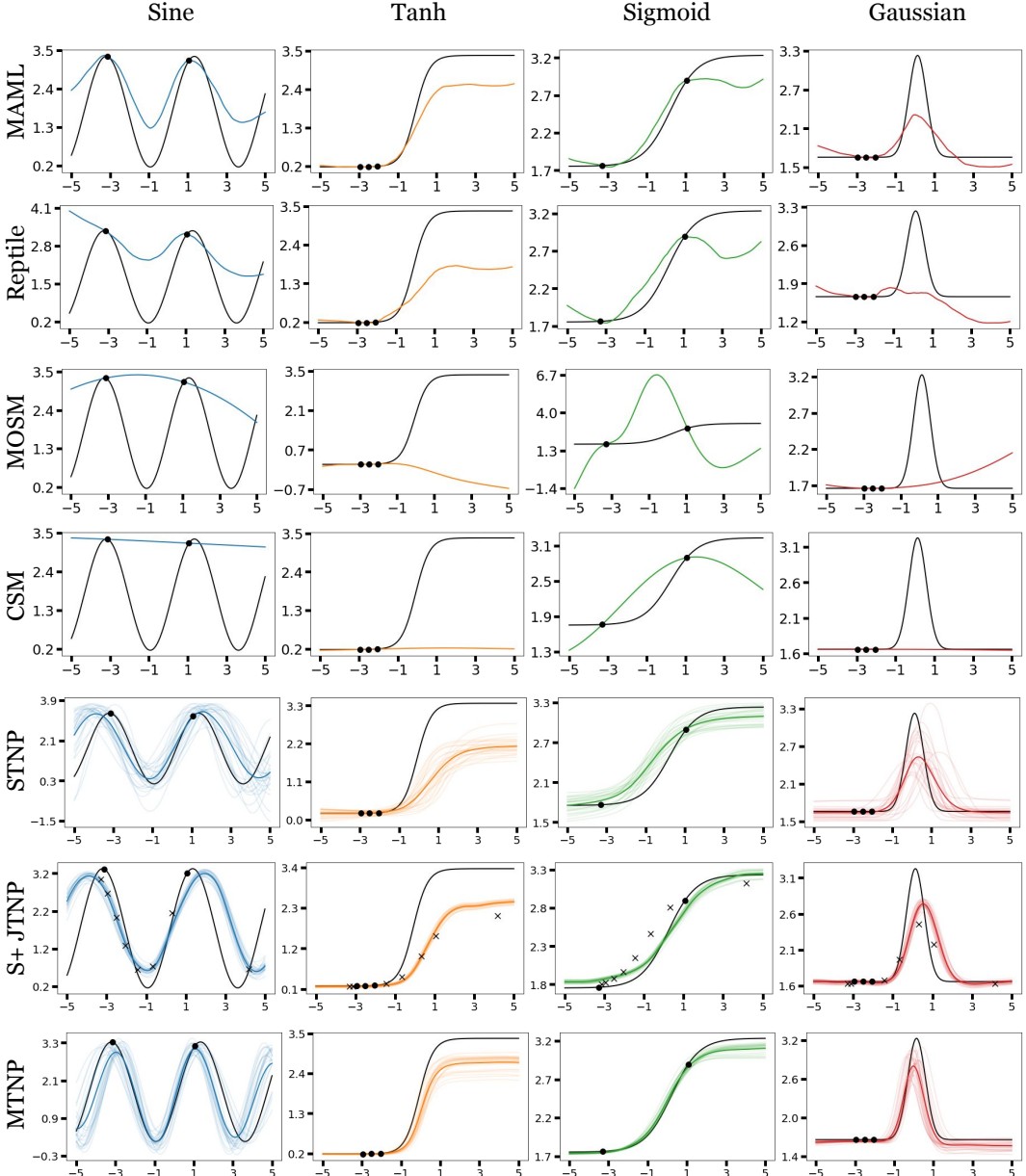

Figure 11: Qualitative results on synthetic task with $\gamma = 0.75$, $m = 10$. For latent variable models (STNP, JTNP, MTNP), we sample the latents 25 times and plot the mean prediction from each sample. For the other models, we plot the mean prediction.

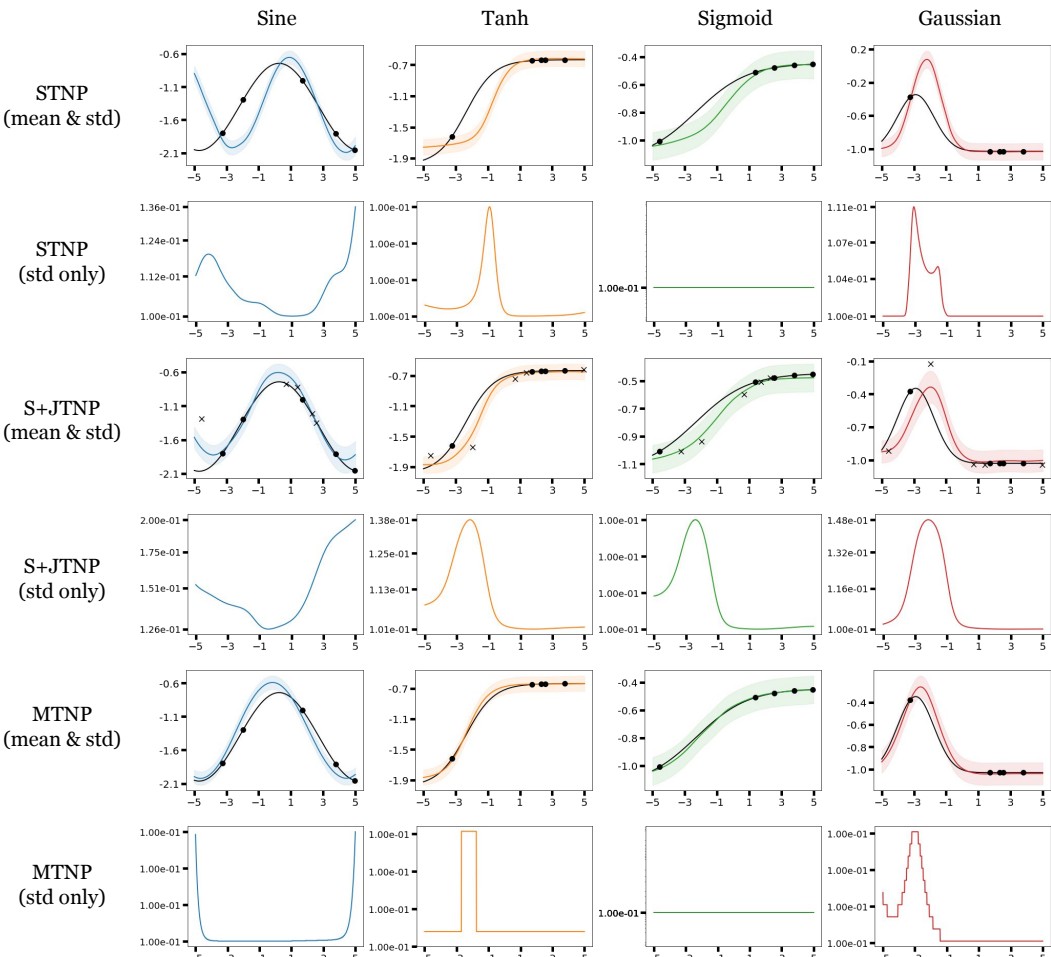

Figure 12: Predicted mean and standard deviation of STNP, S+JTNP, and MTNP with MAP estimations.

## D.2 ADDITIONAL RESULTS ON TOTALLY INCOMPLETE DATA

In this section, we explore the totally incomplete setting where no two task output values are observed at the same input point ($\mathcal{I}(C^t) \cap \mathcal{I}(C^{t'}) = \emptyset, \forall t \neq t'$), to further validate the practical effectiveness of our method. To generate the totally incomplete dataset, we randomly sample context input points for each task independently, then compute the corresponding output points according to the Eq. 65 and Eq. 66. Note that in this case we do not *drop* any output points, so that the context size is $m = |\mathcal{I}(C)| = \sum_{t \in \mathcal{T}} |\mathcal{I}(C^t)|$.

We evaluate the baselines and ours in two different scenarios: (1) training on partially incomplete or complete dataset then testing on totally incomplete dataset and (2) both training and testing on totally incomplete dataset. We observe that in both scenarios, MTNP outperforms the baselines.

Table 12: Average normalized MSE on synthetic tasks in totally incomplete scenario (1), with varying context size ($m$). All models are trained on partially incomplete data with $\gamma = 0.5$ but JTNP is trained with $\gamma = 0$.

| task | Sine | | | Tanh | | |
|---|---|---|---|---|---|---|
| $m$ | 5 | 10 | 20 | 5 | 10 | 20 |
| MAML | $0.6123 \pm 0.0221$ | $0.3715 \pm 0.0231$ | $0.1930 \pm 0.0175$ | $0.4117 \pm 0.0413$ | $0.1230 \pm 0.0143$ | $0.0308 \pm 0.0068$ |
| Reptile | $0.8089 \pm 0.0378$ | $0.5702 \pm 0.0500$ | $0.3482 \pm 0.0447$ | $0.4495 \pm 0.0340$ | $0.1557 \pm 0.0197$ | $0.0487 \pm 0.0116$ |
| MOSM | $1.0718 \pm 0.0854$ | $0.6073 \pm 0.0297$ | $0.3728 \pm 0.1461$ | $1.5199 \pm 0.2179$ | $0.9601 \pm 0.0750$ | $0.5937 \pm 0.0805$ |
| CSM | $1.0887 \pm 0.0703$ | $0.6557 \pm 0.0794$ | $0.3462 \pm 0.0759$ | $1.4156 \pm 0.0876$ | $1.1087 \pm 0.1469$ | $0.8813 \pm 0.1811$ |
| STNP | $0.6491 \pm 0.2299$ | $0.4228 \pm 0.2466$ | $0.2116 \pm 0.1539$ | $0.1753 \pm 0.0890$ | $0.0843 \pm 0.0572$ | $0.0344 \pm 0.0262$ |
| S+JTNP | $0.5162 \pm 0.2061$ | $0.3472 \pm 0.1789$ | $0.1976 \pm 0.1154$ | $0.1411 \pm 0.0751$ | $0.0772 \pm 0.0544$ | $0.0327 \pm 0.0222$ |
| MTNP | $\mathbf{0.3777} \pm 0.1947$ | $\mathbf{0.2045} \pm 0.1290$ | $\mathbf{0.1007} \pm 0.0688$ | $\mathbf{0.0890} \pm 0.0685$ | $\mathbf{0.0293} \pm 0.0248$ | $\mathbf{0.0089} \pm 0.0064$ |

| task | Sigmoid | | | Gaussian | | |
|---|---|---|---|---|---|---|
| $m$ | 5 | 10 | 20 | 5 | 10 | 20 |
| MAML | $0.2220 \pm 0.0288$ | $0.0456 \pm 0.0066$ | $0.0112 \pm 0.0027$ | $0.2281 \pm 0.0147$ | $0.0911 \pm 0.0096$ | $0.0365 \pm 0.0071$ |
| Reptile | $0.3012 \pm 0.0328$ | $0.0630 \pm 0.0108$ | $0.0210 \pm 0.0053$ | $0.3734 \pm 0.0255$ | $0.1435 \pm 0.0194$ | $0.0725 \pm 0.0141$ |
| MOSM | $0.4798 \pm 0.1020$ | $0.2471 \pm 0.0184$ | $0.3070 \pm 0.1893$ | $0.3245 \pm 0.1298$ | $0.2190 \pm 0.0163$ | $0.2023 \pm 0.0318$ |
| CSM | $0.5256 \pm 0.0344$ | $0.6013 \pm 0.1190$ | $0.4581 \pm 0.0706$ | $0.2086 \pm 0.0420$ | $0.1815 \pm 0.0391$ | $0.1456 \pm 0.0151$ |
| STNP | $0.0271 \pm 0.0133$ | $0.0136 \pm 0.0092$ | $0.0050 \pm 0.0034$ | $0.0962 \pm 0.0325$ | $0.0608 \pm 0.0297$ | $0.0366 \pm 0.0198$ |
| S+JTNP | $0.0236 \pm 0.0129$ | $0.0122 \pm 0.0078$ | $0.0053 \pm 0.0028$ | $0.0819 \pm 0.0353$ | $0.0470 \pm 0.0240$ | $0.0263 \pm 0.0137$ |
| MTNP | $\mathbf{0.0152} \pm 0.0133$ | $\mathbf{0.0049} \pm 0.0045$ | $\mathbf{0.0014} \pm 0.0010$ | $\mathbf{0.0535} \pm 0.0268$ | $\mathbf{0.0246} \pm 0.0149$ | $\mathbf{0.0113} \pm 0.0058$ |

Table 13: Average normalized MSE on synthetic tasks in totally incomplete scenario (2), with varying context size ($m$). All models are trained on totally incomplete data but JTNP is trained with $\gamma = 0$.

| task | Sine | | | Tanh | | |
|------|------|------|------|------|------|------|
| $m$ | 5 | 10 | 20 | 5 | 10 | 20 |
| MAML | $0.5906 \pm 0.0249$ | $0.3967 \pm 0.0265$ | $0.2347 \pm 0.0205$ | $0.3366 \pm 0.0322$ | $0.1018 \pm 0.0133$ | $0.0317 \pm 0.0058$ |
| Reptile | $0.7461 \pm 0.0473$ | $0.5945 \pm 0.0616$ | $0.3798 \pm 0.0504$ | $0.3690 \pm 0.0318$ | $0.1202 \pm 0.0164$ | $0.0412 \pm 0.0089$ |
| MOSM | $0.9080 \pm 0.3979$ | $0.6939 \pm 0.4823$ | $0.6504 \pm 0.5446$ | $0.7423 \pm 0.5953$ | $0.6695 \pm 0.5833$ | $0.6132 \pm 0.5515$ |
| CSM | $0.9501 \pm 0.1520$ | $0.7644 \pm 0.2126$ | $0.6486 \pm 0.2249$ | $0.7902 \pm 0.3731$ | $0.7175 \pm 0.3494$ | $0.7691 \pm 0.2691$ |
| STNP | $0.7683 \pm 0.0064$ | $0.5112 \pm 0.0322$ | $0.3111 \pm 0.0086$ | $0.2649 \pm 0.0089$ | $0.1642 \pm 0.0103$ | $0.0679 \pm 0.0065$ |
| S+JTNP | $0.7591 \pm 0.0407$ | $0.5058 \pm 0.0202$ | $0.2848 \pm 0.0258$ | $0.2348 \pm 0.0278$ | $0.1486 \pm 0.0164$ | $0.0593 \pm 0.0070$ |
| MTNP | $\mathbf{0.4860} \pm 0.0129$ | $\mathbf{0.3391} \pm 0.0281$ | $\mathbf{0.1808} \pm 0.0176$ | $\mathbf{0.1596} \pm 0.0191$ | $\mathbf{0.0563} \pm 0.0074$ | $\mathbf{0.0188} \pm 0.0021$ |

| task | Sigmoid | | | Gaussian | | |
|------|---------|------|------|----------|------|------|
| $m$ | 5 | 10 | 20 | 5 | 10 | 20 |
| MAML | $0.1999 \pm 0.0296$ | $0.0396 \pm 0.0056$ | $0.0148 \pm 0.0029$ | $0.1995 \pm 0.0175$ | $0.0862 \pm 0.0061$ | $0.0451 \pm 0.0060$ |
| Reptile | $0.2431 \pm 0.0281$ | $0.0474 \pm 0.0072$ | $0.0177 \pm 0.0039$ | $0.3176 \pm 0.0261$ | $0.1239 \pm 0.0147$ | $0.0694 \pm 0.0124$ |
| MOSM | $0.6438 \pm 0.6566$ | $0.6054 \pm 0.6225$ | $0.6009 \pm 0.6179$ | $0.7974 \pm 0.6765$ | $0.7498 \pm 0.6541$ | $0.7201 \pm 0.6529$ |
| CSM | $0.6620 \pm 0.4619$ | $0.6454 \pm 0.4349$ | $0.7356 \pm 0.3281$ | $0.9119 \pm 0.4539$ | $0.8470 \pm 0.4587$ | $0.9381 \pm 0.3498$ |
| STNP | $0.0482 \pm 0.0014$ | $0.0270 \pm 0.0024$ | $0.0088 \pm 0.0019$ | $0.0745 \pm 0.0009$ | $0.0589 \pm 0.0024$ | $0.0449 \pm 0.0031$ |
| S+JTNP | $0.0412 \pm 0.0025$ | $0.0235 \pm 0.0055$ | $0.0089 \pm 0.0033$ | $0.1136 \pm 0.0099$ | $0.0669 \pm 0.0044$ | $0.0367 \pm 0.0035$ |
| MTNP | $\mathbf{0.0302} \pm 0.0022$ | $\mathbf{0.0092} \pm 0.0010$ | $\mathbf{0.0028} \pm 0.0004$ | $\mathbf{0.0668} \pm 0.0041$ | $\mathbf{0.0367} \pm 0.0018$ | $\mathbf{0.0191} \pm 0.0003$ |

# E  ADDITIONAL RESULTS ON WEATHER TIME-SERIES REGRESSION

In this section, we provide additional results on the time-series regression experiment on weather data, with various missing rates $\gamma$ and also with standard deviation from 5 different random seeds. In this section, we provide additional results on the synthetic experiment, with various missing rates $\gamma$ and also with standard deviation from 5 different random seeds. When the data is highly incomplete (i.e., $\gamma = 0.5, 0.75$), we can see that MTNP clearly outperforms the baselines in general. When the complete or less incomplete data ($\gamma = 0, 0.25$) is given, MTNP still outperforms the gradient-based meta-learning baselines and MOSM while achieves at least competitive performance to CSM, STNP and JTNP.

Table 14: Average MSE and NLL on weather tasks, with $m = 10$ and $\gamma = 0$.

| task | TempMin | | TempMax | | Humidity | |
|------|---------|-----|---------|-----|----------|-----|
| metric | MSE | NLL | MSE | NLL | MSE | NLL |
| MAML | $0.0044 \pm 0.0003$ | - | $0.0059 \pm 0.0004$ | - | $0.0517 \pm 0.0020$ | - |
| Reptile | $0.0038 \pm 0.0002$ | - | $0.0053 \pm 0.0003$ | - | $0.0510 \pm 0.0023$ | - |
| MOSM | $0.0056 \pm 0.0005$ | $-1.0074 \pm 0.1020$ | $0.0106 \pm 0.0021$ | $-0.4788 \pm 0.1607$ | $0.0735 \pm 0.0096$ | $0.7877 \pm 0.1929$ |
| CSM | $0.0056 \pm 0.0019$ | $\mathbf{-1.4274} \pm 0.0863$ | $0.0064 \pm 0.0009$ | $\mathbf{-1.1762} \pm 0.0755$ | $0.0572 \pm 0.0052$ | $0.0482 \pm 0.1332$ |
| STNP | $0.0034 \pm 0.0002$ | $-1.2159 \pm 0.0113$ | $0.0051 \pm 0.0002$ | $-1.1340 \pm 0.0098$ | $0.0487 \pm 0.0013$ | $-0.0937 \pm 0.0760$ |
| JTNP | $0.0034 \pm 0.0002$ | $-1.2128 \pm 0.0076$ | $0.0050 \pm 0.0001$ | $-1.1310 \pm 0.0088$ | $\mathbf{0.0461} \pm 0.0021$ | $\mathbf{-0.2421} \pm 0.0455$ |
| MTNP | $\mathbf{0.0032} \pm 0.0002$ | $-1.2169 \pm 0.0123$ | $\mathbf{0.0048} \pm 0.0002$ | $-1.1389 \pm 0.0067$ | $0.0487 \pm 0.0015$ | $-0.1999 \pm 0.0376$ |

| task | Precip | | Cloud | | Dew | |
|------|--------|-----|-------|-----|-----|-----|
| metric | MSE | NLL | MSE | NLL | MSE | NLL |
| MAML | $0.2576 \pm 0.0068$ | - | $0.2576 \pm 0.0086$ | - | $0.0089 \pm 0.0005$ | - |
| Reptile | $0.2599 \pm 0.0077$ | - | $0.2618 \pm 0.0109$ | - | $0.0080 \pm 0.0004$ | - |
| MOSM | $0.2957 \pm 0.0131$ | $4.1621 \pm 1.0855$ | $0.2997 \pm 0.0203$ | $1.5618 \pm 0.1822$ | $0.0113 \pm 0.0017$ | $-0.3663 \pm 0.1571$ |
| CSM | $0.2708 \pm 0.0130$ | $2.6255 \pm 0.7511$ | $0.2664 \pm 0.0151$ | $0.9500 \pm 0.0802$ | $0.0098 \pm 0.0022$ | $-1.0495 \pm 0.1330$ |
| STNP | $0.2411 \pm 0.0041$ | $2.1338 \pm 0.2856$ | $0.2415 \pm 0.0062$ | $0.8888 \pm 0.0356$ | $0.0071 \pm 0.0004$ | $-1.0475 \pm 0.0210$ |
| JTNP | $\mathbf{0.2166} \pm 0.0031$ | $\mathbf{0.6014} \pm 0.0200$ | $0.2202 \pm 0.0023$ | $\mathbf{0.6446} \pm 0.0469$ | $0.0069 \pm 0.0006$ | $-1.0568 \pm 0.0230$ |
| MTNP | $0.2194 \pm 0.0019$ | $0.6642 \pm 0.0331$ | $\mathbf{0.2144} \pm 0.0066$ | $0.6451 \pm 0.0240$ | $\mathbf{0.0068} \pm 0.0004$ | $\mathbf{-1.0672} \pm 0.0142$ |

Table 15: Average MSE and NLL on weather tasks, with $m = 10$ and $\gamma = 0.25$.

| task | TempMin | | TempMax | | Humidity | |
|------|---------|-----|---------|-----|----------|-----|
| metric | MSE | NLL | MSE | NLL | MSE | NLL |
| MAML | $0.0052 \pm 0.0006$ | - | $0.0070 \pm 0.0005$ | - | $0.0558 \pm 0.0044$ | - |
| Reptile | $0.0042 \pm 0.0002$ | - | $0.0059 \pm 0.0003$ | - | $0.0555 \pm 0.0050$ | - |
| MOSM | $0.0056 \pm 0.0009$ | $-0.9333 \pm 0.1774$ | $0.0106 \pm 0.0020$ | $-0.1676 \pm 0.2981$ | $0.0719 \pm 0.0073$ | $0.8968 \pm 0.2437$ |
| CSM | $0.0057 \pm 0.0010$ | $\mathbf{-1.3140} \pm 0.0921$ | $0.0087 \pm 0.0027$ | $-0.9624 \pm 0.0974$ | $0.0589 \pm 0.0032$ | $0.0951 \pm 0.1031$ |
| STNP | $0.0036 \pm 0.0001$ | $-1.1995 \pm 0.0055$ | $0.0055 \pm 0.0002$ | $-1.1124 \pm 0.0116$ | $0.0516 \pm 0.0031$ | $-0.0911 \pm 0.0907$ |
| S+JTNP | $0.0038 \pm 0.0002$ | $-1.1976 \pm 0.0054$ | $0.0056 \pm 0.0004$ | $-1.1118 \pm 0.0133$ | $\mathbf{0.0491} \pm 0.0017$ | $\mathbf{-0.1976} \pm 0.0593$ |
| MTNP | $\mathbf{0.0033} \pm 0.0002$ | $-1.2103 \pm 0.0094$ | $\mathbf{0.0051} \pm 0.0003$ | $\mathbf{-1.1252} \pm 0.0143$ | $0.0504 \pm 0.0021$ | $-0.1766 \pm 0.0523$ |

| task | Precip | | Cloud | | Dew | |
|------|--------|-----|-------|-----|-----|-----|
| metric | MSE | NLL | MSE | NLL | MSE | NLL |
| MAML | $0.2759 \pm 0.0058$ | - | $0.2713 \pm 0.0148$ | - | $0.0097 \pm 0.0012$ | - |
| Reptile | $0.2801 \pm 0.0092$ | - | $0.2755 \pm 0.0154$ | - | $0.0087 \pm 0.0010$ | - |
| MOSM | $0.3058 \pm 0.0252$ | $4.8069 \pm 0.9710$ | $0.3034 \pm 0.0102$ | $1.6875 \pm 0.2039$ | $0.0108 \pm 0.0017$ | $-0.2865 \pm 0.0982$ |
| CSM | $0.2737 \pm 0.0171$ | $3.3485 \pm 0.8053$ | $0.2842 \pm 0.0239$ | $1.0663 \pm 0.0729$ | $0.0105 \pm 0.0016$ | $-0.9097 \pm 0.1864$ |
| STNP | $0.2459 \pm 0.0077$ | $1.3797 \pm 0.3219$ | $0.2436 \pm 0.0127$ | $0.8129 \pm 0.0471$ | $0.0076 \pm 0.0006$ | $-1.0168 \pm 0.0316$ |
| S+JTNP | $0.2221 \pm 0.0047$ | $\mathbf{0.6164} \pm 0.0352$ | $0.2289 \pm 0.0050$ | $0.6632 \pm 0.0427$ | $0.0074 \pm 0.0004$ | $-1.0430 \pm 0.0272$ |
| MTNP | $\mathbf{0.2216} \pm 0.0037$ | $0.6729 \pm 0.0574$ | $\mathbf{0.2177} \pm 0.0058$ | $\mathbf{0.6461} \pm 0.0195$ | $\mathbf{0.0072} \pm 0.0005$ | $\mathbf{-1.0563} \pm 0.0193$ |

Table 16: Average MSE and NLL on weather tasks, with $m = 10$ and $\gamma = 0.5$.

| task | TempMin | | TempMax | | Humidity | |
|------|---------|---|---------|---|----------|---|
| metric | MSE | NLL | MSE | NLL | MSE | NLL |
| MAML | $0.0067 \pm 0.0009$ | - | $0.0094 \pm 0.0017$ | - | $0.0705 \pm 0.0083$ | - |
| Reptile | $0.0060 \pm 0.0007$ | - | $0.0078 \pm 0.0004$ | - | $0.0691 \pm 0.0092$ | - |
| MOSM | $0.0091 \pm 0.0015$ | $-0.0194 \pm 0.6391$ | $0.0124 \pm 0.0022$ | $-0.0259 \pm 0.2832$ | $0.0827 \pm 0.0093$ | $1.3831 \pm 0.2993$ |
| CSM | $0.0069 \pm 0.0015$ | $-0.8839 \pm 0.2680$ | $0.0123 \pm 0.0053$ | $-0.8522 \pm 0.1301$ | $0.0906 \pm 0.0288$ | $0.6640 \pm 0.5221$ |
| STNP | $0.0046 \pm 0.0004$ | $-1.1514 \pm 0.0181$ | $0.0069 \pm 0.0004$ | $-1.0390 \pm 0.0106$ | $0.0632 \pm 0.0072$ | $0.1273 \pm 0.1898$ |
| S+JTNP | $0.0045 \pm 0.0006$ | $-1.1703 \pm 0.0144$ | $0.0068 \pm 0.0003$ | $-1.0681 \pm 0.0112$ | $0.0607 \pm 0.0053$ | $0.0169 \pm 0.1437$ |
| MTNP | $\mathbf{0.0037} \pm 0.0001$ | $\mathbf{-1.1832} \pm 0.0165$ | $\mathbf{0.0054} \pm 0.0001$ | $\mathbf{-1.1049} \pm 0.0154$ | $\mathbf{0.0546} \pm 0.0021$ | $\mathbf{-0.1006} \pm 0.0696$ |

| task | Precip | | Cloud | | Dew | |
|------|--------|---|-------|---|-----|---|
| metric | MSE | NLL | MSE | NLL | MSE | NLL |
| MAML | $0.3041 \pm 0.0049$ | - | $0.2987 \pm 0.0132$ | - | $0.0106 \pm 0.0010$ | - |
| Reptile | $0.3160 \pm 0.0087$ | - | $0.3047 \pm 0.0152$ | - | $0.0096 \pm 0.0008$ | - |
| MOSM | $0.3021 \pm 0.0161$ | $4.1009 \pm 0.5731$ | $0.3170 \pm 0.0152$ | $2.0663 \pm 0.3535$ | $0.0128 \pm 0.0023$ | $-0.0255 \pm 0.2011$ |
| CSM | $0.2895 \pm 0.0103$ | $3.1897 \pm 0.7841$ | $0.2983 \pm 0.0130$ | $1.2655 \pm 0.2538$ | $0.0118 \pm 0.0016$ | $-0.7243 \pm 0.2799$ |
| STNP | $0.2607 \pm 0.0082$ | $1.1242 \pm 0.2362$ | $0.2631 \pm 0.0044$ | $0.8563 \pm 0.0637$ | $0.0086 \pm 0.0008$ | $-0.9815 \pm 0.0283$ |
| S+JTNP | $0.2348 \pm 0.0038$ | $0.6792 \pm 0.0314$ | $0.2376 \pm 0.0041$ | $0.6812 \pm 0.0231$ | $0.0084 \pm 0.0007$ | $-0.9946 \pm 0.0161$ |
| MTNP | $\mathbf{0.2276} \pm 0.0028$ | $\mathbf{0.6557} \pm 0.0433$ | $\mathbf{0.2215} \pm 0.0043$ | $\mathbf{0.6660} \pm 0.0141$ | $\mathbf{0.0073} \pm 0.0003$ | $\mathbf{-1.0331} \pm 0.0147$ |

Table 17: Average MSE and NLL on weather tasks, with $m = 10$ and $\gamma = 0.75$.

| task | TempMin | | TempMax | | Humidity | |
|------|---------|---|---------|---|----------|---|
| metric | MSE | NLL | MSE | NLL | MSE | NLL |
| MAML | $0.0094 \pm 0.0018$ | - | $0.0141 \pm 0.0031$ | - | $0.0830 \pm 0.0083$ | - |
| Reptile | $0.0083 \pm 0.0014$ | - | $0.0122 \pm 0.0032$ | - | $0.0851 \pm 0.0057$ | - |
| MOSM | $0.0137 \pm 0.0028$ | $0.4165 \pm 0.7186$ | $0.0212 \pm 0.0055$ | $0.5358 \pm 0.3826$ | $0.0890 \pm 0.0126$ | $1.3986 \pm 0.2961$ |
| CSM | $0.0106 \pm 0.0011$ | $-0.3791 \pm 0.5254$ | $0.0174 \pm 0.0064$ | $-0.0270 \pm 0.2872$ | $0.0908 \pm 0.0185$ | $0.9146 \pm 0.1250$ |
| STNP | $0.0072 \pm 0.0013$ | $-1.0291 \pm 0.0498$ | $0.0104 \pm 0.0029$ | $-0.8627 \pm 0.1556$ | $0.0776 \pm 0.0094$ | $0.3167 \pm 0.1057$ |
| S+JTNP | $0.0073 \pm 0.0017$ | $-1.0810 \pm 0.0558$ | $0.0103 \pm 0.0023$ | $-0.9165 \pm 0.0893$ | $0.0803 \pm 0.0095$ | $0.3243 \pm 0.1008$ |
| MTNP | $\mathbf{0.0047} \pm 0.0004$ | $\mathbf{-1.1272} \pm 0.0166$ | $\mathbf{0.0075} \pm 0.0010$ | $\mathbf{-1.0004} \pm 0.0566$ | $\mathbf{0.0644} \pm 0.0040$ | $\mathbf{0.0454} \pm 0.0473$ |

| task | Precip | | Cloud | | Dew | |
|------|--------|---|-------|---|-----|---|
| metric | MSE | NLL | MSE | NLL | MSE | NLL |
| MAML | $0.3903 \pm 0.0621$ | - | $0.3482 \pm 0.0154$ | - | $0.0147 \pm 0.0026$ | - |
| Reptile | $0.4059 \pm 0.0720$ | - | $0.3662 \pm 0.0189$ | - | $0.0143 \pm 0.0022$ | - |
| MOSM | $0.3297 \pm 0.0333$ | $5.5436 \pm 0.5968$ | $0.3256 \pm 0.0216$ | $2.0465 \pm 0.5114$ | $0.0182 \pm 0.0019$ | $0.4010 \pm 0.2871$ |
| CSM | $0.3095 \pm 0.0376$ | $5.1712 \pm 0.7329$ | $0.3144 \pm 0.0171$ | $1.2520 \pm 0.1105$ | $0.0145 \pm 0.0039$ | $-0.4852 \pm 0.2178$ |
| STNP | $0.2877 \pm 0.0082$ | $1.3884 \pm 0.2007$ | $0.2884 \pm 0.0117$ | $0.8719 \pm 0.0649$ | $0.0122 \pm 0.0007$ | $-0.8116 \pm 0.0539$ |
| S+JTNP | $0.2702 \pm 0.0145$ | $0.8052 \pm 0.0799$ | $0.2587 \pm 0.0131$ | $0.7621 \pm 0.0380$ | $0.0106 \pm 0.0011$ | $-0.9028 \pm 0.0447$ |
| MTNP | $\mathbf{0.2433} \pm 0.0105$ | $\mathbf{0.7368} \pm 0.1119$ | $\mathbf{0.2339} \pm 0.0053$ | $\mathbf{0.7009} \pm 0.0316$ | $\mathbf{0.0090} \pm 0.0009$ | $\mathbf{-0.9388} \pm 0.0656$ |

Table 18: Average MSE and NLL on weather tasks, with $m = 20$ and $\gamma = 0$.

| task | TempMin | | TempMax | | Humidity | |
|---|---|---|---|---|---|---|
| metric | MSE | NLL | MSE | NLL | MSE | NLL |
| MAML | $0.0040 \pm 0.0002$ | - | $0.0055 \pm 0.0003$ | - | $0.0449 \pm 0.0017$ | - |
| Reptile | $0.0033 \pm 0.0001$ | - | $0.0047 \pm 0.0002$ | - | $0.0438 \pm 0.0014$ | - |
| MOSM | $0.0035 \pm 0.0002$ | $-1.4556 \pm 0.0625$ | $0.0057 \pm 0.0006$ | $-1.0477 \pm 0.0962$ | $0.0494 \pm 0.0013$ | $0.3296 \pm 0.0594$ |
| CSM | $0.0033 \pm 0.0002$ | $\mathbf{-1.6792} \pm 0.0131$ | $0.0049 \pm 0.0004$ | $\mathbf{-1.4526} \pm 0.0338$ | $0.0458 \pm 0.0032$ | $-0.2356 \pm 0.0377$ |
| STNP | $0.0030 \pm 0.0001$ | $-1.2349 \pm 0.0038$ | $0.0045 \pm 0.0000$ | $-1.1615 \pm 0.0022$ | $\mathbf{0.0418} \pm 0.0013$ | $-0.2829 \pm 0.0509$ |
| JTNP | $0.0031 \pm 0.0001$ | $-1.2255 \pm 0.0060$ | $0.0045 \pm 0.0001$ | $-1.1564 \pm 0.0060$ | $0.0431 \pm 0.0007$ | $\mathbf{-0.3034} \pm 0.0238$ |
| MTNP | $\mathbf{0.0029} \pm 0.0001$ | $-1.2309 \pm 0.0039$ | $\mathbf{0.0042} \pm 0.0001$ | $-1.1636 \pm 0.0047$ | $0.0452 \pm 0.0014$ | $-0.2809 \pm 0.0205$ |

| task | Precip | | Cloud | | Dew | |
|---|---|---|---|---|---|---|
| metric | MSE | NLL | MSE | NLL | MSE | NLL |
| MAML | $0.2316 \pm 0.0043$ | - | $0.2231 \pm 0.0013$ | - | $0.0080 \pm 0.0001$ | - |
| Reptile | $0.2337 \pm 0.0055$ | - | $0.2252 \pm 0.0021$ | - | $0.0068 \pm 0.0001$ | - |
| MOSM | $0.2638 \pm 0.0073$ | $2.6499 \pm 0.4487$ | $0.2520 \pm 0.0065$ | $1.0786 \pm 0.0817$ | $0.0073 \pm 0.0004$ | $-0.9203 \pm 0.0998$ |
| CSM | $0.2482 \pm 0.0057$ | $1.2779 \pm 0.3799$ | $0.2434 \pm 0.0161$ | $0.7473 \pm 0.0361$ | $0.0072 \pm 0.0010$ | $\mathbf{-1.2782} \pm 0.0584$ |
| STNP | $0.2195 \pm 0.0055$ | $1.0330 \pm 0.1193$ | $0.2156 \pm 0.0054$ | $0.6765 \pm 0.0339$ | $\mathbf{0.0062} \pm 0.0002$ | $-1.0897 \pm 0.0101$ |
| JTNP | $\mathbf{0.2116} \pm 0.0020$ | $\mathbf{0.5441} \pm 0.0298$ | $0.2133 \pm 0.0024$ | $\mathbf{0.5956} \pm 0.0106$ | $0.0064 \pm 0.0001$ | $-1.0772 \pm 0.0078$ |
| MTNP | $0.2134 \pm 0.0018$ | $0.5728 \pm 0.0286$ | $\mathbf{0.2084} \pm 0.0052$ | $0.5992 \pm 0.0079$ | $0.0065 \pm 0.0003$ | $-1.0939 \pm 0.0064$ |

Table 19: Average MSE and NLL on weather tasks, with $m = 20$ and $\gamma = 0.25$.

| task | TempMin | | TempMax | | Humidity | |
|---|---|---|---|---|---|---|
| metric | MSE | NLL | MSE | NLL | MSE | NLL |
| MAML | $0.0043 \pm 0.0002$ | - | $0.0057 \pm 0.0003$ | - | $0.0467 \pm 0.0014$ | - |
| Reptile | $0.0037 \pm 0.0002$ | - | $0.0049 \pm 0.0002$ | - | $0.0457 \pm 0.0011$ | - |
| MOSM | $0.0036 \pm 0.0003$ | $-1.3658 \pm 0.1010$ | $0.0064 \pm 0.0016$ | $-0.9779 \pm 0.1294$ | $0.0509 \pm 0.0014$ | $0.4595 \pm 0.1306$ |
| CSM | $0.0044 \pm 0.0017$ | $\mathbf{-1.6228} \pm 0.0354$ | $0.0057 \pm 0.0007$ | $\mathbf{-1.3557} \pm 0.0260$ | $0.0471 \pm 0.0020$ | $-0.1404 \pm 0.1129$ |
| STNP | $0.0032 \pm 0.0001$ | $-1.2247 \pm 0.0050$ | $0.0047 \pm 0.0001$ | $-1.1530 \pm 0.0090$ | $0.0439 \pm 0.0017$ | $-0.2663 \pm 0.0519$ |
| S+JTNP | $0.0033 \pm 0.0001$ | $-1.2175 \pm 0.0064$ | $0.0047 \pm 0.0001$ | $-1.1488 \pm 0.0046$ | $\mathbf{0.0450} \pm 0.0005$ | $\mathbf{-0.2727} \pm 0.0277$ |
| MTNP | $\mathbf{0.0030} \pm 0.0001$ | $-1.2257 \pm 0.0036$ | $\mathbf{0.0043} \pm 0.0002$ | $-1.1591 \pm 0.0057$ | $0.0465 \pm 0.0013$ | $-0.2713 \pm 0.0175$ |

| task | Precip | | Cloud | | Dew | |
|---|---|---|---|---|---|---|
| metric | MSE | NLL | MSE | NLL | MSE | NLL |
| MAML | $0.2379 \pm 0.0079$ | - | $0.2287 \pm 0.0018$ | - | $0.0084 \pm 0.0003$ | - |
| Reptile | $0.2397 \pm 0.0098$ | - | $0.2306 \pm 0.0019$ | - | $0.0071 \pm 0.0002$ | - |
| MOSM | $0.2673 \pm 0.0110$ | $2.6371 \pm 0.6555$ | $0.2691 \pm 0.0091$ | $1.3464 \pm 0.0967$ | $0.0075 \pm 0.0008$ | $-0.8713 \pm 0.0673$ |
| CSM | $0.2554 \pm 0.0034$ | $1.4961 \pm 0.3189$ | $0.2464 \pm 0.0096$ | $0.8126 \pm 0.0324$ | $0.0076 \pm 0.0009$ | $\mathbf{-1.2384} \pm 0.0567$ |
| STNP | $0.2208 \pm 0.0032$ | $0.6287 \pm 0.0876$ | $0.2193 \pm 0.0036$ | $0.6448 \pm 0.0227$ | $\mathbf{0.0064} \pm 0.0002$ | $-1.0780 \pm 0.0129$ |
| S+JTNP | $0.2128 \pm 0.0028$ | $0.5581 \pm 0.0376$ | $0.2167 \pm 0.0032$ | $0.6086 \pm 0.0134$ | $0.0067 \pm 0.0001$ | $-1.0695 \pm 0.0095$ |
| MTNP | $\mathbf{0.2147} \pm 0.0028$ | $\mathbf{0.5722} \pm 0.0251$ | $\mathbf{0.2098} \pm 0.0047$ | $\mathbf{0.5980} \pm 0.0125$ | $0.0067 \pm 0.0003$ | $-1.0874 \pm 0.0053$ |

Table 20: Average MSE and NLL on weather tasks, with $m = 20$ and $\gamma = 0.5$.

| task | TempMin | | TempMax | | Humidity | |
|---|---|---|---|---|---|---|
| metric | MSE | NLL | MSE | NLL | MSE | NLL |
| MAML | $0.0046 \pm 0.0002$ | - | $0.0058 \pm 0.0003$ | - | $0.0505 \pm 0.0028$ | - |
| Reptile | $0.0041 \pm 0.0003$ | - | $0.0053 \pm 0.0001$ | - | $0.0495 \pm 0.0025$ | - |
| MOSM | $0.0048 \pm 0.0009$ | $-1.0919 \pm 0.1379$ | $0.0083 \pm 0.0012$ | $-0.4804 \pm 0.1996$ | $0.0600 \pm 0.0035$ | $0.7389 \pm 0.2151$ |
| CSM | $0.0050 \pm 0.0010$ | $\mathbf{-1.4484} \pm 0.0377$ | $0.0069 \pm 0.0012$ | $-1.1374 \pm 0.1509$ | $0.0584 \pm 0.0087$ | $0.0686 \pm 0.1651$ |
| STNP | $0.0034 \pm 0.0002$ | $-1.2122 \pm 0.0098$ | $0.0051 \pm 0.0002$ | $-1.1368 \pm 0.0092$ | $0.0477 \pm 0.0020$ | $-0.2018 \pm 0.0608$ |
| S+JTNP | $0.0037 \pm 0.0001$ | $-1.2060 \pm 0.0081$ | $0.0051 \pm 0.0003$ | $-1.1320 \pm 0.0017$ | $0.0480 \pm 0.0019$ | $-0.2149 \pm 0.0316$ |
| MTNP | $\mathbf{0.0031} \pm 0.0002$ | $-1.2210 \pm 0.0054$ | $\mathbf{0.0045} \pm 0.0001$ | $\mathbf{-1.1538} \pm 0.0058$ | $\mathbf{0.0475} \pm 0.0012$ | $\mathbf{-0.2483} \pm 0.0232$ |

| task | Precip | | Cloud | | Dew | |
|---|---|---|---|---|---|---|
| metric | MSE | NLL | MSE | NLL | MSE | NLL |
| MAML | $0.2585 \pm 0.0080$ | - | $0.2450 \pm 0.0073$ | - | $0.0089 \pm 0.0009$ | - |
| Reptile | $0.2609 \pm 0.0072$ | - | $0.2491 \pm 0.0115$ | - | $0.0077 \pm 0.0004$ | - |
| MOSM | $0.2904 \pm 0.0177$ | $3.5379 \pm 0.7060$ | $0.2883 \pm 0.0166$ | $1.5341 \pm 0.2511$ | $0.0089 \pm 0.0010$ | $-0.5127 \pm 0.1146$ |
| CSM | $0.2699 \pm 0.0058$ | $2.1170 \pm 0.4957$ | $0.2689 \pm 0.0152$ | $0.9841 \pm 0.1305$ | $0.0101 \pm 0.0014$ | $-0.9491 \pm 0.1008$ |
| STNP | $0.2387 \pm 0.0086$ | $0.7083 \pm 0.1090$ | $0.2371 \pm 0.0081$ | $0.7121 \pm 0.0782$ | $0.0070 \pm 0.0003$ | $-1.0455 \pm 0.0112$ |
| S+JTNP | $0.2211 \pm 0.0022$ | $\mathbf{0.5761} \pm 0.0341$ | $0.2224 \pm 0.0082$ | $0.6393 \pm 0.0206$ | $0.0071 \pm 0.0003$ | $-1.0577 \pm 0.0064$ |
| MTNP | $\mathbf{0.2169} \pm 0.0022$ | $0.5811 \pm 0.0283$ | $\mathbf{0.2129} \pm 0.0039$ | $\mathbf{0.6175} \pm 0.0212$ | $\mathbf{0.0066} \pm 0.0003$ | $\mathbf{-1.0834} \pm 0.0080$ |

Table 21: Average MSE and NLL on weather tasks, with $m = 20$ and $\gamma = 0.75$.

| task | TempMin | | TempMax | | Humidity | |
|---|---|---|---|---|---|---|
| metric | MSE | NLL | MSE | NLL | MSE | NLL |
| MAML | $0.0060 \pm 0.0009$ | - | $0.0087 \pm 0.0012$ | - | $0.0623 \pm 0.0051$ | - |
| Reptile | $0.0056 \pm 0.0008$ | - | $0.0082 \pm 0.0016$ | - | $0.0615 \pm 0.0055$ | - |
| MOSM | $0.0077 \pm 0.0014$ | $-0.1015 \pm 0.3970$ | $0.0107 \pm 0.0030$ | $0.1338 \pm 0.1958$ | $0.0727 \pm 0.0056$ | $1.3645 \pm 0.1313$ |
| CSM | $0.0068 \pm 0.0020$ | $-1.0450 \pm 0.2197$ | $0.0098 \pm 0.0027$ | $-0.7641 \pm 0.0976$ | $0.0663 \pm 0.0062$ | $0.3055 \pm 0.1080$ |
| STNP | $0.0043 \pm 0.0005$ | $-1.1555 \pm 0.0317$ | $0.0070 \pm 0.0010$ | $-1.0324 \pm 0.0627$ | $0.0576 \pm 0.0070$ | $-0.0438 \pm 0.0543$ |
| S+JTNP | $0.0051 \pm 0.0007$ | $-1.1630 \pm 0.0350$ | $0.0069 \pm 0.0005$ | $-1.0567 \pm 0.0248$ | $0.0598 \pm 0.0041$ | $-0.0294 \pm 0.0833$ |
| MTNP | $\mathbf{0.0036} \pm 0.0002$ | $\mathbf{-1.1960} \pm 0.0115$ | $\mathbf{0.0053} \pm 0.0006$ | $\mathbf{-1.1052} \pm 0.0184$ | $\mathbf{0.0533} \pm 0.0047$ | $\mathbf{-0.1471} \pm 0.0571$ |

| task | Precip | | Cloud | | Dew | |
|---|---|---|---|---|---|---|
| metric | MSE | NLL | MSE | NLL | MSE | NLL |
| MAML | $0.3091 \pm 0.0224$ | - | $0.2715 \pm 0.0071$ | - | $0.0108 \pm 0.0010$ | - |
| Reptile | $0.3231 \pm 0.0310$ | - | $0.2816 \pm 0.0075$ | - | $0.0105 \pm 0.0014$ | - |
| MOSM | $0.2947 \pm 0.0173$ | $5.7383 \pm 0.5086$ | $0.3100 \pm 0.0171$ | $1.8804 \pm 0.3589$ | $0.0122 \pm 0.0021$ | $0.2336 \pm 0.3153$ |
| CSM | $0.2833 \pm 0.0070$ | $4.3286 \pm 1.1360$ | $0.2896 \pm 0.0138$ | $1.2055 \pm 0.2523$ | $0.0116 \pm 0.0028$ | $-0.5992 \pm 0.3012$ |
| STNP | $0.2602 \pm 0.0080$ | $0.9782 \pm 0.2730$ | $0.2590 \pm 0.0146$ | $0.7367 \pm 0.0436$ | $0.0093 \pm 0.0007$ | $-0.9411 \pm 0.0460$ |
| S+JTNP | $0.2377 \pm 0.0100$ | $0.6721 \pm 0.0671$ | $0.2354 \pm 0.0052$ | $0.6888 \pm 0.0249$ | $0.0086 \pm 0.0004$ | $-1.0001 \pm 0.0183$ |
| MTNP | $\mathbf{0.2248} \pm 0.0081$ | $\mathbf{0.6188} \pm 0.0566$ | $\mathbf{0.2193} \pm 0.0041$ | $\mathbf{0.6403} \pm 0.0262$ | $\mathbf{0.0078} \pm 0.0008$ | $\mathbf{-1.0210} \pm 0.0187$ |

# F    EXPERIMENTAL DETAILS OF IMAGE 2D FUNCTION REGRESSION

In this section, we describe details of the data generating process and experimental settings of the image function regression experiment.

## F.1    DATASET

We use 30,000 RGB images from CelebA HQ dataset (Liu et al., 2015) for RGB task and the corresponding semantic segmentation masks among 19 semantic classes from CelebA Mask-HQ dataset (Lee et al., 2020) for Segment task. For Edge task, we apply the Sobel filter (Kanopoulos et al., 1988) on the RGB images to generate continuous-valued edges. This corresponds to the Canny edge (Canny, 1986) without non-maximum suppression, which is also used in Zamir et. al. (2018) (Zamir et al., 2018). For PNCC task, we apply a pretrained 3D face reconstruction model on the RGB images to generate PNCC label maps (Guo et al., 2020a). At each pixel, the PNCC label consists of the $(x, y, z)$ coordinate of the facial keypoint located at the pixel. In summary, labels for RGB are 3-dimensional, Edge are 1-dimensional, Segment are 19-dimensional, and PNCC are 3-dimensional vectors. We split the 30,000 images into 27,000 train, 1,500 valid, and 1,500 test images.

## F.2    EVALUATION PROTOCOL

To evaluate the accuracy of the multi-task prediction, we average MSE $MSE = \frac{1}{n} \sum_{i=1}^{n} (y_i^t - \hat{y}_i^t)^2$ on the test images for continuous tasks (RGB, Edge, PNCC), and average mean IoU on the test images for discrete task (Segment). The predictive posterior mean is computed by Monte Carlo sampling, the same as in the 1D experiment. For categorical outputs, we discretize the prediction with the argmax operator $\hat{y}_i^t = \text{argmax}_k \, p(y_i^t = k | x_i, v^t)$.

To evaluate the consistency of predictions across tasks (coherency), we translate each RGB prediction to other task labels. For Edge and PNCC, we use the ground-truth label generation algorithm and the pretrained model used to generate the ground-truth labels for the translation, respectively. For Segment, we fine-tuned DeeplabV3+ (Chen et al., 2018) with ImageNet (Krizhevsky et al., 2012) pretrained ResNet-50 (He et al., 2016) backbone. We refer to the github repository of Yakubovskiy (2019) (Yakubovskiy, 2020) for the DeeplabV3+ model. After the translation, we measure MSE and 1 - mIoU for continuous and discrete tasks respectively, to evaluate the *disagreement* (as oppose to the coherency) between the predictions.

To examine the learned correlation across tasks by MTNP (Table 3 in the main paper), we compare the performance before and after MTNP observes a set of source data. The source data consist of all examples labeled with the source tasks. For example, if the target task is RGB and the source tasks are Edge and Segment, we give Edge and Segment labels for all pixels, while no RGB or PNCC labels are given. Since MTNP requires at least one labeled example for each task, we give a single completely labeled example which is chosen randomly to MTNP as a base context, before MTNP observes the source data. There are total $\binom{4}{1} + \binom{4}{2} + \binom{4}{3} = 14$ different combinations of source tasks exist. By excluding the case where target task is in the set of source tasks, total 7 different combinations of source tasks remain for each target task. To measure the performance gain from task $f^1$ to $f^2$, we average the performance gain of $f^2$ from all sets of source tasks that containing $f^1$. For example, the performance gain from Edge to RGB is computed by averaging performance gains $\delta_{\text{Edge}\to\text{RGB}}$, $\delta_{\text{Edge,Segment}\to\text{RGB}}$, $\delta_{\text{Edge,PNCC}\to\text{RGB}}$, and $\delta_{\text{Edge,Segment,PNCC}\to\text{RGB}}$, where we denote $\delta_{A\to B}$ by the performance gain from source tasks $A$ to target task $B$.

# G   ADDITIONAL RESULTS ON IMAGE 2D FUNCTION REGRESSION

In this section, we provide additional results on the image regression experiment, with various missing rates $\gamma$ and also with standard deviation from 5 different random seeds.

Table 26: Relative performance gain with 5 different random seeds (%).

| Source \ Target | RGB | Edge | Segment | PNCC |
|---|---|---|---|---|
| RGB | - | **53.02 ± 4.71** | 8.73 ± 6.50 | 18.57 ± 12.94 |
| Edge | **6.35 ± 1.95** | - | 8.18 ± 6.63 | 15.70 ± 13.11 |
| Segment | 5.13 ± 2.04 | 33.30 ± 23.94 | - | **29.24 ± 2.62** |
| PNCC | 5.58 ± 2.53 | 31.88 ± 23.64 | **15.88 ± 2.21** | - |

Table 22: Average reconstruction errors (performance) and disagreement of predictions (coherency) on 2D function regression, with varying context size ($m$) and $\gamma = 0$.

| task | RGB (performance) | | |
|------|------|------|------|
| $m$ | 10 | 100 | 512 |
| STNP | $0.0344 \pm 0.0002$ | $0.0101 \pm 0.0000$ | $0.0034 \pm 0.0000$ |
| JTNP | $0.0304 \pm 0.0002$ | $0.0073 \pm 0.0000$ | $0.0021 \pm 0.0000$ |
| MTNP | $\mathbf{0.0295 \pm 0.0001}$ | $\mathbf{0.0066 \pm 0.0000}$ | $\mathbf{0.0015 \pm 0.0000}$ |

| task | Edge (performance) | | | Edge (coherency) | | |
|------|------|------|------|------|------|------|
| $m$ | 10 | 100 | 512 | 10 | 100 | 512 |
| STNP | $0.0342 \pm 0.0001$ | $0.0198 \pm 0.0000$ | $0.0066 \pm 0.0000$ | $0.0307 \pm 0.0002$ | $0.0238 \pm 0.0001$ | $0.0128 \pm 0.0001$ |
| JTNP | $0.0297 \pm 0.0001$ | $0.0124 \pm 0.0000$ | $0.0041 \pm 0.0000$ | $0.0175 \pm 0.0002$ | $0.0120 \pm 0.0001$ | $0.0138 \pm 0.0000$ |
| MTNP | $\mathbf{0.0283 \pm 0.0001}$ | $\mathbf{0.0108 \pm 0.0000}$ | $\mathbf{0.0025 \pm 0.0000}$ | $\mathbf{0.0160 \pm 0.0001}$ | $\mathbf{0.0067 \pm 0.0000}$ | $\mathbf{0.0040 \pm 0.0000}$ |

| task | Segment (performance) | | | Segment (coherency) | | |
|------|------|------|------|------|------|------|
| $m$ | 10 | 100 | 512 | 10 | 100 | 512 |
| STNP | $0.6155 \pm 0.0024$ | $0.3961 \pm 0.0014$ | $0.2072 \pm 0.0012$ | $0.6503 \pm 0.0026$ | $0.5609 \pm 0.0012$ | $0.5197 \pm 0.0009$ |
| JTNP | $0.5746 \pm 0.0031$ | $0.3800 \pm 0.0011$ | $0.2339 \pm 0.0024$ | $0.5256 \pm 0.0009$ | $0.4959 \pm 0.0016$ | $0.5055 \pm 0.0007$ |
| MTNP | $\mathbf{0.5399 \pm 0.0019}$ | $\mathbf{0.3413 \pm 0.0017}$ | $\mathbf{0.1889 \pm 0.0011}$ | $\mathbf{0.5033 \pm 0.0014}$ | $\mathbf{0.4866 \pm 0.0016}$ | $\mathbf{0.4928 \pm 0.0006}$ |

| task | PNCC (performance) | | | PNCC (coherency) | | |
|------|------|------|------|------|------|------|
| $m$ | 10 | 100 | 512 | 10 | 100 | 512 |
| STNP | $0.0068 \pm 0.0001$ | $0.0009 \pm 0.0000$ | $0.0005 \pm 0.0000$ | $0.0317 \pm 0.0005$ | $0.0260 \pm 0.0003$ | $0.0209 \pm 0.0002$ |
| JTNP | $0.0073 \pm 0.0001$ | $0.0017 \pm 0.0000$ | $0.0007 \pm 0.0000$ | $0.0116 \pm 0.0002$ | $0.0146 \pm 0.0001$ | $0.0174 \pm 0.0001$ |
| MTNP | $\mathbf{0.0052 \pm 0.0000}$ | $\mathbf{0.0007 \pm 0.0000}$ | $\mathbf{0.0004 \pm 0.0000}$ | $\mathbf{0.0098 \pm 0.0001}$ | $\mathbf{0.0120 \pm 0.0001}$ | $\mathbf{0.0136 \pm 0.0001}$ |

Figure 13: Qualitative results on 2D function regression. ($\gamma = 0$)

Table 23: Average reconstruction errors (performance) and disagreement of predictions (coherency) on 2D function regression, with varying context size ($m$) and $\gamma = 0.25$.

| task | RGB (performance) | | |
|---|---|---|---|
| $m$ | 10 | 100 | 512 |
| STNP | $0.0378 \pm 0.0002$ | $0.0121 \pm 0.0001$ | $0.0041 \pm 0.0000$ |
| S+JTNP | $0.0342 \pm 0.0003$ | $0.0094 \pm 0.0001$ | $0.0031 \pm 0.0000$ |
| MTNP | $\mathbf{0.0329 \pm 0.0002}$ | $\mathbf{0.0084 \pm 0.0000}$ | $\mathbf{0.0021 \pm 0.0000}$ |

| task | Edge (performance) | | | Edge (coherency) | | |
|---|---|---|---|---|---|---|
| $m$ | 10 | 100 | 512 | 10 | 100 | 512 |
| STNP | $0.0349 \pm 0.0001$ | $0.0223 \pm 0.0001$ | $0.0085 \pm 0.0000$ | $0.0309 \pm 0.0002$ | $0.0254 \pm 0.0002$ | $0.0146 \pm 0.0001$ |
| S+JTNP | $0.0312 \pm 0.0001$ | $0.0152 \pm 0.0000$ | $0.0061 \pm 0.0000$ | $0.0184 \pm 0.0002$ | $0.0119 \pm 0.0001$ | $0.0139 \pm 0.0000$ |
| MTNP | $\mathbf{0.0298 \pm 0.0001}$ | $\mathbf{0.0133 \pm 0.0000}$ | $\mathbf{0.0040 \pm 0.0000}$ | $\mathbf{0.0176 \pm 0.0001}$ | $\mathbf{0.0079 \pm 0.0001}$ | $\mathbf{0.0046 \pm 0.0000}$ |

| task | Segment (performance) | | | Segment (coherency) | | |
|---|---|---|---|---|---|---|
| $m$ | 10 | 100 | 512 | 10 | 100 | 512 |
| STNP | $0.6315 \pm 0.0040$ | $0.4243 \pm 0.0021$ | $0.2463 \pm 0.0017$ | $0.6587 \pm 0.0012$ | $0.5766 \pm 0.0014$ | $0.5246 \pm 0.0011$ |
| S+JTNP | $0.5940 \pm 0.0024$ | $0.4013 \pm 0.0013$ | $0.2741 \pm 0.0018$ | $0.5303 \pm 0.0011$ | $0.5037 \pm 0.0008$ | $0.5076 \pm 0.0013$ |
| MTNP | $\mathbf{0.5615 \pm 0.0022}$ | $\mathbf{0.3677 \pm 0.0009}$ | $\mathbf{0.2374 \pm 0.0014}$ | $\mathbf{0.5103 \pm 0.0028}$ | $\mathbf{0.4910 \pm 0.0011}$ | $\mathbf{0.4935 \pm 0.0010}$ |

| task | PNCC (performance) | | | PNCC (coherency) | | |
|---|---|---|---|---|---|---|
| $m$ | 10 | 100 | 512 | 10 | 100 | 512 |
| STNP | $0.0079 \pm 0.0001$ | $0.0011 \pm 0.0000$ | $0.0005 \pm 0.0000$ | $0.0334 \pm 0.0003$ | $0.0267 \pm 0.0001$ | $0.0218 \pm 0.0002$ |
| S+JTNP | $0.0082 \pm 0.0001$ | $0.0018 \pm 0.0000$ | $0.0008 \pm 0.0000$ | $0.0109 \pm 0.0001$ | $0.0141 \pm 0.0001$ | $0.0181 \pm 0.0001$ |
| MTNP | $\mathbf{0.0061 \pm 0.0000}$ | $\mathbf{0.0009 \pm 0.0000}$ | $\mathbf{0.0005 \pm 0.0000}$ | $\mathbf{0.0098 \pm 0.0001}$ | $\mathbf{0.0117 \pm 0.0001}$ | $\mathbf{0.0135 \pm 0.0002}$ |

Figure 14: Qualitative results on 2D function regression. ($\gamma = 0.25$)

Table 24: Average reconstruction errors (performance) and disagreement of predictions (coherency) on 2D function regression, with varying context size ($m$) and $\gamma = 0.5$.

| task | RGB (performance) | | |
|---|---|---|---|
| $m$ | 10 | 100 | 512 |
| STNP | $0.0446 \pm 0.0005$ | $0.0154 \pm 0.0001$ | $0.0054 \pm 0.0000$ |
| S+JTNP | $0.0422 \pm 0.0004$ | $0.0129 \pm 0.0001$ | $0.0046 \pm 0.0000$ |
| MTNP | $\mathbf{0.0407 \pm 0.0005}$ | $\mathbf{0.0113 \pm 0.0000}$ | $\mathbf{0.0032 \pm 0.0000}$ |

| task | Edge (performance) | | | Edge (coherency) | | |
|---|---|---|---|---|---|---|
| $m$ | 10 | 100 | 512 | 10 | 100 | 512 |
| STNP | $0.0359 \pm 0.0001$ | $0.0256 \pm 0.0001$ | $0.0116 \pm 0.0000$ | $0.0317 \pm 0.0002$ | $0.0271 \pm 0.0002$ | $0.0173 \pm 0.0001$ |
| S+JTNP | $0.0337 \pm 0.0001$ | $0.0191 \pm 0.0000$ | $0.0090 \pm 0.0000$ | $0.0200 \pm 0.0002$ | $0.0122 \pm 0.0001$ | $0.0142 \pm 0.0001$ |
| MTNP | $\mathbf{0.0324 \pm 0.0001}$ | $\mathbf{0.0167 \pm 0.0000}$ | $\mathbf{0.0060 \pm 0.0000}$ | $\mathbf{0.0196 \pm 0.0001}$ | $\mathbf{0.0097 \pm 0.0002}$ | $\mathbf{0.0053 \pm 0.0000}$ |

| task | Segment (performance) | | | Segment (coherency) | | |
|---|---|---|---|---|---|---|
| $m$ | 10 | 100 | 512 | 10 | 100 | 512 |
| STNP | $0.6641 \pm 0.0021$ | $0.4669 \pm 0.0009$ | $0.2969 \pm 0.0019$ | $0.6710 \pm 0.0021$ | $0.5966 \pm 0.0027$ | $0.5353 \pm 0.0012$ |
| S+JTNP | $0.6322 \pm 0.0024$ | $0.4347 \pm 0.0013$ | $0.3176 \pm 0.0018$ | $0.5317 \pm 0.0015$ | $0.5169 \pm 0.0019$ | $0.5126 \pm 0.0006$ |
| MTNP | $\mathbf{0.6095 \pm 0.0025}$ | $\mathbf{0.4010 \pm 0.0016}$ | $\mathbf{0.2891 \pm 0.0019}$ | $\mathbf{0.5224 \pm 0.0021}$ | $\mathbf{0.4956 \pm 0.0011}$ | $\mathbf{0.4948 \pm 0.0004}$ |

| task | PNCC (performance) | | | PNCC (coherency) | | |
|---|---|---|---|---|---|---|
| $m$ | 10 | 100 | 512 | 10 | 100 | 512 |
| STNP | $0.0102 \pm 0.0002$ | $0.0015 \pm 0.0000$ | $0.0006 \pm 0.0000$ | $0.0382 \pm 0.0005$ | $0.0271 \pm 0.0003$ | $0.0232 \pm 0.0002$ |
| S+JTNP | $0.0104 \pm 0.0002$ | $0.0022 \pm 0.0000$ | $0.0009 \pm 0.0000$ | $0.0104 \pm 0.0002$ | $0.0134 \pm 0.0001$ | $0.0192 \pm 0.0002$ |
| MTNP | $\mathbf{0.0082 \pm 0.0001}$ | $\mathbf{0.0012 \pm 0.0000}$ | $\mathbf{0.0006 \pm 0.0000}$ | $\mathbf{0.0098 \pm 0.0001}$ | $\mathbf{0.0112 \pm 0.0001}$ | $\mathbf{0.0132 \pm 0.0001}$ |

Figure 15: Qualitative results on 2D function regression. ($\gamma = 0.5$)

Table 25: Average reconstruction errors (performance) and disagreement of predictions (coherency) on 2D function regression, with varying context size ($m$) and $\gamma = 0.75$.

| task | RGB (performance) | | |
|------|------|------|------|
| $m$ | 10 | 100 | 512 |
| STNP | $0.0512 \pm 0.0005$ | $0.0224 \pm 0.0001$ | $0.0086 \pm 0.0000$ |
| S+JTNP | $0.0496 \pm 0.0005$ | $0.0203 \pm 0.0001$ | $0.0079 \pm 0.0000$ |
| MTNP | $\mathbf{0.0483 \pm 0.0002}$ | $\mathbf{0.0178 \pm 0.0001}$ | $\mathbf{0.0058 \pm 0.0000}$ |

| task | Edge (performance) | | | Edge (coherency) | | |
|------|------|------|------|------|------|------|
| $m$ | 10 | 100 | 512 | 10 | 100 | 512 |
| STNP | $0.0366 \pm 0.0001$ | $0.0302 \pm 0.0001$ | $0.0176 \pm 0.0000$ | $0.0323 \pm 0.0003$ | $0.0294 \pm 0.0001$ | $0.0222 \pm 0.0001$ |
| S+JTNP | $0.0356 \pm 0.0001$ | $0.0251 \pm 0.0001$ | $0.0145 \pm 0.0000$ | $0.0214 \pm 0.0002$ | $0.0138 \pm 0.0000$ | $0.0157 \pm 0.0001$ |
| MTNP | $\mathbf{0.0344 \pm 0.0001}$ | $\mathbf{0.0223 \pm 0.0001}$ | $\mathbf{0.0103 \pm 0.0000}$ | $\mathbf{0.0202 \pm 0.0001}$ | $\mathbf{0.0130 \pm 0.0001}$ | $\mathbf{0.0069 \pm 0.0000}$ |

| task | Segment (performance) | | | Segment (coherency) | | |
|------|------|------|------|------|------|------|
| $m$ | 10 | 100 | 512 | 10 | 100 | 512 |
| STNP | $0.6907 \pm 0.0018$ | $0.5351 \pm 0.0021$ | $0.3712 \pm 0.0014$ | $0.6804 \pm 0.0015$ | $0.6297 \pm 0.0023$ | $0.5601 \pm 0.0014$ |
| S+JTNP | $0.6611 \pm 0.0013$ | $0.4916 \pm 0.0024$ | $0.3778 \pm 0.0031$ | $0.5361 \pm 0.0016$ | $0.5385 \pm 0.0004$ | $0.5298 \pm 0.0009$ |
| MTNP | $\mathbf{0.6485 \pm 0.0029}$ | $\mathbf{0.4573 \pm 0.0025}$ | $\mathbf{0.3472 \pm 0.0017}$ | $\mathbf{0.5307 \pm 0.0012}$ | $\mathbf{0.5031 \pm 0.0014}$ | $\mathbf{0.4962 \pm 0.0010}$ |

| task | PNCC (performance) | | | PNCC (coherency) | | |
|------|------|------|------|------|------|------|
| $m$ | 10 | 100 | 512 | 10 | 100 | 512 |
| STNP | $0.0123 \pm 0.0001$ | $0.0030 \pm 0.0001$ | $0.0008 \pm 0.0000$ | $0.0453 \pm 0.0005$ | $0.0283 \pm 0.0003$ | $0.0255 \pm 0.0001$ |
| S+JTNP | $0.0126 \pm 0.0001$ | $0.0036 \pm 0.0001$ | $0.0011 \pm 0.0000$ | $0.0114 \pm 0.0002$ | $0.0121 \pm 0.0001$ | $0.0195 \pm 0.0003$ |
| MTNP | $\mathbf{0.0103 \pm 0.0001}$ | $\mathbf{0.0023 \pm 0.0000}$ | $\mathbf{0.0007 \pm 0.0000}$ | $\mathbf{0.0091 \pm 0.0001}$ | $\mathbf{0.0102 \pm 0.0001}$ | $\mathbf{0.0126 \pm 0.0001}$ |

Figure 16: Qualitative results on 2D function regression. ($\gamma = 0.75$)

# H ABLATION STUDY

In this section, we provide results of ablation study on the implementations of MTNP. We conduct experiments on synthetic and weather datasets to analyze the effect of various design choices we made for MTNP.

## H.1 ABLATION STUDY ON SELF-ATTENTION LAYERS AND PARAMETRIC POOLING

In this study, we explore the effect of using self-attention layers before pooling operations and using PMA layer for pooling in MTNP. The variants of MTNP are as follows. (1) MTNP-A: MTNP without self-attention and using average pooling, (2) MTNP-P: MTNP without self-attention and using PMA, (3) MTNP-SA: MTNP with self-attention and using average pooling, (4) MTNP-SP: MTNP with self-attention and using PMA. The results are summarized below. Note that we consistently use MTNP-SP architecture for the experiments in Section 5. As shown in the result, MTNP with self-attention outperforms the one without self-attention by a large margin, implying that self-attention is critical in MTNP.

Table 27: Average normalized MSE on synthetic tasks, with varying context size ($m$) and $\gamma = 0.5$.

| task | Sine | | | Tanh | | |
|---|---|---|---|---|---|---|
| $m$ | 5 | 10 | 20 | 5 | 10 | 20 |
| MTNP-A | $0.4157 \pm 0.0174$ | $0.2855 \pm 0.0259$ | $0.1913 \pm 0.0148$ | $0.1121 \pm 0.0219$ | $0.0549 \pm 0.0108$ | $0.0320 \pm 0.0017$ |
| MTNP-P | $0.3568 \pm 0.0208$ | $0.2090 \pm 0.0088$ | $0.1185 \pm 0.0066$ | $0.0780 \pm 0.0131$ | $0.0296 \pm 0.0035$ | $0.0153 \pm 0.0013$ |
| MTNP-SA | $0.2668 \pm 0.0107$ | $0.1299 \pm 0.0118$ | $0.0543 \pm 0.0120$ | $0.0452 \pm 0.0085$ | $\mathbf{0.0103} \pm 0.0020$ | $\mathbf{0.0033} \pm 0.0003$ |
| MTNP-SP | $\mathbf{0.2636} \pm 0.0105$ | $\mathbf{0.1137} \pm 0.0078$ | $\mathbf{0.0485} \pm 0.0034$ | $\mathbf{0.0435} \pm 0.0047$ | $0.0115 \pm 0.0021$ | $0.0040 \pm 0.0002$ |

| task | Sigmoid | | | Gaussian | | |
|---|---|---|---|---|---|---|
| $m$ | 5 | 10 | 20 | 5 | 10 | 20 |
| MTNP-A | $0.0158 \pm 0.0028$ | $0.0082 \pm 0.0012$ | $0.0050 \pm 0.0002$ | $0.0711 \pm 0.0054$ | $0.0454 \pm 0.0042$ | $0.0321 \pm 0.0018$ |
| MTNP-P | $0.0119 \pm 0.0021$ | $0.0052 \pm 0.0005$ | $0.0033 \pm 0.0003$ | $0.0519 \pm 0.0029$ | $0.0270 \pm 0.0031$ | $0.0161 \pm 0.0008$ |
| MTNP-SA | $0.0071 \pm 0.0018$ | $0.0015 \pm 0.0003$ | $\mathbf{0.0005} \pm 0.0001$ | $\mathbf{0.0345} \pm 0.0048$ | $\mathbf{0.0132} \pm 0.0007$ | $\mathbf{0.0065} \pm 0.0010$ |
| MTNP-SP | $\mathbf{0.0066} \pm 0.0019$ | $\mathbf{0.0014} \pm 0.0001$ | $0.0006 \pm 0.0001$ | $0.0360 \pm 0.0018$ | $\mathbf{0.0132} \pm 0.0008$ | $0.0069 \pm 0.0012$ |

Table 28: Average MSE and NLL on weather tasks, with $m = 10$ and $\gamma = 0.5$.

| task | TempMin | | TempMax | | Humidity | |
|---|---|---|---|---|---|---|
| metric | MSE | NLL | MSE | NLL | MSE | NLL |
| MTNP-A | $0.0067 \pm 0.0007$ | $-1.0773 \pm 0.0231$ | $0.0119 \pm 0.0012$ | $-0.8658 \pm 0.0405$ | $0.0745 \pm 0.0033$ | $-0.0471 \pm 0.0278$ |
| MTNP-P | $0.0046 \pm 0.0003$ | $-1.1614 \pm 0.0129$ | $0.0071 \pm 0.0002$ | $-1.0166 \pm 0.0145$ | $0.0581 \pm 0.0029$ | $-0.1463 \pm 0.0232$ |
| MTNP-SA | $0.0039 \pm 0.0001$ | $\mathbf{-1.1881} \pm 0.0074$ | $0.0058 \pm 0.0005$ | $-1.0990 \pm 0.0203$ | $\mathbf{0.0542} \pm 0.0024$ | $\mathbf{-0.2083} \pm 0.0283$ |
| MTNP-SP | $\mathbf{0.0037} \pm 0.0001$ | $-1.1832 \pm 0.0165$ | $\mathbf{0.0054} \pm 0.0001$ | $\mathbf{-1.1049} \pm 0.0154$ | $0.0546 \pm 0.0021$ | $-0.1006 \pm 0.0696$ |

| task | Precip | | Cloud | | Dew | |
|---|---|---|---|---|---|---|
| metric | MSE | NLL | MSE | NLL | MSE | NLL |
| MTNP-A | $0.2514 \pm 0.0049$ | $0.6655 \pm 0.0212$ | $0.2578 \pm 0.0070$ | $0.7186 \pm 0.0369$ | $0.0114 \pm 0.0012$ | $-0.9103 \pm 0.0316$ |
| MTNP-P | $0.2400 \pm 0.0072$ | $\mathbf{0.6205} \pm 0.0210$ | $0.2360 \pm 0.0070$ | $\mathbf{0.6532} \pm 0.0166$ | $0.0083 \pm 0.0001$ | $-1.0183 \pm 0.0039$ |
| MTNP-SA | $\mathbf{0.2262} \pm 0.0041$ | $0.6270 \pm 0.0288$ | $0.2242 \pm 0.0042$ | $0.6825 \pm 0.0280$ | $0.0074 \pm 0.0004$ | $\mathbf{-1.0409} \pm 0.0200$ |
| MTNP-SP | $0.2276 \pm 0.0028$ | $0.6557 \pm 0.0433$ | $\mathbf{0.2215} \pm 0.0043$ | $0.6660 \pm 0.0141$ | $\mathbf{0.0073} \pm 0.0003$ | $-1.0331 \pm 0.0147$ |

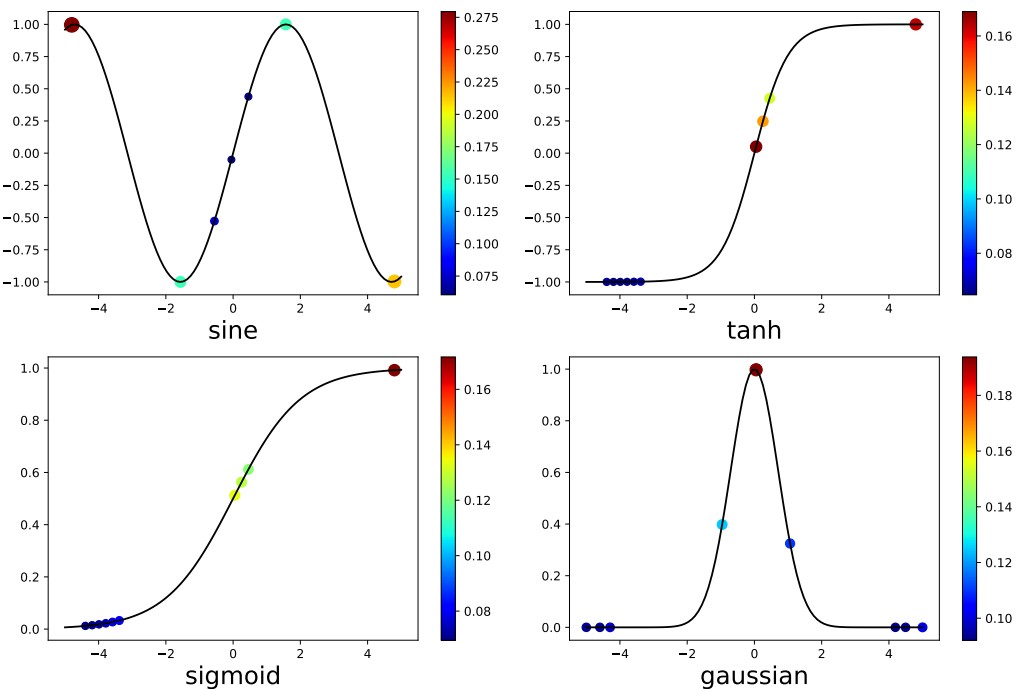

Figure 17: Visualization of per-task attention weights of MTNP assigned on different context points in synthetic tasks. Size and color of markers represent the amount of attention weight assigned on each point.

To investigate whether the self-attention modules operate as desired in per-task attention stage, we visualize the attention weights assigned to each context point. For each task, we compute an averaged attention weight placed on a set of context points by averaging dimensions of the attention map of shape $(nL, nH, nQ, nK)$ to $(1, 1, 1, nK)$ where $nL$ and $nH$ denote the number of self-attention layers and heads in each layer, and $nQ = nK$ refer to the number of query $(Q)$ and key $(K)$ vectors and $Q = K$. Note that this averaged attention weight can be interpreted as the averaged importance put on each of the context points by the self-attention module. Results are visualized in Figure 17. As shown in the figure, points located at representative positions or at sparse regions are attended more than others. This shows that self-attention in per-task paths operates as desired.

## H.2 EFFECT OF PARAMETER-SHARING IN PER-TASK BRANCHES

In this study, we explore the effect of parameter-sharing the per-task encoder networks and decoder networks in STNP and MTNP. The variants of STNP and MTNP are as follows. (1) STNP-S: STNP with shared encoder and decoder, (2) STNP-TS: STNP with task-specific encoders and decoders, (3) MTNP-S: MTNP with shared encoder and decoder in per-task branches, (4) MTNP-TS: MTNP with task-specific encoders and decoders in per-task branches. Note that we consistently use STNP-TS and MTNP-S architectures for the 1D experiments (synthetic and weather) and STNP-TS and MTNP-TS architectures for the 2D experiments (CelebA) in Section 5

As can be seen in the tables, in the weather dataset, we observe that the models with parameter sharing (STNP-S & MTNP-S) show comparable performances to their respective non-sharing baselines (STNP-TS & MTNP-TS). On the other hand, the parameter sharing technique slightly improves the performances of the models in the 1-D synthetic case. We conjecture that the utilization of the same architecture and its parameter for all tasks acts as a good inductive bias to the models considering that all tasks share the same global parameter a,b,c,w. Nonetheless, we still find that MTNPs consistently outperform STNPs regardless of whether the parameters are shared or not, validating the effectiveness of MTNP to capture and exploit functional correlation for multi-task learning problems.

Table 29: Average normalized MSE on synthetic tasks, with varying context size ($m$) and $\gamma = 0.5$.

| task | Sine | | | Tanh | | |
|---|---|---|---|---|---|---|
| $m$ | 5 | 10 | 20 | 5 | 10 | 20 |
| STNP-S | $0.5387 \pm 0.0270$ | $0.2596 \pm 0.0351$ | $0.0845 \pm 0.0216$ | $0.1255 \pm 0.0143$ | $0.0459 \pm 0.0095$ | $0.0123 \pm 0.0019$ |
| STNP-TS | $0.5212 \pm 0.0157$ | $0.2609 \pm 0.0382$ | $0.0993 \pm 0.0182$ | $0.1307 \pm 0.0134$ | $0.0468 \pm 0.0074$ | $0.0159 \pm 0.0028$ |
| MTNP-S | $\mathbf{0.2636} \pm 0.0105$ | $\mathbf{0.1137} \pm 0.0078$ | $\mathbf{0.0485} \pm 0.0034$ | $\mathbf{0.0435} \pm 0.0047$ | $\mathbf{0.0115} \pm 0.0021$ | $\mathbf{0.0040} \pm 0.0002$ |
| MTNP-TS | $0.2901 \pm 0.0164$ | $0.1297 \pm 0.0079$ | $0.0532 \pm 0.0050$ | $0.0618 \pm 0.0069$ | $0.0172 \pm 0.0041$ | $0.0063 \pm 0.0006$ |

| task | Sigmoid | | | Gaussian | | |
|---|---|---|---|---|---|---|
| $m$ | 5 | 10 | 20 | 5 | 10 | 20 |
| STNP-S | $0.0177 \pm 0.0034$ | $0.0050 \pm 0.0009$ | $0.0012 \pm 0.0003$ | $0.0801 \pm 0.0124$ | $0.0371 \pm 0.0048$ | $0.0181 \pm 0.0028$ |
| STNP-TS | $0.0203 \pm 0.0034$ | $0.0067 \pm 0.0013$ | $0.0025 \pm 0.0005$ | $0.0799 \pm 0.0098$ | $0.0409 \pm 0.0041$ | $0.0222 \pm 0.0042$ |
| MTNP-S | $\mathbf{0.0066} \pm 0.0019$ | $\mathbf{0.0014} \pm 0.0001$ | $\mathbf{0.0006} \pm 0.0001$ | $\mathbf{0.0360} \pm 0.0018$ | $\mathbf{0.0132} \pm 0.0008$ | $\mathbf{0.0069} \pm 0.0012$ |
| MTNP-TS | $0.0093 \pm 0.0015$ | $0.0031 \pm 0.0010$ | $0.0014 \pm 0.0001$ | $0.0426 \pm 0.0025$ | $0.0173 \pm 0.0019$ | $0.0100 \pm 0.0006$ |

Table 30: Average MSE and NLL on weather tasks, with $m = 10$ and $\gamma = 0.5$.

| task | TempMin | | TempMax | | Humidity | |
|---|---|---|---|---|---|---|
| metric | MSE | NLL | MSE | NLL | MSE | NLL |
| STNP-S | $0.0049 \pm 0.0003$ | $-1.1381 \pm 0.0223$ | $0.0066 \pm 0.0006$ | $-1.0527 \pm 0.0241$ | $0.0621 \pm 0.0069$ | $0.0240 \pm 0.1210$ |
| STNP-TS | $0.0046 \pm 0.0004$ | $-1.1514 \pm 0.0181$ | $0.0069 \pm 0.0004$ | $-1.0390 \pm 0.0106$ | $0.0632 \pm 0.0072$ | $0.1273 \pm 0.1898$ |
| MTNP-S | $0.0037 \pm 0.0001$ | $\mathbf{-1.1832} \pm 0.0165$ | $0.0054 \pm 0.0001$ | $\mathbf{-1.1049} \pm 0.0154$ | $0.0546 \pm 0.0021$ | $\mathbf{-0.1006} \pm 0.0696$ |
| MTNP-TS | $\mathbf{0.0036} \pm 0.0002$ | $-1.1818 \pm 0.0076$ | $\mathbf{0.0053} \pm 0.0003$ | $-1.0885 \pm 0.0246$ | $\mathbf{0.0519} \pm 0.0013$ | $-0.0662 \pm 0.0828$ |

| task | Precip | | Cloud | | Dew | |
|---|---|---|---|---|---|---|
| metric | MSE | NLL | MSE | NLL | MSE | NLL |
| STNP-S | $0.2675 \pm 0.0104$ | $0.9537 \pm 0.1347$ | $0.2629 \pm 0.0043$ | $0.7847 \pm 0.0503$ | $0.0084 \pm 0.0007$ | $-0.9877 \pm 0.0312$ |
| STNP-TS | $0.2607 \pm 0.0082$ | $1.1242 \pm 0.2362$ | $0.2631 \pm 0.0044$ | $0.8563 \pm 0.0637$ | $0.0086 \pm 0.0008$ | $-0.9815 \pm 0.0283$ |
| MTNP-S | $0.2276 \pm 0.0028$ | $\mathbf{0.6557} \pm 0.0433$ | $\mathbf{0.2215} \pm 0.0043$ | $\mathbf{0.6660} \pm 0.0141$ | $0.0073 \pm 0.0003$ | $\mathbf{-1.0331} \pm 0.0147$ |
| MTNP-TS | $\mathbf{0.2187} \pm 0.0043$ | $0.7213 \pm 0.0953$ | $0.2253 \pm 0.0100$ | $0.6663 \pm 0.0283$ | $\mathbf{0.0071} \pm 0.0003$ | $-1.0232 \pm 0.0144$ |

## H.3 EFFECT OF LATENT AND DETERMINISTIC ENCODERS

In this study, we compare STNP, JTNP, and MTNP without using deterministic encoder, each corresponds to STNP-L, JTNP-L, and MTNP-L in the table. Note that these variants correspond to the direct NP implementations of STNP/JTNP/MTNP rather than ANP. The results are provided in the tables below. The overall trends are the same as the models with deterministic encoder, which demonstrates that the effectiveness of MTNP does not depend on a specific choice of architecture (vanilla NP or ANP).

Table 31: Average normalized MSE on synthetic tasks, with varying context size ($m$) and $\gamma = 0.5$.

| task | Sine | | | Tanh | | |
|---|---|---|---|---|---|---|
| $m$ | 5 | 10 | 20 | 5 | 10 | 20 |
| STNP-L | $0.5615 \pm 0.0199$ | $0.3006 \pm 0.0407$ | $0.1254 \pm 0.0159$ | $0.1362 \pm 0.0119$ | $0.0585 \pm 0.0109$ | $0.0234 \pm 0.0020$ |
| S+JTNP-L | $0.4151 \pm 0.0273$ | $0.2560 \pm 0.0184$ | $0.1395 \pm 0.0106$ | $0.1055 \pm 0.0171$ | $0.0506 \pm 0.0054$ | $0.0224 \pm 0.0023$ |
| MTNP-L | $\mathbf{0.3003} \pm 0.0147$ | $\mathbf{0.1543} \pm 0.0130$ | $\mathbf{0.0755} \pm 0.0080$ | $\mathbf{0.0563} \pm 0.0069$ | $\mathbf{0.0162} \pm 0.0020$ | $\mathbf{0.0067} \pm 0.0003$ |

| task | Sigmoid | | | Gaussian | | |
|---|---|---|---|---|---|---|
| $m$ | 5 | 10 | 20 | 5 | 10 | 20 |
| STNP-L | $0.0213 \pm 0.0025$ | $0.0086 \pm 0.0008$ | $0.0045 \pm 0.0006$ | $0.0809 \pm 0.0101$ | $0.0405 \pm 0.0063$ | $0.0234 \pm 0.0026$ |
| S+JTNP-L | $0.0201 \pm 0.0030$ | $0.0106 \pm 0.0008$ | $0.0061 \pm 0.0003$ | $0.0687 \pm 0.0056$ | $0.0376 \pm 0.0014$ | $0.0227 \pm 0.0019$ |
| MTNP-L | $\mathbf{0.0096} \pm 0.0022$ | $\mathbf{0.0027} \pm 0.0005$ | $\mathbf{0.0012} \pm 0.0002$ | $\mathbf{0.0417} \pm 0.0037$ | $\mathbf{0.0169} \pm 0.0010$ | $\mathbf{0.0091} \pm 0.0007$ |

Table 32: Average MSE and NLL on weather tasks, with $m = 10$ and $\gamma = 0.5$.

| task | TempMin | | TempMax | | Humidity | |
|---|---|---|---|---|---|---|
| metric | MSE | NLL | MSE | NLL | MSE | NLL |
| STNP-L | $0.0065 \pm 0.0010$ | $-1.0783 \pm 0.0207$ | $0.0102 \pm 0.0008$ | $-0.9202 \pm 0.0205$ | $0.0828 \pm 0.0067$ | $0.0341 \pm 0.0401$ |
| S+JTNP-L | $0.0085 \pm 0.0016$ | $-1.0632 \pm 0.0252$ | $0.0136 \pm 0.0021$ | $-0.8601 \pm 0.0236$ | $0.0966 \pm 0.0108$ | $0.0737 \pm 0.0699$ |
| MTNP-L | $\mathbf{0.0046} \pm 0.0005$ | $\mathbf{-1.1580} \pm 0.0167$ | $\mathbf{0.0072} \pm 0.0005$ | $\mathbf{-1.0203} \pm 0.0171$ | $\mathbf{0.0596} \pm 0.0027$ | $\mathbf{-0.1554} \pm 0.0274$ |

| task | Precip | | Cloud | | Dew | |
|---|---|---|---|---|---|---|
| metric | MSE | NLL | MSE | NLL | MSE | NLL |
| STNP-L | $0.3381 \pm 0.0026$ | $0.8792 \pm 0.0008$ | $0.3411 \pm 0.0049$ | $0.8798 \pm 0.0066$ | $0.0106 \pm 0.0005$ | $-0.8804 \pm 0.0118$ |
| S+JTNP-L | $0.2832 \pm 0.0095$ | $0.7347 \pm 0.0179$ | $0.2851 \pm 0.0123$ | $0.7692 \pm 0.0111$ | $0.0127 \pm 0.0007$ | $-0.8827 \pm 0.0220$ |
| MTNP-L | $\mathbf{0.2432} \pm 0.0046$ | $\mathbf{0.6267} \pm 0.0220$ | $\mathbf{0.2372} \pm 0.0056$ | $\mathbf{0.6659} \pm 0.0163$ | $\mathbf{0.0085} \pm 0.0003$ | $\mathbf{-1.0006} \pm 0.0065$ |

We also compare MTNP with deterministic encoder only, which corresponds to MTNP-D in the table below. To emphasize the benefits of generative modeling of MTNPs, we include MTNP evaluated on the best sample among 25 predictive samples, which corresponds to MTNP-best in the table. We observe that MTNP and MTNP-D are comparable in synthetic and weather datasets, which seems reasonable as we designed the deterministic encoder to mimic the latent encoder of MTNP (*e.g.*, they employ both per-task and across-task inferences). However, we can see that MTNP-best clearly outperforms MTNP-D, which implies that MTNP can generate more accurate samples while MTNP-D cannot.

Table 33: Average normalized MSE on synthetic tasks, with varying context size ($m$) and $\gamma = 0.5$.

| task | Sine | | | Tanh | | |
|---|---|---|---|---|---|---|
| $m$ | 5 | 10 | 20 | 5 | 10 | 20 |
| MTNP-D | $0.1963 \pm 0.0039$ | $0.0985 \pm 0.0059$ | $0.0447 \pm 0.0034$ | $0.0423 \pm 0.0079$ | $0.0110 \pm 0.0015$ | $0.0042 \pm 0.0007$ |
| MTNP | $0.2636 \pm 0.0105$ | $0.1137 \pm 0.0078$ | $0.0485 \pm 0.0034$ | $0.0435 \pm 0.0047$ | $0.0115 \pm 0.0021$ | $0.0040 \pm 0.0002$ |
| MTNP-best | $\mathbf{0.1239} \pm 0.0087$ | $\mathbf{0.0491} \pm 0.0068$ | $\mathbf{0.0176} \pm 0.0032$ | $\mathbf{0.0169} \pm 0.0028$ | $\mathbf{0.0032} \pm 0.0009$ | $\mathbf{0.0009} \pm 0.0001$ |

| task | Sigmoid | | | Gaussian | | |
|---|---|---|---|---|---|---|
| $m$ | 5 | 10 | 20 | 5 | 10 | 20 |
| MTNP-D | $0.0073 \pm 0.0011$ | $0.0015 \pm 0.0004$ | $0.0006 \pm 0.0002$ | $0.0281 \pm 0.0018$ | $0.0127 \pm 0.0015$ | $0.0073 \pm 0.0012$ |
| MTNP | $0.0066 \pm 0.0019$ | $0.0014 \pm 0.0001$ | $0.0006 \pm 0.0001$ | $0.0360 \pm 0.0018$ | $0.0132 \pm 0.0008$ | $0.0069 \pm 0.0012$ |
| MTNP-best | $\mathbf{0.0025} \pm 0.0007$ | $\mathbf{0.0004} \pm 0.0001$ | $\mathbf{0.0002} \pm 0.0000$ | $\mathbf{0.0146} \pm 0.0021$ | $\mathbf{0.0046} \pm 0.0006$ | $\mathbf{0.0021} \pm 0.0005$ |

Table 34: Average MSE and NLL on weather tasks, with $m = 10$ and $\gamma = 0.5$.

| task | TempMin | | TempMax | | Humidity | |
|---|---|---|---|---|---|---|
| metric | MSE | NLL | MSE | NLL | MSE | NLL |
| MTNP-D | $0.0039 \pm 0.0001$ | $-1.1892 \pm 0.0079$ | $0.0059 \pm 0.0002$ | $-1.0944 \pm 0.0168$ | $0.0535 \pm 0.0023$ | $-0.2142 \pm 0.0359$ |
| MTNP | $0.0037 \pm 0.0001$ | $-1.1832 \pm 0.0165$ | $0.0054 \pm 0.0001$ | $-1.1049 \pm 0.0154$ | $0.0546 \pm 0.0021$ | $-0.1006 \pm 0.0696$ |
| MTNP-best | $\mathbf{0.0032} \pm 0.0001$ | $\mathbf{-1.2143} \pm 0.0118$ | $\mathbf{0.0047} \pm 0.0001$ | $\mathbf{-1.1441} \pm 0.0111$ | $\mathbf{0.0489} \pm 0.0019$ | $\mathbf{-0.2320} \pm 0.0460$ |

| task | Precip | | Cloud | | Dew | |
|---|---|---|---|---|---|---|
| metric | MSE | NLL | MSE | NLL | MSE | NLL |
| MTNP-D | $0.2257 \pm 0.0035$ | $0.6034 \pm 0.0315$ | $0.2260 \pm 0.0038$ | $0.6536 \pm 0.0130$ | $0.0077 \pm 0.0003$ | $-1.0434 \pm 0.0150$ |
| MTNP | $0.2276 \pm 0.0028$ | $0.6557 \pm 0.0433$ | $0.2215 \pm 0.0043$ | $0.6660 \pm 0.0141$ | $0.0073 \pm 0.0003$ | $-1.0331 \pm 0.0147$ |
| MTNP-best | $\mathbf{0.2160} \pm 0.0026$ | $\mathbf{0.5491} \pm 0.0269$ | $\mathbf{0.2091} \pm 0.0041$ | $\mathbf{0.6044} \pm 0.0090$ | $\mathbf{0.0065} \pm 0.0003$ | $\mathbf{-1.0763} \pm 0.0102$ |

## H.4  EFFECT OF TASK EMBEDDINGS

In this study, we compare to different types of task embeddings for MTNP. MTNP-Onehot uses one-hot encoded vector for task embedding $e^t$ while MTNP-learnable uses learnable vector for the task embedding. Note that we consistently use MTNP-learnable for the 1D experiments in Section 5

Table 35: Average normalized MSE on synthetic tasks, with varying context size ($m$) and $\gamma = 0.5$.

| task | Sine | | | Tanh | | |
|---|---|---|---|---|---|---|
| $m$ | 5 | 10 | 20 | 5 | 10 | 20 |
| MTNP-Onehot | **0.2507** ± 0.0174 | **0.1047** ± 0.0123 | **0.0426** ± 0.0075 | 0.0440 ± 0.0050 | **0.0110** ± 0.0014 | **0.0036** ± 0.0005 |
| MTNP-Learnable | 0.2636 ± 0.0105 | 0.1137 ± 0.0078 | 0.0485 ± 0.0034 | **0.0435** ± 0.0047 | 0.0115 ± 0.0021 | 0.0040 ± 0.0002 |

| task | Sigmoid | | | Gaussian | | |
|---|---|---|---|---|---|---|
| $m$ | 5 | 10 | 20 | 5 | 10 | 20 |
| MTNP-Onehot | 0.0067 ± 0.0013 | **0.0014** ± 0.0003 | **0.0005** ± 0.0001 | **0.0338** ± 0.0017 | **0.0130** ± 0.0009 | **0.0062** ± 0.0008 |
| MTNP-Learnable | **0.0066** ± 0.0019 | **0.0014** ± 0.0001 | 0.0006 ± 0.0001 | 0.0360 ± 0.0018 | 0.0132 ± 0.0008 | 0.0069 ± 0.0012 |

Table 36: Average MSE and NLL on weather tasks, with $m = 10$ and $\gamma = 0.5$.

| task | TempMin | | TempMax | | Humidity | |
|---|---|---|---|---|---|---|
| metric | MSE | NLL | MSE | NLL | MSE | NLL |
| MTNP-Onehot | **0.0036** ± 0.0002 | **-1.1975** ± 0.0088 | 0.0055 ± 0.0001 | -1.1046 ± 0.0090 | **0.0526** ± 0.0014 | -0.0835 ± 0.1149 |
| MTNP-Learnable | 0.0037 ± 0.0001 | -1.1832 ± 0.0165 | **0.0054** ± 0.0001 | **-1.1049** ± 0.0154 | 0.0546 ± 0.0021 | **-0.1006** ± 0.0696 |

| task | Precip | | Cloud | | Dew | |
|---|---|---|---|---|---|---|
| metric | MSE | NLL | MSE | NLL | MSE | NLL |
| MTNP-Onehot | **0.2265** ± 0.0034 | 0.7003 ± 0.0482 | 0.2244 ± 0.0044 | 0.6909 ± 0.0264 | **0.0070** ± 0.0004 | **-1.0479** ± 0.0155 |
| MTNP-Learnable | 0.2276 ± 0.0028 | **0.6557** ± 0.0433 | **0.2215** ± 0.0043 | **0.6660** ± 0.0141 | 0.0073 ± 0.0003 | -1.0331 ± 0.0147 |

We observe that MTNP with one-hot embedding is comparable to MTNP with learnable embedding. To further investigate the effect of learnable embedding, we visualize the learned task embedding by MTNP using the t-SNE algorithm. We include the visualization results in Figure 18 and 19. As shown in the figure, we find that the learned task embeddings are well-separated from each other and uniformly distributed on the embedding space. From the observations, we conjecture that well-separated task embeddings are sufficient task information for MTNP.

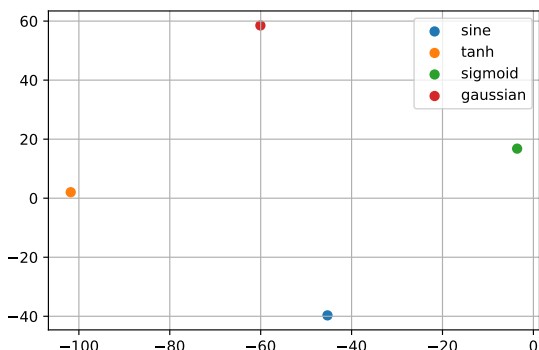

Figure 18: t-SNE plot (with 2 components) of the learned task embeddings of MTNP in synthetic tasks.

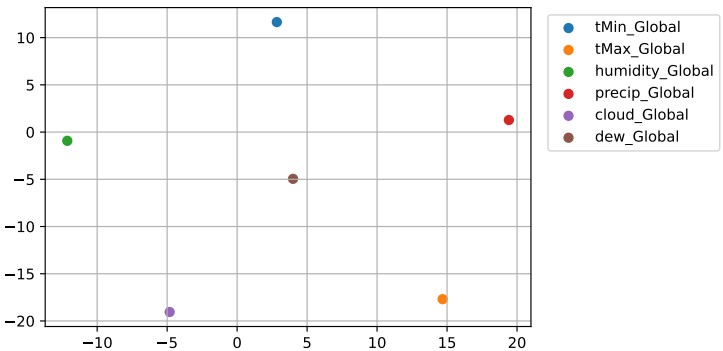

Figure 19: t-SNE plot (with 2 components) of the learned task embeddings of MTNP in weather tasks.

