# OpenReview forum: "Multi-Task Processes"
_ICLR.cc/2022/Conference — ICLR 2022 Poster_

### Official Review · Reviewer_ACBa · 2021-10-31

**Correctness:** 3
**Technical Novelty And Significance:** 3
**Empirical Novelty And Significance:** 3
**Recommendation:** 6
**Confidence:** 4

**Main Review:**

Strengths:

(1) The paper gives a novel perspective on improving neural processes for multi-task learning.

(2) This paper has comprehensive evaluations about the effects of the number of the context data, the various missing rates, task correlation and so on.

Weaknesses:

(1) In the implementation of the latent encoder, what’s the role of self-attentions before the pool operations?
Besides MSE, could the author provide some visualization results to demonstrate the difference between latent encoder with self-attention and without self-attention?

(2) It’s unclear to me about the motivation of replacing the average pooling operation in the stochastic path with a Pooling by Multihead Attention (PMA) layer. Could the author provide more information about the seed vector in the PMA? Is it also task-specific? What’s the advantage of the parametric pooling over the non-parametric pooling?

(3) In Figure3, why we cannot observe a similar conclusion as NPs/ ANPs/DSVNP, the lower variance around the context points?

(4) What’s the functional correlation captured by a global variable z for multi-task learning?

(5) To make the results convincing, more than two previous approaches should be compared in the experiments.


**Summary Of The Paper:**

This paper proposes a hierarchical latent variable model based on deep neural processes to model multiple functions jointly from incomplete data.

**Summary Of The Review:**

The advantages of NPs for multi-task learning should be further interpreted.

---

> ### Author Response · Authors · 2021-11-23
> **Rebuttal from Authors of Paper952 to Reviewer ACBa (4/4)**
>
> > **Q5.** To make the results convincing, more than two previous approaches should be compared in the experiments.
>
> **A5.** We appreciate the reviewer’s suggestion. For more comprehensive comparisons, we also included meta-learning baselines that use the same meta-train/meta-test data with our method. We chose MAML [b] and Reptile [c] as they are model-agnostic meta-learning methods that can be applied to our multi-task regression setting with incomplete data. The meta-training involved bi-level optimization where the inner loop optimizes the loss for context data and the outer loop optimizes the loss for target data. We employed a similar architecture to MTP for the baselines that consist of an encoder network shared by all tasks and task-specific decoder networks. For fair comparisons, we controlled the total number of parameters of the models similar to NP baselines (STP, JTP, MTP). We include the results in Section 5 in the main paper, Appendix D and E, and also summarize them below.
>
> [Results on synthetic tasks, $\gamma = 0.5$.]
>
> | Tasks |  | Sine |  |  | Tanh |  |  | Sigmoid |  |  | Gaussian |  |
> |:---:|:---:|:---:|:---:|:---:|:---:|:---:|:---:|:---:|:---:|:---:|:---:|:---:|
> | m | 5 | 10 | 20 | 5 | 10 | 20 | 5 | 10 | 20 | 5 | 10 | 20 |
> | MAML | 0.2962 | 0.1582 | 0.0701 | 0.0991 | 0.0342 | 0.0131 | 0.0321 | 0.0119 | 0.0069 | 0.0696 | 0.0353 | 0.0174 |
> | Reptile | 0.5164 | 0.2886 | 0.1414 | 0.1656 | 0.0557 | 0.0291 | 0.0619 | 0.0220 | 0.0181 | 0.1371 | 0.0679 | 0.0374 |
> | STP | 0.5212 | 0.2609 | 0.0993 | 0.1307 | 0.0468 | 0.0159 | 0.0203 | 0.0067 | 0.0025 | 0.0799 | 0.0409 | 0.0222 |
> | STP+JTP | 0.3848 | 0.2340 | 0.1114 | 0.1015 | 0.0418 | 0.0168 | 0.0163 | 0.0065 | 0.0032 | 0.0613 | 0.0318 | 0.0161 |
> | MTP | **0.2636** | **0.1137** | **0.0485** | **0.0435** | **0.0115** | **0.0040** | **0.0066** | **0.0014** | **0.0006** | **0.0360** | **0.0132** | **0.0069** |
>
>
> [Results on weather tasks, $\gamma = 0.5$ and $m = 10$.]
>
> | Tasks | TempMin |  | TempMax |  | Humidity |  | Precip |  | Cloud |  | Dew |  |
> |---:|---:|---:|---:|---:|---:|---:|---:|---:|---:|---:|---:|---:|
> | metric | MSE | NLL | MSE | NLL | MSE | NLL | MSE | NLL | MSE | NLL | MSE | NLL |
> | MAML | 0.0067 | - | 0.0094 | - | 0.0705 | - | 0.3041 | - | 0.2987 | - | 0.0106 | - | 0.0106 | - |
> | Reptile | 0.0060 | - | 0.0078 | - | 0.0691 | - | 0.3160 | - | 0.3047 | - | 0.0096 | - | 0.0096 | - |
> | STP | 0.0046 | -1.1514 | 0.0069 | -1.0390 | 0.0632 | 0.1273 | 0.2607 | 1.1242 | 0.2631 | 0.8563 | 0.0086 | -0.9815 | 0.0086 | -0.9815 |
> | STP+JTP | 0.0045 | -1.1703 | 0.0068 | -1.0681 | 0.0607 | 0.0169 | 0.2348 | 0.6792 | 0.2376 | 0.6812 | 0.0084 | -0.9946 | 0.0084 | -0.9946 |
> | MTP | **0.0037** | **-1.1832** | **0.0054** | **-1.1049** | **0.0546** | **-0.1006** | **0.2276** | **0.6557** | **0.2215** | **0.6660** | **0.0073** | **-1.0331** | **0.0073** | **-1.0331** |
>
>
> We observe that when the data is highly incomplete ($\gamma = 0.5, 0.75$), these two meta-learning baselines perform better than STP and/or JTP in synthetic tasks. This demonstrates that using inter-task correlation through multi-task data is beneficial when the effective number of observable examples is small (either small m or high gamma). Also, unlike JTP, they can inherently handle incomplete data and thus do not have to suffer from noisy imputations. However, they still perform worse than MTP in general and are comparable to STP and JTP when the data is complete ($\gamma = 0$). Also, they generally perform poorly in weather tasks. We noticed that the models fail to capture the global shape of each task, overfitting the context points, which is illustrated in qualitative results in Appendix D.1 of the main paper. We conjecture that the overfitting comes from the lack of global inference on the function space, which is especially important when there exists a large amount of observation noises as in weather data.
>
> We also notice that the gradient-based meta-learning approaches are much inefficient in the meta-testing time compared to NP baselines, because they require several gradient steps for adaptation. For example, in our synthetic experiment, they required 1000 backward steps (with minibatch-size 4) per set of tasks to converge, while NP baselines require only a single forward step per set of tasks. NPs and MTPs are also more reliable as there are no optimization hyperparameters (such as fine-tuning learning rates or steps) in their adaptation.
>
> [b] Finn et al. Model-agnostic meta-learning for fast adaptation of deep networks. In ICML, 2017.
> [c] Nichol et al. On first-order meta-learning algorithms. arXiv preprint arXiv:1803.02999 (2018).
>
> &nbsp;
>
> ---

---

> > ### Comment · Reviewer_ACBa · 2021-11-30
> > **Reply**
> >
> > I thank the authors for their response, and most of my concerns are addressed by the rebuttal. After reading the other reviews and the corresponding responses, I share the same impression with Reviewer eyrZ that several techniques used in the model lack motivations, e.g., attentive neural processes, learnable task embedding, and parameter sharing. Particularly, the simpler option like one-hot vector acting as task embedding also shows comparable even better performance. In my view, the story of this paper should start from multi-task learning. Thus, the main motivation should focus on addressing the challenges of multi-task learning. Obviously, the authors ignore this and just use (attentive) neural processes for multi-task learning.
> >
> > In addition, I agree with the comments of renaming from the Reviewer F6YH. The title of this paper multi-task processes is too generic. Several multi-task learning papers use stochastic processes, especially multi-task Gaussian processes. If readers have a limited background in multi-task learning, the title is misleading.

---

> ### Author Response · Authors · 2021-11-23
> **Rebuttal from Authors of Paper952 to Reviewer ACBa (3/4)**
>
> ---
>
> > **Q3.** In Figure3, why we cannot observe a similar conclusion as NPs/ ANPs/DSVNP, the lower variance around the context points?
>
> **A3.** The variance near the context points is indeed lower than the others, but the presentation in Figure 3 was not clear enough to show them. We changed the presentation of Figure 3 and summarize the changes below.
>
> We first would like to mention that, as mentioned in the paper, we largely base the implementation of STP/JTP/MTP on that of ANP, where its decoder is set to have a fixed minimum standard deviation of 0.1. Due to the inheritance, the minimum predictive standard deviation of STP/JTP/MTP is 0.1. In fact, we find that predicted standard deviations at context points are 0.1 in almost all cases. To confirm this, please refer to Figure 12 where we also render the predictive standard deviations only.
>
> That said, we also observe visualization issues in the original figure where the predictive standard deviations are depicted wrongly due to the low aspect ratio of the figures compared to axes scales. That is, the standard deviation (≈0.1) at a point on the relatively flat section of the function is shown to be high compared to the one at a point with high function curvature.
>
> Due to the potentially misleading aspects, we updated the contents of the figure by visualizing the model uncertainties rather than observation noises of the models. That is, we plot posterior predictive samples (lighter colors) and their mean (darker color) for each task and model (see Appendix C.2 for the detailed sampling method). As can be seen in the updated figure, we observe that overall uncertainties modeled by MTP are more accurate than the baselines, where some of the predictive samples on Tanh, Sigmoid, Gaussian functions are even almost matched to the true function.
>
> &nbsp;
>
> ---
>
> > **Q4.** What’s the functional correlation captured by a global variable z for multi-task learning?
>
> **A4.** One task is correlated with another by the amount of information shared by the two. For example, in our 1D synthetic function regression setting, the global parameters $a, b, c, w$ govern and correlate the functions. In the 2D image completion scenario, Segment and PNCC are assumed to show high correlation since the two render 2D and 3D positions of facial landmarks respectively, whereas RGB and Edge are correlated by the information like color intensity and the gradients of an image.
>
> In our paper, the ability of MTP to capture such functional correlation and to utilize it for solving multiple tasks is demonstrated in a variety of settings. For example, the results on the 1-D synthetic case in Figure (3)-b demonstrate that introducing the global latent variable z is beneficial especially when tasks are correlated (i.e., MTP and MTP-G outperform MTP-T). In addition, the results on the weather data in Figure 4 show that observing additional information from one task (Cloud) greatly improves the predictive distribution of another task (Precip). Finally, the results in the 2D image completion scenario in Table 4 further demonstrate that MTP can indeed discover the two groups of highly correlated 2D tasks (RGB-Edge & Segment-PNCC) and exploit the captured functional correlations to improve the predictive distribution.
>
> &nbsp;
>
> ---

---

> ### Author Response · Authors · 2021-11-23
> **Rebuttal from Authors of Paper952 to Reviewer ACBa (2/4)**
>
> ---
>
> > **Q2.** It’s unclear to me about the motivation of replacing the average pooling operation in the stochastic path with a Pooling by Multihead Attention (PMA) layer. Could the author provide more information about the seed vector in the PMA? Is it also task-specific? What’s the advantage of the parametric pooling over the non-parametric pooling?
>
> **A2.** PMA is a parametric pooling method over an input set of elements. In particular, PMA introduces a set of $k$ learnable $d$-dimensional seed vectors $S \in \mathbb{R}^{k \times d}$ to extract $k$ set-level representations over an input set of $n$ features $Z \in \mathbb{R}^{n \times d}$ via the multi-head attention $\textit{Attn}(Q, K, V)$ where $Q = S$ and $K = V = Z$. This kind of learnable parametric pooling can be beneficial especially when an influence of each set element on a set-level task is not necessarily equal since the seed vectors can learn to discover more salient elements over another and make them have more influence on the set-level representations. Thanks to this learnable parametric approach, PMA has proven its effectiveness on a wide variety of set-level tasks (e.g. maximum value regression, amortized clustering, and point cloud classification), outperforming widely used non-parametric pooling methods (e.g. mean, sum, and max). Note that the number of seed vectors $k$ is chosen depending on the set-level tasks, e.g. $k=1$ in maximum value regression or $k=c$ in clustering and classification where $c$ is the number of clusters and classes, respectively.
>
> Our motivation for adopting PMA as the pooling operation is quite similar to that of the self-attention mechanism: we don’t want our model to ignore the granularity of informativeness in a set of representations and treat each source of information equally via the average pooling. In our implementation, we use PMA in both the intra-task and inter-task inference stages and set $k=1$ since we want to obtain a single set-level representation over per-task observations and inter-task representations. Also, in the case of using a single shared encoder for all tasks as in 1D synthetic and weather datasets, PMA is also shared across tasks both in each of the intra-task and inter-task inference stages but not between the stages.
>
> To investigate the effectiveness of PMA, we conduct an ablation study and compare MTPs trained with or without self-attention followed by PMA or average pooling as follows:
>
> (1) (MTP-SP) MTP with self-attention followed by PMA.
> (2) (MTP-SA) MTP with self-attention followed by average pooling.
> (3) (MTP-P) MTP without self-attention followed by PMA.
> (4) (MTP-A) MTP without self-attention followed by average pooling.
>
> Note that we consistently use MTP-SP for all experiments in Section 5.
> Results are summarized in Table 29, 30 in Appendix H.1 of the main paper and are also summarized below.
>
> [Results on synthetic tasks, $\gamma=0.5$.]
>
> | Tasks |  | Sine |  |  | Tanh |  |  | Sigmoid |  |  | Gaussian |  |
> |:---:|:---:|:---:|:---:|:---:|:---:|:---:|:---:|:---:|:---:|:---:|:---:|:---:|
> | m | 5 | 10 | 20 | 5 | 10 | 20 | 5 | 10 | 20 | 5 | 10 | 20 |
> | MTP-A | 0.4157 | 0.2855 | 0.1913 | 0.1121 | 0.0549 | 0.0320 | 0.0158 | 0.0082 | 0.0050 | 0.0711 | 0.0454 | 0.0321 |
> | MTP-P | 0.3568 | 0.2090 | 0.1185 | 0.0780 | 0.0296 | 0.0153 | 0.0119 | 0.0052 | 0.0033 | 0.0519 | 0.0270 | 0.0161 |
> | MTP-SA | 0.2668 | 0.1299 | 0.0543 | 0.0452 | **0.0103** | **0.0033** | 0.0071 | 0.0015 | **0.0005** | **0.0345** | **0.0132** | **0.0065** |
> | MTP-SP | **0.2636** | **0.1137** | **0.0485** | **0.0435** | 0.0115 | 0.0040 | **0.0066** | **0.0014** | 0.0006 | 0.0360 | **0.0132** | 0.0069 |
>
>
> [Results on weather tasks, $\gamma=0.5$ and $m=10$.]
>
> | Tasks | TempMin |  | TempMax |  | Humidity |  | Precip |  | Cloud |  | Dew |  |
> |---:|---:|---:|---:|---:|---:|---:|---:|---:|---:|---:|---:|---:|
> | metric | MSE | NLL | MSE | NLL | MSE | NLL | MSE | NLL | MSE | NLL | MSE | NLL |
> | MTP-A | 0.0067 | -1.0773 | 0.0119 | -0.8658 | 0.0745 | -0.0471 | 0.2514 | 0.6655 | 0.2578 | 0.7186 | 0.0114 | -0.9103 |
> | MTP-P | 0.0046 | -1.1614 | 0.0071 | -1.0166 | 0.0581 | -0.1463 | 0.2400 | **0.6205** | 0.2360 | **0.6532** | 0.0083 | -1.0183 |
> | MTP-SA | 0.0039 | **-1.1881** | 0.0058 | -1.0990 | **0.0542** | **-0.2083** | **0.2262** | 0.6270 | 0.2242 | 0.6825 | 0.0074 | **-1.0409** |
> | MTP-SP | **0.0037** | -1.1832 | **0.0054** | **-1.1049** | 0.0546 | -0.1006 | 0.2276 | 0.6557 | **0.2215** | 0.6660 | **0.0073** | -1.0331 |
>
>
> As shown in the results, we observe that replacing PMA with average pooling does not show dramatic improvement when trained with self-attention (MTP-SP vs MTP-SA), while it greatly improves the overall performances when trained without the self-attention (MTP-P vs MTP-A). These results imply that in the absence of self-attention, the seed vector of PMA plays a similar role to self-attention layers by assigning different weights to each task.
>
> &nbsp;
>
> ---

---

> ### Author Response · Authors · 2021-11-23
> **Rebuttal from Authors of Paper952 to Reviewer ACBa (1/4)**
>
> ---
>
> > **Q1.** In the implementation of the latent encoder, what’s the role of self-attentions before the pool operations? Besides MSE, could the author provide some visualization results to demonstrate the difference between latent encoder with self-attention and without self-attention?
>
> **A1.** We thank the insightful comments. Before explaining in detail the role of self-attentions in our MTP, let us first present quantitative comparison results between MTP trained with and without self-attention modules. For the variant without self-attention (MTP w/o Self-Attn), we replace self-attention follwed by pooling with just average pooling. The results are shown in Table 27, 28 in Appendix H.1 of the main paper and are also summarized below.
>
> [Results on synthetic tasks, $\gamma = 0.5$.]
>
> | Tasks |  | Sine |  |  | Tanh |  |  | Sigmoid |  |  | Gaussian |  |
> |:---:|:---:|:---:|:---:|:---:|:---:|:---:|:---:|:---:|:---:|:---:|:---:|:---:|
> | m | 5 | 10 | 20 | 5 | 10 | 20 | 5 | 10 | 20 | 5 | 10 | 20 |
> | MTP w/o Self-Attn | 0.4157 | 0.2855 | 0.1913 | 0.1121 | 0.0549 | 0.0320 | 0.0158 | 0.0082 | 0.0050 | 0.0711 | 0.0454 | 0.0321 |
> | MTP | **0.2636** | **0.1137** | **0.0485** | **0.0435** | **0.0115** | **0.0040** | **0.0066** | **0.0014** | **0.0006** | **0.0360** | **0.0132** | **0.0069** |
>
>
> [Results on weather tasks, $\gamma = 0.5$ and $m = 10$.]
>
> | Tasks | TempMin |  | TempMax |  | Humidity |  | Precip |  | Cloud |  | Dew |  |
> |---:|---:|---:|---:|---:|---:|---:|---:|---:|---:|---:|---:|---:|
> | metric | MSE | NLL | MSE | NLL | MSE | NLL | MSE | NLL | MSE | NLL | MSE | NLL |
> | MTP w/o Self-Attn | 0.0067 | -1.0773 | 0.0119 | -0.8658 | 0.0745 | -0.0471 | 0.2514 | 0.6655 | 0.2578 | 0.7186 | 0.0114 | -0.9103 |
> | MTP | **0.0037** | **-1.1832** | **0.0054** | **-1.1049** | **0.0546** | **-0.1006** | **0.2276** | **0.6557** | **0.2215** | **0.6660** | **0.0073** | **-1.0331** |
>
>
> As shown in the result, MTP with self-attention outperforms the one without self-attention by a large margin, implying that self-attention is critical in MTP.
>
> To further analyze the role of the self-attention, we would like to remind the fact that the task-agnostic global knowledge is inferred by the latent encoder through the following steps:
> (1) per-task knowledge extraction by aggregating intra-task observations.
> (2) global knowledge extraction by aggregating inter-task representations.
>
> And we find that the self-attention mechanism is playing a critical role in maximizing the informativeness of the extracted knowledge in each step by learning to take more information from one observation/representation over another via the attention mechanism.
>
> Specifically, when inferring per-task knowledge from intra-task observations (1), it is beneficial for the model to recognize and exploit the fact that not all context points could have the same information for characterizing the task. For example, in the 1-D synthetic case, some context points at or near the middle of the mode would be more informative to characterize the gaussian function compared to the ones observed in the tails of the curve. Similarly, when boiling down per-task representations into a single task-agnostic global knowledge (2), it is often desirable to attend one representation over another since the informativeness of the per-task representations could vary depending on the size and quality of the contexts observed in each task. In summary, self-attention in both per-task and across-task inference paths improves the expressive power of the model.
>
> To investigate whether the self-attention modules operate as desired in both stages, we conduct additional experiments on 1-D synthetic curve regression setting. In both cases, we examine whether the self-attention can recognize the “good” observations/representations over another and let them influence the aggregated representations more by inspecting its attention weights.
>
> To see the first case (1), for each task, we compute an averaged attention weight placed on a set of context points by averaging dimensions of the attention map of shape $(nL, nH, nQ, nK)$ to $(1, 1, 1, nK)$ where $nL$ and $nH$ denote the number of self-attention layers and heads in each layer, and $nQ=nK$ refer to the number of query ($Q$) and key ($K$) vectors and $Q=K$. Note that this averaged attention weight can be interpreted as the averaged importance put on each of the context points by the self-attention module. Results are visualized in Figure 17 in Appendix H.1 of the revised version. As shown in the figure, points located at representative positions or at sparse regions are attended more than others. This shows that self-attention in per-task paths operates as desired.

---

> > ### Author Response · Authors · 2021-11-23
> > **Rebuttal from Authors of Paper952 to Reviewer ACBa (1/4) (continued)**
> >
> > To inspect the second case (2), we opt to measure the degree of alignment between the informativeness of per-task representations and the weighted importance over them by the attention module. In this experiment, we control the informativeness of the per-task representations by manipulating the number of observable contexts for each task. Specifically, we sample $N=1000$ sets of T multi-task functions where one randomly chosen task could have $nC \in \\{2, 7, 12, 17\\}$ context points while the other three tasks could only have one context point respectively. Following [a], we then measure the correlation $\rho(I_\text{data}, I_\text{attn}) \in [0,1]$ between the context size $I_\text{data}$ and the weighted importance $I_\text{attn}$,
> > where each $(i,j)$-th entry of $I_\text{data} \in \mathbb{R}^{N \times T}$ denotes the number of contexts for $j$-th task in $i$-th set and each $(i,j)$-th entry of $I_\text{attn} \in \mathbb{R}^{N \times T}$ is obtained by averaging attention weights assigned on $j$-th task in $i$-th set along the Query axis, different heads, and different layers. Also, we compare the measured correlation across MTPs trained under the settings with three different levels of inter-task correlation, namely totally-correlated, partially-correlated, and independent settings as in Section 5.1. The results are summarized below.
> >
> > | Independent Tasks | Partially Correlated Tasks | Totally Correlated Tasks |
> > |:---:|:---:|:---:|
> > | 0.2709 | 0.4616 | 0.5078 |
> >
> > As shown in the table, MTPs trained with totally and partially correlated tasks show the most and second-highest correlations, achieving 0.51 and 0.46, respectively, while the MTP trained with independent tasks show marginal correlations as 0.27. The result demonstrates that the attention modules in across-task path assign higher weights to more informative tasks that consist of abundant global knowledge (*e.g.*, when tasks are highly correlated), which is a desired behavior.
> >
> > [a] Morcos et. al., Insights on representational similarity in neural networks with canonical correlation, In NeurIPS, 2018.
> >
> > &nbsp;
> >
> > ---

---

### Official Review · Reviewer_eyrZ · 2021-10-31

**Correctness:** 3
**Technical Novelty And Significance:** 2
**Empirical Novelty And Significance:** 3
**Recommendation:** 5
**Confidence:** 5

**Main Review:**

Strengths:

(1) In the theoretical aspect, it is interesting the hierarchical architecture to incorporate both task-specific and task-agnostic functional representations.

(2) Experiments on the real-world tasks demonstrate the advantages of the proposed methods in a practical.

(3) This paper is well written and easy to follow.

Weaknesses:

(1) In this paper, the motivation for introducing NPs to multi-task learning is not clear. Why it’s necessary to design the model based on NPs, especially ANPs?

(2) The paper lacks important baselines.

-The proposed method applies the ANPs as the backbone, which has the deterministic encoder and the latent encoder.
MTP/STP/JTP has a totally different module for the deterministic encoder. To show the effectiveness of the designed hierarchical latent model in the latent encoder, it’s necessary to compare with a) MTP/STP/JTP without deterministic encoder and b) MTP without latent encoder.

-For MTP, the hidden layers of the encoder and the decoder are shared by all tasks; For STP, the hidden layers of encoders and decoders are task-specific without any shared mechanism. To investigate the models’ ability of incorporating correlation across tasks, it would be convincing to have a comparison with such baselines: a) STP with the shared encoder and decoder for all tasks and b) MTP with the task-specific hidden layers/ deterministic encoder/decoder.

(3) What is the motivation of introducing the learnable task-embedding \exp^t? It seems to add task-level information for each sample. Could the authors visualize the difference between the learned task-embeddings of different tasks? Is it possible to utilize a fixed one-hot vector with the corresponding task label?

(4) Why add a learnable task embedding \exp^t for MTP but not for STP or JTP? It does not seem to be a fair comparison.

(5) During training, the prior in the ELBO is approximated by the posterior conditioned on context samples. There are no theoretical guarantees on the bound defined by the approximation prior. This bound could not be tight enough to be practical in practice.

(6) In the experiments, are results of all tasks obtained simultaneously? It would be better to add the average results to show the overall performance of multiple tasks.


**Summary Of The Paper:**

This paper introduced multi-task processes (MTPs), a new variant of neural processes, which jointly infers multiple heterogeneous functions to given possibly incomplete data. The authors conducted thorough experiments on several tasks to demonstrate the effectiveness of the proposed MTPs.

**Summary Of The Review:**

My main concerns are about the motivation of introducing NPs for multi-task learning, the lack of important baselines, and the tightness of the bound.

---

> ### Author Response · Authors · 2021-11-23
> **Rebuttal from Authors of Paper952 to Reviewer eyrZ (5/5)**
>
> ---
>
> **Q6.** In the experiments, are results of all tasks obtained simultaneously? It would be better to add the average results to show the overall performance of multiple tasks.
>
> **A6.** Yes. In our multi-task setting, all models including the single-task baseline (STP) predict outputs for all tasks simultaneously. However, reporting the average performance of all tasks could be misleading since the scale of MSEs and NLLs is different across tasks (*e.g.*, synthetic and weather datasets) and sometimes not comparable (*e.g.*, continuous and categorical outputs in 2D dataset). Also, naively averaging the results can lead to either underestimation of easier tasks or overestimation of harder tasks. Therefore, we report individual performance for each task.
>
> &nbsp;
>
> ---

---

> ### Author Response · Authors · 2021-11-23
> **Rebuttal from Authors of Paper952 to Reviewer eyrZ (4/5)**
>
> ---
>
> > **Q3-2.** Is it possible to utilize a fixed one-hot vector with the corresponding task label? Could the authors visualize the difference between the learned task-embeddings of different tasks?
>
> **A3-2.** We consider the type of task embedding as a design choice and we simply adopt learnable task embedding since it is widely used with self-attention mechanisms. On top of that, we can utilize a fixed one-hot vector for inducing task information in MTP as suggested by the reviewer. To investigate the effect of a specific choice of task embedding, we compare the learnable task embeddings with the simple one-hot encoding scheme on synthetic and weather datasets. We include the quantitative results in Table 35, 36 in Appendix H.4 of the revised version and also summarize below.
>
> [Results on synthetic tasks, $\gamma = 0.5$.]
>
> | Tasks |  | Sine |  |  | Tanh |  |  | Sigmoid |  |  | Gaussian |  |
> |:---:|:---:|:---:|:---:|:---:|:---:|:---:|:---:|:---:|:---:|:---:|:---:|:---:|
> | m | 5 | 10 | 20 | 5 | 10 | 20 | 5 | 10 | 20 | 5 | 10 | 20 |
> | MTP-Onehot | **0.2507** | **0.1047** | **0.0426** | 0.0440 | **0.0110** | **0.0036** | 0.0067 | **0.0014** | **0.0005** | **0.0338** | **0.0130** | **0.0062** |
> | MTP-Learnable | 0.2636 | 0.1137 | 0.0485 | **0.0435** | 0.0115 | 0.0040 | **0.0066** | **0.0014** | 0.0006 | 0.0360 | 0.0132 | 0.0069 |
>
>
> [Results on weather tasks, $\gamma = 0.5$ and $m = 10$.]
>
> | Tasks | TempMin |  | TempMax |  | Humidity |  | Precip |  | Cloud |  | Dew |  |
> |---:|---:|---:|---:|---:|---:|---:|---:|---:|---:|---:|---:|---:|
> | metric | MSE | NLL | MSE | NLL | MSE | NLL | MSE | NLL | MSE | NLL | MSE | NLL |
> | MTP-Onehot | **0.0036** | **-1.1975** | 0.0055 | -1.1046 | **0.0526** | -0.0835 | **0.2265** | 0.7003 | 0.2244 | 0.6909 | **0.0070** | **-1.0479** |
> | MTP-Learnable | 0.0037 | -1.1832 | **0.0054** | **-1.1049** | 0.0546 | **-0.1006** | 0.2276 | **0.6557** | **0.2215** | **0.6660** | 0.0073 | -1.0331 |
>
>
> We observe that MTP with one-hot embedding is comparable to MTP with learnable embedding. To further investigate the effect of learnable embedding, we visualize the learned task embedding by MTP using the t-SNE algorithm. We include the visualization results in Figures 18, 19 in Appendix H.4. As shown in the figures, we find that the learned task embeddings are well-separated from each other and uniformly distributed on the embedding space. From the observations, we conjecture that well-separated task embeddings are sufficient task information for MTP, while complex correlation across tasks are captured by the shared latent variable $z$. It also explains why one-hot encoding also performs descent as task embedding. Please note that one-hot encoding followed by a linear layer functions exactly the same as the learnable embedding.
>
> &nbsp;
>
> ---
>
> > **Q5.**  During training, the prior in the ELBO is approximated by the posterior conditioned on context samples. There are no theoretical guarantees on the bound defined by the approximation prior. This bound could not be tight enough to be practical in practice.
>
> **A5.** We first acknowledge that the terminology used in Section 2.1, Appendix A.3 and A.4 was confusing. We have noticed that the approximation of prior by variational posterior can be seen as parameter sharing between prior and posterior networks [b]. In this perspective, there is no “approximation” of the prior by the posterior but an inductive bias between the prior model and variational posterior. Thus the objective can be indeed as much tight as typical ELBO, which aligns with the practical success of the NP family and MTPs. Note that the inductive bias of parameter sharing also induces the desired regularizer: alignment between the summary of context and target, as explained in Section 2.2 and 3.2. For a detailed justification, we refer the reviewer to the discussions in the response **A3** of Reviewer F6YH.
>
> We revised Section 2.1 and Appendix A.3-4 according to the discussions above, which makes the explanation of ELBO objectives clearer. We hope this handles the reviewer’s concern and are happy to elaborate more on any remaining concerns.
>
> [b] ​​Gordon, Jonathan. Advances in Probabilistic Meta-Learning and the Neural Process Family. Diss. University of Cambridge, 2021.
>
> &nbsp;
>
> ---

---

> ### Author Response · Authors · 2021-11-23
> **Rebuttal from Authors of Paper952 to Reviewer eyrZ (3/5)**
>
> ---
>
> > **Q2.** The paper lacks important baselines.
>
> >> **Q2-2.** For MTP, the hidden layers of the encoder and the decoder are shared by all tasks; For STP, the hidden layers of encoders and decoders are task-specific without any shared mechanism. To investigate the models’ ability of incorporating correlation across tasks, it would be convincing to have a comparison with such baselines: a) STP with the shared encoder and decoder for all tasks and b) MTP with the task-specific hidden layers/ deterministic encoder/decoder.
>
> > **Q3-1.** What is the motivation of introducing the learnable task-embedding $\exp^t$?
>
> > **Q4.** Why add a learnable task embedding \exp^t for MTP but not for STP or JTP? It does not seem to be a fair comparison.
>
> **A2-2 & A3-1 & A4**. We appreciate the reviewer’s suggestion. To share the encoder parameters across tasks while learning the task-specific embedding, it is necessary to be able to modulate the encoder parameters conditioned on the task. This is natural in MTP since its functional form is defined over the product space $\mathcal{X} \times \mathcal{T}$ of the input data space $\mathcal{X}$ and the task index space $\mathcal{T}$, while it is not straightforward in STP and JTP as their input space is defined only in the data space $\mathcal{X}$. For this reason, we only applied parameter sharing with task embedding in MTP. That said, we agree with the reviewer that it is fair to inspect the effect of parameter sharing to better examine the models’ ability to capture functional correlations. Therefore, we conducted an ablation study in 1D synthetic and weather data with the following four baselines as suggested by the reviewer:
>
> (1) STP-S: STP with shared encoder and decoder. In this model, we additionally introduce learnable task embeddings as in MTP so that the model can learn task-specific embeddings. In particular, per-task hidden representations $s^t_i = \psi_s(x_i,y^t_i)$ are added by learnable task embeddings $e^t$ before being fed into the intra-task self-attention layers.
>
> (2) STP-TS: STP with task-specific encoders and decoders. (used in the experiments in Section 5)
>
> (3) MTP-TS: MTP with task-specific encoders for per-task inference path and decoders. (used in the experiments in Section 5.3)
>
> (4) MTP-S: MTP with shared encoder for per-task inference path and decoder. (used in the experiments in Section 5.1, 5.2)
>
> Note that we consistently used STP-TS and MTP-S for 1D experiments, and MTP-TS for 2D experiments.
> The results are included in Table 29, 30 in Appendix H.2 of the main paper and also summarized below.

---

> > ### Author Response · Authors · 2021-11-23
> > **Rebuttal from Authors of Paper952 to Reviewer eyrZ (3/5) (continued)**
> >
> > [Results on synthetic tasks, $\gamma = 0.5$.]
> >
> > | Tasks |  | Sine |  |  | Tanh |  |  | Sigmoid |  |  | Gaussian |  |
> > |:---:|:---:|:---:|:---:|:---:|:---:|:---:|:---:|:---:|:---:|:---:|:---:|:---:|
> > | m | 5 | 10 | 20 | 5 | 10 | 20 | 5 | 10 | 20 | 5 | 10 | 20 |
> > | STP-S | 0.5387 | 0.2596 | 0.0845 | 0.1255 | 0.0459 | 0.0123 | 0.0177 | 0.0050 | 0.0012 | 0.0801 | 0.0371 | 0.0181 |
> > | STP-TS | 0.5212 | 0.2609 | 0.0993 | 0.1307 | 0.0468 | 0.0159 | 0.0203 | 0.0067 | 0.0025 | 0.0799 | 0.0409 | 0.0222 |
> > | MTP-S | **0.2636** | **0.1137** | **0.0485** | **0.0435** | **0.0115** | **0.0040** | **0.0066** | **0.0014** | **0.0006** | **0.0360** | **0.0132** | **0.0069** |
> > | MTP-TS | 0.2901 | 0.1297 | 0.0532 | 0.0618 | 0.0172 | 0.0063 | 0.0093 | 0.0031 | 0.0014 | 0.0426 | 0.0173 | 0.0100 |
> >
> > [Results on weather tasks, $\gamma = 0.5$ and $m = 10$.]
> >
> > | Tasks | TempMin |  | TempMax |  | Humidity |  | Precip |  | Cloud |  | Dew |  |
> > |---:|---:|---:|---:|---:|---:|---:|---:|---:|---:|---:|---:|---:|
> > | metric | MSE | NLL | MSE | NLL | MSE | NLL | MSE | NLL | MSE | NLL | MSE | NLL |
> > | STP-S | 0.0049 | -1.1381 | 0.0066 | -1.0527 | 0.0621 | 0.0240 | 0.2675 | 0.9537 | 0.2629 | 0.7847 | 0.0084 | -0.9877 |
> > | STP-TS | 0.0046 | -1.1514 | 0.0069 | -1.0390 | 0.0632 | 0.1273 | 0.2607 | 1.1242 | 0.2631 | 0.8563 | 0.0086 | -0.9815 |
> > | MTP-S | 0.0037 | **-1.1832** | 0.0054 | **-1.1049** | 0.0546 | **-0.1006** | 0.2276 | **0.6557** | **0.2215** | **0.6660** | 0.0073 | **-1.0331** |
> > | MTP-TS | **0.0036** | -1.1818 | **0.0053** | -1.0885 | **0.0519** | -0.0662 | **0.2187** | 0.7213 | 0.2253 | 0.6663 | **0.0071** | -1.0232 |
> >
> > As can be seen in the tables, in the weather dataset, we observe that the models with parameter sharing (STP-S & MTP-S) show comparable performances to their respective non-sharing baselines (STP-TS & MTP-TS). On the other hand, the parameter sharing technique slightly improves the performances of the models in the 1-D synthetic case. We conjecture that the utilization of the same architecture and its parameter for all tasks acts as a good inductive bias to the models considering that all tasks share the same global parameter a,b,c,w. Nonetheless, we still find that MTPs consistently outperform STPs regardless of whether the parameters are shared or not, validating the effectiveness of MTP to capture and exploit functional correlation for multi-task learning problems.
> >
> > &nbsp;
> >
> > ---

---

> ### Author Response · Authors · 2021-11-23
> **Rebuttal from Authors of Paper952 to Reviewer eyrZ (2/5)**
>
> ---
>
> > **Q2.** The paper lacks important baselines.
>
> >> **Q2-1.** The proposed method applies the ANPs as the backbone, which has the deterministic encoder and the latent encoder. MTP/STP/JTP has a totally different module for the deterministic encoder. To show the effectiveness of the designed hierarchical latent model in the latent encoder, it’s necessary to compare with a) MTP/STP/JTP without deterministic encoder and b) MTP without latent encoder.
>
> **A2-1a.** We appreciate the reviewer’s comments and suggestions. First, we would like to clarify that the design of the deterministic encoder is *dependent* on LVM. ​The deterministic encoder produces an additional representation from the context that is passed to the decoder along with a latent variable and a target input, serving as a deterministic counterpart of the latent variable. Therefore, the deterministic representation should not break the dependency between the context, the latent variable, and the target input, since it will change the underlying graphical model otherwise. It makes STP, JTP, and MTP have different deterministic encoder structures to reflect their respective LVMs. Specifically, the deterministic representations of STP should be encoded independently for each task for per-task latent variables, while the one in JTP is obtained globally over the entire context $C$ for the single global latent variable. On the other hand, MTP’s decoder takes per-task latent variables $v^1, \cdots, v^T$ which depend on the whole context $C$. Thus we employ across-task self-attention on the outputs of per-task cross-attention to give inter-task dependency to the deterministic representations for $v^1, \cdots, v^T$.
>
> Due to such inherent differences in deterministic encoder, it is difficult to isolate the sole effectiveness of STP, JTP, and MTP in their formulation from the architectural differences. To get rid of the effect of a deterministic encoder, we can compare STP/JTP/MTP without the deterministic encoder. Note that these variants correspond to the direct NP implementations of STP/JTP/MTP rather than ANP. The results are included in Tables 31, 32 in Appendix H.3 of the main paper and are also summarized below.
>
> [Results on synthetic tasks, $\gamma = 0.5$.]
>
> | Tasks |  | Sine |  |  | Tanh |  |  | Sigmoid |  |  | Gaussian |  |
> |:---:|:---:|:---:|:---:|:---:|:---:|:---:|:---:|:---:|:---:|:---:|:---:|:---:|
> | m | 5 | 10 | 20 | 5 | 10 | 20 | 5 | 10 | 20 | 5 | 10 | 20 |
> | STP-L | 0.5615 | 0.3006 | 0.1254 | 0.1362 | 0.0585 | 0.0234 | 0.0213 | 0.0086 | 0.0045 | 0.0809 | 0.0405 | 0.0234 |
> | STP-L + JTP-L | 0.4151 | 0.2560 | 0.1395 | 0.1055 | 0.0506 | 0.0224 | 0.0201 | 0.0106 | 0.0061 | 0.0687 | 0.0376 | 0.0227 |
> | MTP-L | **0.3003** | **0.1543** | **0.0755** | **0.0563** | **0.0162** | **0.0067** | **0.0096** | **0.0027** | **0.0012** | **0.0417** | **0.0169** | **0.0091** |
>
>
> [Results on weather tasks, $\gamma = 0.5$ and $m = 10$.]
>
> | Tasks | TempMin |  | TempMax |  | Humidity |  | Precip |  | Cloud |  | Dew |  |
> |---:|---:|---:|---:|---:|---:|---:|---:|---:|---:|---:|---:|---:|
> | metric | MSE | NLL | MSE | NLL | MSE | NLL | MSE | NLL | MSE | NLL | MSE | NLL |
> | STP-L | 0.0065 | -1.0783 | 0.0102 | -0.9202 | 0.0828 | 0.0341 | 0.3381 | 0.8792 | 0.3411 | 0.8798 | 0.0106 | -0.8804 |
> | STP-L + JTP-L | 0.0085 | -1.0632 | 0.0136 | -0.8601 | 0.0966 | 0.0737 | 0.2832 | 0.7347 | 0.2851 | 0.7692 | 0.0127 | -0.8827 |
> | MTP-L | **0.0046** | **-1.1580** | **0.0072** | **-1.0203** | **0.0596** | **-0.1554** | **0.2432** | **0.6267** | **0.2372** | **0.6659** | **0.0085** | **-1.0006** |
>
>
> The overall trends are the same as the main table in Section 5, which demonstrates that the effectiveness of MTP does not depend on a specific choice of architecture (NP or ANP).
>
>
> In addition, note that STP, JTP, and MTP not only have different LVMs but also different functional forms (i.e., the function space underlying the stochastic processes). For comparisons on LVMs with fixed functional form, we respectively remind the reviewer that the ablation study in Section 5.1 (Figure 3(b)) presents this comparison. Specifically, it compares the non-hierarchical variants of MTP, MTP-T, and MTP-G, which correspond to MTP with task-specific latents only and global latent only, respectively, all of which share the same functional form with MTPs. The result and discussion in Section 5.1 show that introducing both task-specific latents and global latent is the most effective for capturing various degrees of inter-task correlation present in the data, which justifies the hierarchical LVM formulation of MTP.
>
> &nbsp;
>
> ---

---

> > ### Author Response · Authors · 2021-11-23
> > **Rebuttal from Authors of Paper952 to Reviewer eyrZ (2/5) (continued)**
> >
> > **A2-1.b** We would like to emphasize that our main focus is to extend NPs to multi-task settings by a probabilistic model so that it inherits advantages of NPs such as generative modeling and uncertainty estimation. The lack of generative modeling could be problematic when the relationship between tasks is non-deterministic.
> >
> > Nevertheless, we still can compare MTP without latent encoder (MTP-D) as an additional baseline. To demonstrate how the probabilistic model can be effective compared to the deterministic model, we also evaluated MTP using its best sample among 25 posterior predictive samples (MTP-best). The result is included in Table 33, 34 in Appendix H.3 of the revised version and also summarized below.
> >
> > [Results on synthetic tasks, $\gamma = 0.5$.]
> >
> > | Tasks |  | Sine |  |  | Tanh |  |  | Sigmoid |  |  | Gaussian |  |
> > |:---:|:---:|:---:|:---:|:---:|:---:|:---:|:---:|:---:|:---:|:---:|:---:|:---:|
> > | m | 5 | 10 | 20 | 5 | 10 | 20 | 5 | 10 | 20 | 5 | 10 | 20 |
> > | MTP-D | 0.1963 | 0.0985 | 0.0447 | 0.0423 | 0.0110 | 0.0042 | 0.0073 | 0.0015 | 0.0006 | 0.0281 | 0.0127 | 0.0073 |
> > | MTP | 0.2636 | 0.1137 | 0.0485 | 0.0435 | 0.0115 | 0.0040 | 0.0066 | 0.0014 | 0.0006 | 0.0360 | 0.0132 | 0.0069 |
> > | MTP-best | **0.1239** | **0.0491** | **0.0176** | **0.0169** | **0.0032** | **0.0009** | **0.0025** | **0.0004** | **0.0002** | **0.0146** | **0.0046** | **0.0021** |
> >
> >
> > [Results on weather tasks, $\gamma = 0.5$ and $m = 10$.]
> >
> > | Tasks | TempMin |  | TempMax |  | Humidity |  | Precip |  | Cloud |  | Dew |  |
> > |---:|---:|---:|---:|---:|---:|---:|---:|---:|---:|---:|---:|---:|
> > | metric | MSE | NLL | MSE | NLL | MSE | NLL | MSE | NLL | MSE | NLL | MSE | NLL |
> > | MTP-D | 0.0039 | -1.1892 | 0.0059 | -1.0944 | 0.0535 | -0.2142 | 0.2257 | 0.6034 | 0.2260 | 0.6536 | 0.0077 | -1.0434 |
> > | MTP | 0.0037 | -1.1832 | 0.0054 | -1.1049 | 0.0546 | -0.1006 | 0.2276 | 0.6557 | 0.2215 | 0.6660 | 0.0073 | -1.0331 |
> > | MTP-best | **0.0032** | **-1.2143** | **0.0047** | **-1.1441** | **0.0489** | **-0.2320** | **0.2160** | **0.5491** | **0.2091** | **0.6044** | **0.0065** | **-1.0763** |
> >
> >
> > We observe that MTP and MTP-D are comparable in synthetic and weather datasets, which seems reasonable as we designed the deterministic encoder to mimic the latent encoder of MTP (e.g., they employ both per-task and across-task inferences). However, we can see that MTP-best clearly outperforms MTP-D, which implies that MTP can generate more accurate samples while MTP-D cannot. Please note that removing the latent path from the MTPs induces Conditional Neural Processes (CNPs) of MTPs and is still considered as our model.
> >
> > &nbsp;
> >
> > ---

---

> ### Author Response · Authors · 2021-11-23
> **Rebuttal from Authors of Paper952 to Reviewer eyrZ (1/5)**
>
> ---
>
> > **Q1.** In this paper, the motivation for introducing NPs to multi-task learning is not clear. Why it’s necessary to design the model based on NPs, especially ANPs?
>
> **A1.** NP is a flexible meta-learning method that can adapt to a broad class of functions. As explained in Section 1 in the main paper, it has many advantages such as fast adaptation to an unseen task only with a single forward step of a neural network, the ability to sample multiple functions via generative modeling and to estimate uncertainty. Also, NP can process an arbitrary number of context examples in a single adaptation step since it treats training data as a set.
>
> These advantages can be applied to multi-task learning. For example, when the relationship between tasks is non-deterministic, the ability to handle stochastic correlation among multiple functions or to estimate the uncertainty of the prediction becomes beneficial. Also, the set-encoding and decoding architecture is adequate for handling incomplete multi-task data. For these reasons, it is reasonable to extend NPs to multi-task settings. We chose ANP as a specific architecture among the NP family, as it is known to be robust to overfitting and has high expressive power [a]. Note that our main contribution is to provide a general framework that extends NPs to multi-task settings while the implementation of MTP is not restricted to ANP.
>
> We hope this response clarifies the motivation for introducing (A)NPs in multi-task learning and are happy to elaborate more on any remaining concerns.
>
> [a] Kim et al. Attentive neural processes. In ICLR, 2019.
>
> &nbsp;
>
> ---

---

> ### Comment · Reviewer_eyrZ · 2021-11-29
> **some remaining concerns**
>
> Thanks for the authors’ response. I appreciate their efforts in conducting more experiments.
>
> I still have some concerns:
>
> For A1: The motivation for introducing neural processes in multi-task learning is still not clear. It is not convinging by simply enumeriating the advantages of regular neural processes as the motivations of this work. In multi-task learning, what are the benefits of learning non-deterministic relationships? What are the advantages of neural processes to learn such relationships, compared with other Bayesian MTL models? Such questions are important to explain why the proposed method is not a simple application of neural processes in multi-task learning.
>
> For A2-2: The method should have consistent architecture (shared encoder/decoder or not) in the experiments. The proposed method shares the encoder and decoder for 1-D experiments (MTP-S) but not for 2-D experiments (MTP-TS). As the authors explained, sharing parameters is a good inductive bias for multi-task learning, why not share them for 2-D experiments?
>
> For A5: Why does the inductive bias of parameter sharing between variational posterior and prior leads to the objective as tight as typical ELBO? If it is true, it means the performance of the sharing parameters or not could be the same. It’s better to verify this by experiments.
>
> For A3-2: The visualization seems too perfect. The results show that the learned task-embeddings are uniformly distributed on the embedding space. If the task embeddings are sufficient task information for MTP, “Tanh” should be closer to “Sigmoid” than “Sine” because “Tanh” and “Sigmoid” are similar (as shown in Figure3 (a)).

---

> > ### Author Response · Authors · 2021-11-30
> > **Rebuttal for the remaining concerns (2/2)**
> >
> > > **Q5.** Why does the inductive bias of parameter sharing between variational posterior and prior leads to the objective as tight as typical ELBO?
> >
> > **A5.** We did not intend that “the inductive bias of parameter sharing between variational posterior and prior leads to the objective as tight as typical ELBO”. We respectively believe that the reviewer was confused with our previous response and want to clarify them here.
> >
> > The term *approximated prior* was misused in our previous draft, which was based on alternative interpretation of the ELBO objective with different graphical model. In the graphical model, context and target are not distinguished and the conditional prior $p(z, v^{1:T}|C)$ is induced via Bayes' rule as follows:
> > $$
> > p(z, v^{1:T}|C) = \frac{p(z, v^{1:T}) \prod_{t \in \mathcal{T}} \prod_{i \in \mathcal{I}(C)} p(y_i^t|x_i^t, v^t)}{\int \int p(z, v^{1:T}) \prod_{t \in \mathcal{T}} \prod_{i \in \mathcal{I}(C)} p(y_i^t|x_i^t, v^t) dzdv^{1:T}}.
> > $$
> > As this term is intractable, we have to approximate it with a variational posterior $q(z, v^{1:T}|C)$, which produces approximate ELBO.
> >
> > In contrast, with our conditional graphical model, context and target are distinguished and the prior $p(z, v^{1:T}|C)$ is modeled via a separate inference path, together with the generative model $p(Y_D^{1:T}|X_D^{1:T}, v^{1:T})$. Then the only intractable part is the conditional posterior $p(z,v^{1:T}|C,D)$, and this is approximated by the variational posterior $q(z,v^{1:T}|D)$ as in typical variational inference, *i.e.,* the ELBO is the exact lower bound for $\log p(Y_D^{1:T}|X_D^{1:T},C)$.
> >
> > Then we have two design choices for the prior model $p(z,v^{1:T}|C)$. One option is to employ a separate encoder network, which is adopted in DSVNP [c], and another option is to share the encoder network with the variational posterior $q(z,v^{1:T}|D)$, which is adopted in ANP [a]. We adopted the latter one in this paper, since it is proven to work well in practice.
> >
> > For a more comprehensive discussion about two different interpretations of NP, please refer Section 2.6 of [b].
> >
> > [a] Kim et al. Attentive neural processes. In ICLR, 2019.
> > [b] Gordon, Jonathan. Advances in Probabilistic Meta-Learning and the Neural Process Family. Diss. University of Cambridge, 2021.
> > [c] Wang and Hoof. Doubly stochastic variational inference for neural processes with hierarchical latent variables. In ICML, 2020.
> >
> > ---
> >
> > > **Q3-2.** Regarding task embedding.
> >
> > **A3-2.** In our framework, it is not necessary that the task embedding capture inter-task correlation and it is totally fine that the learned task embeddings are uniformly distributed. This is because the inter-task correlation can be captured throughout the whole across-task paths, which include self-attention layers and PMA pooling layers in both latent encoder and deterministic encoder. In other words, once we have separated task-specific contexts using the uniformly distributed task embeddings, the following attention layers can capture and exploit inter-task correlation to encode the global latent by assigning different weights to the tasks. However, as our model consists of multiple self-attention layers with multiple heads, it is not easy to visualize the task correlation directly from the attention layers. Therefore, we have presented the learned inter-task correlation of MTP by measuring performance gain through knowledge transfer between tasks in Table 4 (Section 5). The result shows that the captured correlation aligns to a rough intuition about the tasks, *i.e.*, tasks encoding high-level information (Segment, PNCC) and low-level information (RGB, Edge) are clustered, respectively.

---

> > ### Author Response · Authors · 2021-11-30
> > **Rebuttal for the remaining concerns (1/2)**
> >
> > > **Q1.** motivation of extending NP for MTL
> >
> > **A1.** We appreciate the comment. Compared to Bayesian MTL methods, the main advantage of extending NP to MTL is that we can “meta-train” the correlation across tasks (e.g., how one task is related to infer another) and exploit this relationship to novel MTL domains (i.e., meta-testing). Especially, combined with our MTP formulation, such ability to meta-train multi-task correlation allows us to apply MTL in more challenging settings where (meta-test) dataset is composed of incomplete task labels: although labels for some tasks are missing in data, MTP can infer the task using the learned correlation with other tasks whose labels are available. This ability to meta-train is a unique advantage of MTP compared to other Bayesian MTL methods.
> >
> > Compared to other meta-learning approaches, MTP can learn stochastic correlation across tasks and perform global inference on the function space. In real-world scenarios, it is common that tasks are not deterministically related. For example in our 2D experiments, observing pixels from Segment does not fully determine RGB image, thus in this case the ability to capture stochastic inter-task relationships is crucial. Without such ability, a model would always produce the mean face conditioned on the segmentation map. Even if the tasks have a deterministic relationship, as in our totally correlated synthetic dataset (first row of Figure 3(b)), it becomes stochastic when only a few context points are given. For example, given points on the flat region from the Gaussian task, it is not easy to exactly specify all the global parameters $a, b, c, w$ (*e.g.*, knowing the mode is necessary to infer the scale $a$), thus it is better to stochastically transfer the global knowledge extracted from the Gaussian task to the others. We have shown the benefits of probabilistic modeling empirically in **A7**, and also by comparison with gradient-based meta-learning methods (MAML, Reptile) in Section 5 of the updated draft. Please also read **A1** to Reviewer F6YH for more comprehensive discussions.
> >
> > ---
> >
> > > **Q2-2.** regarding architecture of MTP
> >
> > **A2-2.** First of all, we would like to emphasize that parameter-sharing on a per-task encoder is an implementational choice and not the main contribution of our work. MTP can always be implemented without parameter sharing, and this does not restrict the ability of MTP to exploit inter-task correlation, as shown in the 2D experiments. Note that the core mechanism that enables MTP to share knowledge across tasks is the hierarchical latent variable model (implemented by the across-task path of Figure 2), not the parameter sharing on per-task encoders.
> >
> > We introduced parameter sharing in 1D experiments because tasks in those experiments are 1D signals that have a homogeneous output space ($R^1$), thus it is trivial to share all per-task components (all layers in the per-task encoders can be shared only by introducing task embeddings). This enables MTP to efficiently scale up to a large number of tasks, and also acts as a good inductive bias about the homogeneous output spaces. In our experiments in rebuttal (**A2-2**), we indeed observe that the performance improvement by parameter sharing is marginal, although it can greatly improve efficiency.
> >
> > On the other hand, the tasks in the 2D experiments have heterogeneous output spaces (different dimensionality, categorical and continuous) and it is non-trivial to share the per-task components. As the first layer of the encoder (which gets concatenation of $(x, y)$) and the last layer of the decoder (which produces $y$) cannot be shared among tasks with different output dimensionality, we should choose where to separate per-task paths in the encoder and the decoder. The sweet spot that maximizes the model’s performance, as well as parameter efficiency, would indeed depend on how the output spaces are similar (which has nothing to do with how the tasks are correlated) and can be found empirically. Since finding such a sweet spot is not the main focus of our paper, we implemented MTPs for 2D experiments without parameter sharing.
> >
> > Just to be sure, we would like to clarify that the term “parameter sharing” appears in our paper in two different contexts. One is parameter sharing in the encoder across tasks, which is what we are discussing in this response, and the other is between the variational posterior and prior.

---

### Official Review · Reviewer_GYAF · 2021-11-02

**Correctness:** 4
**Technical Novelty And Significance:** 3
**Empirical Novelty And Significance:** 3
**Recommendation:** 8
**Confidence:** 3

**Main Review:**

**Strengths:**
The manuscript is well-written and easy to follow. The motivations for why such a model is needed are clearly laid out and the model is differentiated from the STP and JTP formulations.

The proposed formulation was supplemented with theoretical results to show that the MTP generative model corresponds to a stochastic process. I think that this is an important contribution; however, I did not verify the correctness of the proof.

It was also useful to include the details of the specific architecture that was used in the experiments.

The experiments were well-constructed and were able to showcase some of the properties of the MTP such as its sensitivity to task-relatedness (Figure 3b) and its ability to transfer information from other tasks (Figure 4).


**Weaknesses:**
One of the important considerations for the success of this method is the selection of the context set. More specifically, in many multi-task datasets, it is very rare that two tasks have observed outputs for the same input location. How robust is this method to such a scenario? This point could be nicely added to the toy example as an ablation exercise. More generally, how sensitive is the performance of the MTP to the size and quality of the context set? Are there any methods or heuristics to choosing a good context set?

The paper had no discussion of the possibility of negative transfer with this model. Conditioning all task-specific latents on a global latent variable might cause such an issue if the tasks were not related. Can the authors comment on the MTP’s ability to deal with this issue?

The ELBO in equation (7) has nested expectations in it. I suspect that the Monte Carlo estimate of its gradient might have higher variance than that of STP and JTP due to this. Is this a problem in practice? How does the computational overhead of the MTP compare to the other two methods?

Minor point: Although aesthetically pleasing, I found the different colours representing the different tasks in the figures confusing. I suggest either adding a legend or a comment to explain the difference in the colours.

Reproducibility: Although the experimental details were given in the appendix, there was no indication that the source code is going to be released.


**Summary Of The Paper:**

This work proposes a multi-task learning architecture for neural processes termed the Multi-task process (MTP). The MTP model conditions task-specific latent variables on a global latent variable that is responsible for information sharing between the tasks, and is able to handle both the isotopic and the heterotopic cases. The model was instantiated with an attentive neural process architecture and the generative model of the MTP was shown to correspond to a stochastic process by a Kolmogorov Extension Theorem argument.

**Summary Of The Review:**

In my opinion, this work is original and well-presented. The paper managed to discuss the proposed method in detail and the experimental evaluation was well-constructed. I think this is overall an impactful work and is significant to the probabilistic ML community.

---

> ### Author Response · Authors · 2021-11-23
> **Rebuttal from Authors of Paper952 to Reviewer GYAF (3/3)**
>
> ---
>
> > **Q4.** The paper had no discussion of the possibility of negative transfer with this model. Conditioning all task-specific latents on a global latent variable might cause such an issue if the tasks were not related. Can the authors comment on the MTP’s ability to deal with this issue?
>
> **A4.** We appreciate the reviewer’s comment. First, we would like to kindly remind the reviewer that an ablation study in Section 5.1 (Figure 3(b)) is indeed investigating the robustness of MTPs on various degrees of inter-task correlation. In the ablation study, we observed that MTP did not particularly suffer from the negative transfer when all tasks are independent, as its performance was comparable to MTP-T, a variant of MTP which does not involve any global interaction and consists of task-specific latent variables only. This can be attributed to the design of the hierarchical LVM ($v^t \sim q(v^t|z,C^t)$), where any uninformative knowledge contained in the global latent z can be totally ignored when inferring the task-specific latents $v^{1:T}$ so that only the task-specific knowledge from $C^t$ is used. In contrast, we observe that MTP-G, a variant of MTP which consists only of a global latent suffers from the negative transfer. When the tasks are correlated, on the other hand, MTP significantly outperformed all baselines. We revised the discussion in the main paper to explicitly discuss the negative transfer.
>
> &nbsp;
>
> ---
>
> > **Q5.** The ELBO in equation (7) has nested expectations in it. I suspect that the Monte Carlo estimate of its gradient might have higher variance than that of STP and JTP due to this. Is this a problem in practice? How does the computational overhead of the MTP compare to the other two methods?
>
> **A5.** While our objective involves nested expectations, we approximated it with a single-sample Monte-Carlo estimate as in typical applications of variational Bayes [c]. We could stabilize the optimization with the linear scheduling of “beta” parameters (coefficients multiplied to the KL divergences) and a single ancestral sampling was enough during training. This allows us to not increase computational overhead compared to the non-hierarchical models. At the inference time, we sampled 25 functions from MTP by sampling a global latent and task-specific latents 5 times for each, whose computation is 5x larger than STP/JTP. However, as the sampling can be parallelized as a single forward step, the increase of computational overhead was marginal.
>
> [c] Kingma and Welling. Auto-encoding variational bayes. In ICLR, 2014.
>
> &nbsp;
>
> ---
>
> > **Q6.** Minor point: Although aesthetically pleasing, I found the different colours representing the different tasks in the figures confusing. I suggest either adding a legend or a comment to explain the difference in the colours.
>
> **A6.** We appreciate the reviewer’s suggestion. We added a comment that different colors indicate different tasks in the caption.
>
> &nbsp;
>
> ---
>
> > **Q7.** Reproducibility: Although the experimental details were given in the appendix, there was no indication that the source code is going to be released.
>
> **A7.** We are planning to release the source code and datasets upon the acceptance of this paper. We will add the link to the GitHub repository in the camera-ready version.
>
> &nbsp;
>
> ---

---

> > ### Comment · Reviewer_GYAF · 2021-11-28
> > **Acknowledgement of the authors' response**
> >
> > Dear authors,
> >
> > Thank you for your response. I have read your rebuttal, and I am satisfied with it.
> >
> > Kind regards, reviewer GYAF

---

> ### Author Response · Authors · 2021-11-23
> **Rebuttal from Authors of Paper952 to Reviewer GYAF (2/3)**
>
> ---
>
> **Q2.** More generally, how sensitive is the performance of the MTP to the size and quality of the context set?
>
> **A2.** As all the data-driven models do, MTP performs better when we give more context points, and we give more complete context points. This trend is found in all baselines including MTP, which can be seen in the tables in Section 5.1 and Appendix D, E, and G in the main paper. To compare the sensitivity of the models, we plot the MSEs of the models on synthetic datasets, by varying either context size or incompleteness. As we can see in Figure 6, 7 in the main paper, MTP is the most robust against both context size and quality (incompleteness).
>
> &nbsp;
>
> ---
>
> **Q3.** Are there any methods or heuristics to choosing a good context set?
>
> **A3.** In general, a good context set should contain the representative points of the tasks. For example in synthetic tasks, a good context set should contain much information about the global knowledge (the shape parameters $a, b, c, w$). This could be the mode(s) of sine and gaussian or the center point of the tanh and sigmoid. In general, however, the optimal sampling strategy for a context set is different per function thus likely to be unknown in advance. In this case, uniformly sampling the context given domain would be a good strategy in general. To check this conjecture, we evaluated MTP in synthetic tasks by giving the uniform grid of $(-5, 5)$ as context points (e.g., context input set of size of 3 are chosen to be $\\{-5, 0, 5\\}$), rather than randomly sampling them from $(-5, 5)$. We provide the result below.
>
> | Tasks |  | Sine |  |  | Tanh |  |  | Sigmoid |  |  | Gaussian |  |
> |:---:|:---:|:---:|:---:|:---:|:---:|:---:|:---:|:---:|:---:|:---:|:---:|:---:|
> | m | 5 | 10 | 20 | 5 | 10 | 20 | 5 | 10 | 20 | 5 | 10 | 20 |
> | MTP | 0.2636 | 0.1137 | 0.0485 | 0.0435 | 0.0115 | 0.0040 | 0.0066 | 0.0014 | 0.0006 | 0.0360 | 0.0132 | 0.0069 | 0.0106 |
> | MTP (uniform) | **0.2119** | **0.0952** | **0.0457** | **0.0212** | **0.0070** | **0.0029** | **0.0032** | **0.0010** | **0.0005** | **0.0214** | **0.0098** | **0.0057** | **0.0096** |
>
>
> We observe that using uniformly selected context points, MTP predicts all tasks more accurately, which shows that such context sets are likely more informative. A similar heuristic may be applied to other data, where representative points would help the model to capture the global shape of the function accurately and infer the global latent z better.
>
> &nbsp;
>
> ---

---

> ### Author Response · Authors · 2021-11-23
> **Rebuttal from Authors of Paper952 to Reviewer GYAF (1/3)**
>
> ---
>
> > **Q1.** One of the important considerations for the success of this method is the selection of the context set. More specifically, in many multi-task datasets, it is very rare that two tasks have observed outputs for the same input location. How robust is this method to such a scenario? This point could be nicely added to the toy example as an ablation exercise.
>
> **A1.** We appreciate the reviewer’s suggestion. In the original draft, we have validated the impact of multi-labeled context data in Appendix D by varying the missing rate of the labels in the context data in meta-testing. However, as our framework can generally be applied to the totally incomplete scenario, both for training and testing, we conducted additional experiments on the synthetic dataset. We generated totally incomplete context data by choosing different input points for each task and assigned the corresponding values (according to Eq (66) in the revised version).
>
> For more comprehensive comparisons, we also included meta-learning baselines that use the same meta-train/meta-test data with our method. We chose MAML [a] and Reptile [b] as they are model-agnostic meta-learning methods that can be applied to our multi-task regression setting with incomplete data. The meta-training involved bi-level optimization where the inner loop optimizes the loss for context data and the outer loop optimizes the loss for target data. We employed a similar architecture to MTP for the baselines that consist of an encoder network shared by all tasks and task-specific decoder networks. For fair comparisons, we controlled the total number of parameters of the models similar to NP baselines (STP, JTP, MTP).
>
> To study the robustness of our method in the totally incomplete scenario, we compare two groups of models. The first group is trained with partially incomplete data (with a missing rate 0.5) and then tested with totally incomplete data. The second group is both trained and tested with totally incomplete data. Since JTP can be trained only with complete data, we compare it in the first group. The results are included below and Appendix D.2 in the revised version.
>
> [Table for the first group.]
>
> | Tasks |  | Sine |  |  | Tanh |  |  | Sigmoid |  |  | Gaussian |  |
> |:---:|:---:|:---:|:---:|:---:|:---:|:---:|:---:|:---:|:---:|:---:|:---:|:---:|
> | m | 5 | 10 | 20 | 5 | 10 | 20 | 5 | 10 | 20 | 5 | 10 | 20 |
> | MAML | 0.6123 | 0.3715 | 0.1930 | 0.4117 | 0.1230 | 0.0308 | 0.2220 | 0.0456 | 0.0112 | 0.2281 | 0.0911 | 0.0365 |
> | Reptile | 0.8089 | 0.5702 | 0.3482 | 0.4495 | 0.1557 | 0.0487 | 0.3012 | 0.0630 | 0.0210 | 0.3734 | 0.1435 | 0.0725 |
> | MOSM | 1.0718 | 0.6073 | 0.3728 | 1.5199 | 0.9601 | 0.5937 | 0.4798 | 0.2471 | 0.3070 | 0.3245 | 0.2190 | 0.2023 |
> | CSM | 1.0887 | 0.6557 | 0.3462 | 1.4156 | 1.1087 | 0.8813 | 0.5256 | 0.6013 | 0.4581 | 0.2086 | 0.1815 | 0.1456 |
> | STP | 0.6491 | 0.4228 | 0.2116 | 0.1753 | 0.0843 | 0.0344 | 0.0271 | 0.0136 | 0.0050 | 0.0962 | 0.0608 | 0.0366 |
> | STP+JTP | 0.5162 | 0.3472 | 0.1976 | 0.1411 | 0.0772 | 0.0327 | 0.0236 | 0.0122 | 0.0053 | 0.0819 | 0.0470 | 0.0263 |
> | MTP | **0.3777** | **0.2045** | **0.1007** | **0.0890** | **0.0293** | **0.0089** | **0.0152** | **0.0049** | **0.0014** | **0.0535** | **0.0246** | **0.0113** |
>
>
> [Table for the second group.]
>
> | Tasks |  | Sine |  |  | Tanh |  |  | Sigmoid |  |  | Gaussian |  |
> |:---:|:---:|:---:|:---:|:---:|:---:|:---:|:---:|:---:|:---:|:---:|:---:|:---:|
> | m | 5 | 10 | 20 | 5 | 10 | 20 | 5 | 10 | 20 | 5 | 10 | 20 |
> | MAML | 0.5906 | 0.3967 | 0.2347 | 0.3366 | 0.1018 | 0.0317 | 0.1999 | 0.0396 | 0.0148 | 0.1995 | 0.0862 | 0.0451 |
> | Reptile | 0.7461 | 0.5945 | 0.3798 | 0.3690 | 0.1202 | 0.0412 | 0.2431 | 0.0474 | 0.0177 | 0.3176 | 0.1239 | 0.0694 |
> | MOSM | 0.9080 | 0.6939 | 0.6504 | 0.7423 | 0.6695 | 0.6132 | 0.6438 | 0.6054 | 0.6009 | 0.7974 | 0.7498 | 0.7201 |
> | CSM | 0.9501 | 0.7644 | 0.6486 | 0.7902 | 0.7175 | 0.7691 | 0.6620 | 0.6454 | 0.7356 | 0.9119 | 0.8470 | 0.9381 |
> | STP | 0.7683 | 0.5112 | 0.3111 | 0.2649 | 0.1642 | 0.0679 | 0.0482 | 0.0270 | 0.0088 | 0.0745 | 0.0589 | 0.0449 |
> | STP+JTP | 0.7591 | 0.5058 | 0.2848 | 0.2348 | 0.1486 | 0.0593 | 0.0412 | 0.0235 | 0.0089 | 0.1136 | 0.0669 | 0.0367 |
> | MTP | **0.4860** | **0.3391** | **0.1808** | **0.1596** | **0.0563** | **0.0188** | **0.0302** | **0.0092** | **0.0028** | **0.0668** | **0.0367** | **0.0191** |

---

> > ### Author Response · Authors · 2021-11-23
> > **Rebuttal from Authors of Paper952 to Reviewer GYAF (1/3) (continued)**
> >
> > We observe that when the data is highly incomplete ($\gamma = 0.5, 0.75$), these two meta-learning baselines perform better than STP and/or JTP in synthetic tasks. This demonstrates that using inter-task correlation through multi-task data is beneficial when the effective number of observable examples is small (either small m or high gamma). Also, unlike JTP, they can inherently handle incomplete data and thus do not have to suffer from noisy imputations. However, they still perform worse than MTP in general and are comparable to STP and JTP when the data is complete ($\gamma = 0$). Also, they generally perform poorly in weather tasks. We noticed that the models fail to capture the global shape of each task, overfitting the context points, which is illustrated in qualitative results in Appendix D.1 and E of the main paper. We conjecture that the overfitting comes from the lack of global inference on the function space, which is especially important when there exist a large amount of observation noises as in weather data.
> >
> > [a] Finn et al. Model-agnostic meta-learning for fast adaptation of deep networks. In ICML, 2017.
> > [b] Nichol et al. On first-order meta-learning algorithms. arXiv preprint arXiv:1803.02999 (2018).

---

### Official Review · Reviewer_F6YH · 2021-11-03

**Correctness:** 3
**Technical Novelty And Significance:** 3
**Empirical Novelty And Significance:** 2
**Recommendation:** 6
**Confidence:** 4

**Main Review:**

The model is an extension of Neural Processes (NP) to the multi-task learning case. This work is valuable to the community, providing a new probabilistic multi-task learning model that is able to handle incomplete (missing) data.
The paper is generally well-written.

**Meta-learning, multi-task, and GP baselines**

The work extends a meta-learning method (NP) to multi-task. This means that is it targeted at meta-learning and not applicable to the more classic multi-task setting (i.e. one set of test tasks). Hence, comparisons to GP baselines are not really fair (multi-task vs meta-learning). MTP uses much more data (training) while GP methods are given only a test set.

How do you train GP-based models? Is it only using test data? Why were those baseline chosen?

Train tasks could be used to pretrain kernel parameters in the models you chose. While in MOGPs where correlations are modelled via latent variables, for example [1], those variables could be pretrained and later used as a prior correlation on test tasks.

Given that you are using GP baselines there is some missed related work on meta-learning in GP:  [2], [3].

As it is done now, the GP baselines do not seem like a fair comparison.

The naive NP-based models are performing worse than MTP, but this is quite natural since they fundamentally lack the ability to model cross-correlation (in incomplete data).

**Sloppy notation/language around VI, errors in the supplement**

I found the notation and language about the Variational Inference part to be quite sloppy and confusing.

In 2.1 the conditional prior $p(z|C_t)$ is called both "posterior" and "prior". What do you mean by intractable prior? It makes sense that the posterior $p(z|C, D)$ is intractable and approximated with $q(z|D)$, but not so much for $p(z|C_t)$ (according to the graphical model).  Maybe just parameterize  $p(z|C_t)$ with the same NN model instead of using $q(z|C_t)$. It will not make a practical difference, but the explanation would be more clear.

The derivations in A.3 of the supplement are not accurate. While the end result is correct, the derivation is very misleading. Eq(31) is not = eq(32).  The denominator in (32) should be $q(z|D)$ and instead of $=$ it is $\geq$. Same for eq(36) and eq(37).

**Training on multiple sets of tasks**

If I understand correctly, the latent variable $z$ is shared within a set of tasks, but independent between sets of tasks. Is this correct? This is not clear from the text.

**Minor comments**

Consider renaming to Multi-task Neural Processes (MTNP), it would be much more clear.

The cubic complexity of GPs is correct in the naive case. However, given the wide use of much more data-efficient sparse GP methods, I think it is misleading to simply state "cubic complexity".

Sec.1 "Then we define MTPs over the combined function space by extending the conditional Latent Variable Model (LVM) of NPs". Rephrase this sentence to be more general. It is unclear what the "conditional Latent Variable Model (LVM)" is without checking Garnelo et al., 2018b.

Better is in eq(10) and (11) you write explicitly that it is q(z|C) and q(v_t|z, C_t).

Sec. 4 "MTPs learn inter-task correlation from multiple context and target pairs through meta-training, which enables accurate inference with a few observations. In contrast, MOGPs learn the correlation only from the context data given at inference time, thus require a lot of observations to produce accurate predictions." This is misleading since you could use "training tasks" in MOGPs to either pretrain the kernel parameters or even include them as different tasks.

Sec. 5 no error bars in the tables, only means. Would be good if you can squeeze std in the tables. At least explicitly refer to the supplement for this information.

Figures 3 and 4 are of low resolution, which is particularly problematic when it comes to reding the legend.

[1] Dai, Zhenwen, Mauricio A. Álvarez, and Neil D. Lawrence. "Efficient modeling of latent information in supervised learning using gaussian processes." *arXiv preprint arXiv:1705.09862*(2017).

[2] Fortuin, Vincent, and Gunnar Rätsch. "Deep mean functions for meta-learning in gaussian processes." *arXiv preprint arXiv:1901.08098* (2019).

[3] Titsias, Michalis K., et al. "Information Theoretic Meta Learning with Gaussian Processes." *arXiv preprint arXiv:2009.03228*(2020).


**Summary Of The Paper:**

The paper presents a novel model for multi-task learning with missing data based on Neural Processes (NP). Inter-task correlations are modelled via a shared latent variable. The model has been tested on 1 synthetic and 2 real-world datasets and is experimentally shown to perform better against 4 baselines including 2 naive extensions of NP to the multi-task setting and 2 multi-output Gaussian process models (with spectral kernels).

**Summary Of The Review:**

Overall the method is novel and useful. However, the chosen baselines are quite weak. The explanation of the inference part needs correction and clarifications.

UPDATE: I have increased my recommendation from 5 to 6, given the clarifications and extra experiments provided in the rebuttal.

---

> ### Author Response · Authors · 2021-11-23
> **Rebuttal from Authors of Paper952 to Reviewer F6YH (5/5)**
>
> ---
>
> > **Q11.** Sec. 5 no error bars in the tables, only means. Would be good if you can squeeze std in the tables. At least explicitly refer to the supplement for this information.
>
> **A11.** We appreciate the reviewer’s suggestion. Due to the page limit and presentation issue, we added the standard deviation for each quantitative metric in Appendix D, E, and G. We included the explicit reference to Appendix in Section 5 of the main paper.
>
> &nbsp;
>
> ---
>
> > **Q12.** Figures 3 and 4 are of low resolution, which is particularly problematic when it comes to reding the legend.
>
> **A12.** We appreciate the reviewer’s comment. We increased the size of legends in Figures 3 and 4.
>
> &nbsp;
>
> ---

---

> > ### Comment · Reviewer_F6YH · 2021-11-28
> > **Acknowledging the authors' response**
> >
> > Thank you for the thorough rebuttal. You have addressed most of my concerns, particularly regarding the experiments.

---

> ### Author Response · Authors · 2021-11-23
> **Rebuttal from Authors of Paper952 to Reviewer F6YH (4/5)**
>
> > **Q5.** If I understand correctly, the latent variable is shared within a set of tasks, but independent between sets of tasks. Is this correct? This is not clear from the text.
>
> **A5.** The reviewer understood it correctly. The global latent variable $z$ is inferred from the context data $C$ of a set of tasks and shared within the set. For different sets of tasks, the variable is inferred independently, since the conditioning context $C$ is different for different sets of tasks. Please note that this is reflected in the conditional LVM of JTP (Eq (4)) or MTP (Eq (5)).
>
> &nbsp;
>
> ---
>
> > **Q6.** Consider renaming to Multi-task Neural Processes (MTNP), it would be much more clear.
>
> **A6.** We appreciate the reviewer’s suggestion and will change the title of the paper accordingly.
>
> &nbsp;
>
> ---
>
> > **Q7.** The cubic complexity of GPs is correct in the naive case. However, given the wide use of much more data-efficient sparse GP methods, I think it is misleading to simply state "cubic complexity".
>
> **A7.** We appreciate the comment. We revised the description regarding the cubic complexity of GPs to explicitly mention “GPs without any approximations”.
>
> &nbsp;
>
> ---
>
> > **Q8.** Sec.1 "Then we define MTPs over the combined function space by extending the conditional Latent Variable Model (LVM) of NPs". Rephrase this sentence to be more general. It is unclear what the "conditional Latent Variable Model (LVM)" is without checking Garnelo et al., 2018b.
>
> **A8.** We appreciate the reviewer’s comment. We rephrased the sentence to “Then we define a Latent Variable Model (LVM) of MTP that theoretically induces a stochastic process over the combined function space.”.
>
> &nbsp;
>
> ---
>
> > **Q9.** Better is in eq(10) and (11) you write explicitly that it is $q(z|C)$ and $q(v^t|z, C^t)$.
>
> **A9.** We appreciate the reviewer’s suggestion. We revised the notation accordingly in eq(10) and (11).
>
> &nbsp;
>
> ---
>
> > **Q10.** Sec. 4 "MTPs learn inter-task correlation from multiple context and target pairs through meta-training, which enables accurate inference with a few observations. In contrast, MOGPs learn the correlation only from the context data given at inference time, thus require a lot of observations to produce accurate predictions." This is misleading since you could use "training tasks" in MOGPs to either pretrain the kernel parameters or even include them as different tasks.
>
> **A10.** As we discussed in **A1**, we did not consider either pretraining the kernel parameters or including training tasks to testing tasks since these are not the conventional usage of MOGPs we included as baselines, with our best knowledge. We acknowledged that such methods can be used to MOGPs to exploit training tasks. Therefore, we toned the arguments down and added the comparisons to MOGPs with kernel pertaining.
>
> &nbsp;
>
> ---

---

> ### Author Response · Authors · 2021-11-23
> **Rebuttal from Authors of Paper952 to Reviewer F6YH (3/5)**
>
> ---
>
> > **Q4.** The derivations in A.3 of the supplement are not accurate. While the end result is correct, the derivation is very misleading. Eq(31) is not = eq(32). The denominator in (32) should be q(z|D) and instead of = it is $\ge$. Same for eq(36) and eq(37).
>
> **A4.** We appreciate the comment. We found that we omitted the assumption on context and target data: $C \subset D$, and modified Section 2.2 in the paper to include the assumption. This is not necessary when we describe NPs, but commonly adopted in practice [a,c,g]. With this assumption, Eq (31) and (33) (Eq (31) and Eq (32) in the previous draft) are the same because $p_\theta(z|X_D^{1:T}, Y_D^{1:T}, C) = p_\theta(z|D)$, then it follows that
>
> $$
> \log p_\theta(Y_D^{1:T}|X_D^{1:T}, C)
> = E_{q_{\phi}(z|D)}[\log p_\theta(Y_D^{1:T}|X_D^{1:T}, C)]
> = E_{q_{\phi}(z|D)}[\log \frac{p_\theta(Y_D^{1:T}, z|X_D^{1:T}, C)}{p_\theta(z|X_D^{1:T}, Y_D^{1:T}, C)}]
> = E_{q_{\phi}(z|D)}[\log \frac{p_\theta(Y_D^{1:T}|X_D^{1:T}, C, z) p_\theta(z|X_D^{1:T}, C)}{p_\theta(z|D)}]
> $$
>
> Similarly, Eq (37) and (39) (Eq (36) and (37) in the previous draft) are the same because $p_\theta(v^{1:T}|X_D^{1:T},Y_D^{1:T},C,z) = p_\theta(v^{1:T}|D,z)$, then it follows that
>
> $$
> \log p_\theta(Y_D^{1:T}|X_D^{1:T}, C)
> = E_{q_\phi(z|D)}[\log p_\theta(Y_D^{1:T}|X_D^{1:T}, C)]
> = E_{q_{\phi}(z|D)}[\log \frac{p_\theta(Y_D^{1:T}, z|X_D^{1:T}, C)}{p_\theta(z|X_D^{1:T}, Y_D^{1:T}, C)}]
> = E_{q_{\phi}(z|D)}[\log \frac{p_\theta(Y_D^{1:T}|X_D^{1:T}, C, z) p_\theta(z|X_D^{1:T}, C)}{p_\theta(z|D)}]
> $$
>
> Note that even without the assumption, we may derive a similar objective where $D$ (or $D^t$) in the KL divergence terms is replaced with $C \cup D$ (or $C^t \cup D^t$).
>
> [a] Kim et al. Attentive neural processes. In ICLR, 2019.
> [c] Wang and Hoof. Doubly stochastic variational inference for neural processes with
> hierarchical latent variables. In ICML, 2020.
> [g] Garnelo et al. Neural processes. In ICML Workshop, 2018.
>
> &nbsp;
>
> ---

---

> > ### Comment · Reviewer_F6YH · 2021-11-28
> > **There are still errors in the supplement**
> >
> > While you clarification is useful, it still does not address my concern. Yes, eq(1) $\>=$ eq(36), but your line of derivations is very weird. How do you go from eq(31) to eq(32)? If you follow Jensen's inequality, then (omitting subscripts and superscripts): $\log p(Y|X,C)= \log \int {p(Y,z|X,C)} dz \geq E_{q(z|D)}[\log \frac{p(Y|X,C,z)p(z|C)}{q(z|D)}]=E_{q(z|D)}[\log p(Y|X,C,z)] - KL\big(q(z|D)||p(z|C)\big)$.
> > If you follow a different line of reasoning, please explain it.

---

> > > ### Author Response · Authors · 2021-11-29
> > > **We appreciate the comment.**
> > >
> > > We appreciate the comment. We would like to clarify that the reviewer's derivation is also correct and indeed follows a more standard way to derive the variational lower-bound.
> > >
> > > We followed an alternative derivation, which uses $D_{KL}\left(q(z|D)||p(z|D)\right) \ge 0$. This kind of derivation can be found in [h] (Section 2.2), where the only difference from ours is the presence of conditioning (on $X, C$).
> > > In this response, we explain the process used to derive Eq.(32) in detail.
> > >
> > > Consider the following equations, where we omitted superscripts and subscripts from Eq.(31)=Eq.(32).
> > > $$
> > > \log p(Y|X,C) = E_{q(z|D)}\left[\log p(Y|X, C)\right] = E_{q(z|D)} \left[ \log \frac{p(z, Y|X, C)}{p(z|Y, X, C)} \right].
> > > $$
> > > The first equation is trivial since $\log p(Y|X, C)$ does not involve $z$. Then it suffices to show
> > > $$
> > > p(Y|X, C) = \frac{p(z, Y|X, C)}{p(z|Y, X, C)}.
> > > $$
> > > For notational simplicity, let $W := (X, C)$. Then by the definition of conditional probability, we get
> > > $$
> > > p(z|Y, W) = \frac{p(z, Y, W)}{p(Y, W)} = \frac{p(z, Y|W)p(W)}{p(Y|W)p(W)} = \frac{p(z, Y| W)}{p(Y|W)}.
> > > $$
> > > By transposing $p(z|Y, W)$ and $p(Y|W)$ to each other sides, we get
> > > $$
> > > p(Y|W) = \frac{p(z, Y|W)}{p(z|Y, W)}.
> > > $$
> > > Finally, by re-subtituting $W$ with $(X, C)$, we get
> > > $$
> > > p(Y|X, C) = \frac{p(z, Y|X, C)}{p(z|Y, X, C)}.
> > > $$
> > > We used similar reasoning to derive Eq.(38). Note that an alternative form of Jensen's inequality is used in Eq.(36) (and Eq.(43)), i.e., $D_{KL}\left(q(z|D)||p(z|D)\right) \ge 0$.
> > > This derivation reaches the same lower-bound as the reviewer's.
> > >
> > > We deeply appreciate the thoughtful comment and will make this clear in the revision.
> > >
> > > [h] Kingma & Welling. An Introduction to Variational Autoencoders. Foundations and Trends® in Machine Learning. 2019.

---

> > > > ### Comment · Reviewer_F6YH · 2021-11-29
> > > > **Thank you for the clarification**
> > > >
> > > > Thank you for the clarification. You are absolutely right, my mistake here. The is no need to add anything to the equations, but it will be good if you mention that you follow [h].

---

> ### Author Response · Authors · 2021-11-23
> **Rebuttal from Authors of Paper952 to Reviewer F6YH (2/5)**
>
> ---
>
> **Q2.** Given that you are using GP baselines there is some missed related work on meta-learning in GP: [2], [3].
>
> **A2.** We appreciate the comment. We added the discussions in the related work in Section 4.
>
> &nbsp;
>
> ---
>
> **Q3.** In 2.1 the conditional prior is called both "posterior" and "prior". What do you mean by intractable prior? It makes sense that the posterior $p(z|C,D)$ is intractable and approximated with $q(z|D)$, but not so much for $p(z|C^t)$ (according to the graphical model). Maybe just parameterize $p(z|C^t)$ with the same NN model instead of using $q(z|C^t)$. It will not make a practical difference, but the explanation would be more clear.
>
> **A3.** We appreciate the reviewer’s comment. As suggested, we found that some explanations of the graphical model and its training objective could be confusing. We revised Section 2.1 and 2.2 to clarify confusing terminologies and explanations, where the changes are summarized as follows.
>
> First, we unified the terminology for $p(z|C^t)$ to conditional prior.
>
> Second, regarding the explanation of NP’s training objective, we agree with the reviewer and revise it following the reviewer’s perspective as follows. The conditional LVM $p(Y_D^t|X_D,C^t)$ consists of two components, conditional prior $p(z|C^t)$ and generative model $p(Y_D^t|X_D,z)$, and we model them with an encoder network $p(z|C^t) = p_\theta(z|C^t)$ and a decoder network $p(Y_D^t|X_D,z) = p_\theta(Y_D^t|X_D,z)$, respectively. However, since the conditional likelihood $p(Y_D^t|X_D,C^t)$ involves intractable integration, we should employ variational posterior inference. Then we share the parameters of the encoder network $p_\theta$ for conditional prior and the network $q_\phi$ for variational posterior, which results in the objective (Eq.(2)). In other words, the same objective does not involve “approximation of intractable prior” but injects an inductive bias through parameter tying between prior and variational posterior networks.
>
> Our previous explanation of the objective was based on an alternative interpretation of NPs, where context and target variables are identified only by conditioning [f]. Here unconditional latent variable model $p(Y_D^t|X_D) = \int p(Y_D^t|X_D,z) p(z)dz$ is defined first and the conditional likelihood $p(Y_D^t|X_D,C^t)$ for $C^t \subset D^t$ is derived via Bayes rule. In this perspective, the conditional prior
> $$
> p(z|C^t) = \frac{p(z) \prod_{i \in \mathcal{I}(C^t)} p(y_i^t|x_i, z)}{\int p(z) \prod_{i \in \mathcal{I}(C^t)} p(y_i^t|x_i, z) dz}
> $$
> is intractable, and we need to approximate it using either a separate network or the variational posterior network, where we adopted the latter.
>
> Nevertheless, we found that the parameter-tying perspective is more clear and aligns to our conditional LVM definition and the proof of proposition 1. Therefore, we revised the main paper and proofs in the Appendix with this perspective.
> Finally, we revised the explanation of the training objective of MTPs in the same vein. The encoder network $q_\phi$ amortizes both the variational posterior $q_\phi(z, v^{:T}|D)$ and the conditional prior $p_\theta(z, v^{1:T}|C)$.
>
> [f] ​​Gordon, Jonathan. Advances in Probabilistic Meta-Learning and the Neural Process Family. Diss. University of Cambridge, 2021.
>
> &nbsp;
>
> ---

---

> ### Author Response · Authors · 2021-11-23
> **Rebuttal from Authors of Paper952 to Reviewer F6YH (1/5)**
>
> ---
>
> > **Q1.** How do you train GP-based models? Is it only using test data? Why were those baseline chosen? Train tasks could be used to pretrain kernel parameters in the models you chose. While in MOGPs where correlations are modelled via latent variables, for example [1], those variables could be pretrained and later used as a prior correlation on test tasks.
>
> **A1.** We appreciate the reviewer’s thoughtful comments and suggestions. In this response, we would like to clarify why we considered MOGPs as our baselines and discuss how we reflected the reviewer’s concern in our rebuttal.
> In this paper, we investigate extending NPs to consider multiple correlated tasks, where context data can be given incompletely. To the best of our knowledge, the idea of modeling multiple tasks as a set of stochastic processes has been only investigated in the GP literature (MOGPs), which also can handle incomplete data. Considering its relevance to our work and that GPs have been served as standard baselines for stochastic processes in NP literature [a,b,c], it seemed natural to consider MOGPs as our baseline. To apply MOGPs, we followed the original settings, where the kernel parameters are trained on training data of each set of tasks (which corresponds to context data of each set of tasks in testing sets, in our notation).
>
> However, we acknowledge that direct comparisons to MOGPs can be unfair due to different training settings. As our best attempt to make the comparison fair, we followed the reviewer’s suggestions and modified MOGPs to pretrain the kernel parameters. Specifically, we pretrain MOGPs with respect to the kernel parameters using the same meta-training dataset with MTP, and transfer the learned kernel parameters as prior in meta-testing. This allows both MOGPs and MTPs to be trained and evaluated under the same setting. To prevent overfitting, we early-stopped the pretraining based on NLL. We include the results in Section 5 in the main paper, Appendix D and E, and also summarize them below.
>
> [Results on synthetic tasks, $\gamma = 0.5$.]
>
> | Tasks |  | Sine |  |  | Tanh |  |  | Sigmoid |  |  | Gaussian |  |
> |:---:|:---:|:---:|:---:|:---:|:---:|:---:|:---:|:---:|:---:|:---:|:---:|:---:|
> | m | 5 | 10 | 20 | 5 | 10 | 20 | 5 | 10 | 20 | 5 | 10 | 20 |
> | MOSM (w/ pretraining) | 0.7852 | 0.4410 | **0.0298**| 0.4912 | 0.1444 | 0.1618 | 0.0720 | 0.0127 | 0.0013 | 0.3329 | 0.0857 | 0.0190 |
> | CSM (w/ pretraining) | 0.8529 | 0.3587 | 0.1537 | 0.6884 | 0.3669 | 0.0726 | 0.2437 | 0.0730 | 0.0137 | 0.1525  | 0.0961 | 0.0407 |
> | MOSM (w/o pretraining) | 1.8653 | 0.7057 | 0.4452 | 1.9980 | 0.6220 | 0.3625 | 1.6795 | 0.3758 | 0.1769 | 1.5285 | 0.4438 | 0.2045 |
> | CSM (w/o pretraining) | 0.5516 | 0.6724 | 0.6975 | 0.3580 | 0.4445 | 0.5134 | 0.2584 | 0.3003 | 0.3821 | 0.2992 | 0.3345 | 0.4240 |
> | STP | 0.5212 | 0.2609 | 0.0993 | 0.1307 | 0.0468 | 0.0159 | 0.0203 | 0.0067 | 0.0025 | 0.0799 | 0.0409 | 0.0222 |
> | STP+JTP | 0.3848 | 0.2340 | 0.1114 | 0.1015 | 0.0418 | 0.0168 | 0.0163 | 0.0065 | 0.0032 | 0.0613 | 0.0318 | 0.0161 |
> | MTP | **0.2636** | **0.1137** | 0.0485 | **0.0435** | **0.0115** | **0.0040** | **0.0066** | **0.0014** | **0.0006** | **0.0360** | **0.0132** | **0.0069** |

---

> > ### Author Response · Authors · 2021-11-23
> > **Rebuttal from Authors of Paper952 to Reviewer F6YH (1/5) (continued)**
> >
> > For more fair comparisons, we also included meta-learning baselines that use the same meta-train/meta-test data with our method. We chose MAML [d] and Reptile [e] as they are model-agnostic meta-learning methods that can be applied to our multi-task regression setting with incomplete data. The meta-training involved bi-level optimization where the inner loop optimizes the loss for context data and the outer loop optimizes the loss for target data. We employed a similar architecture to MTP for the baselines that consist of an encoder network shared by all tasks and task-specific decoder networks. For fair comparisons, we controlled the total number of parameters of the models similar to NP baselines (STP, JTP, MTP). We include the results in Section 5 in the main paper, Appendix D.1 and E, and also summarize them below.
> >
> > [Results on synthetic tasks, $\gamma = 0.5$.]
> >
> > | Tasks |  | Sine |  |  | Tanh |  |  | Sigmoid |  |  | Gaussian |  |
> > |:---:|:---:|:---:|:---:|:---:|:---:|:---:|:---:|:---:|:---:|:---:|:---:|:---:|
> > | m | 5 | 10 | 20 | 5 | 10 | 20 | 5 | 10 | 20 | 5 | 10 | 20 |
> > | MAML | 0.2962 | 0.1582 | 0.0701 | 0.0991 | 0.0342 | 0.0131 | 0.0321 | 0.0119 | 0.0069 | 0.0696 | 0.0353 | 0.0174 |
> > | Reptile | 0.5164 | 0.2886 | 0.1414 | 0.1656 | 0.0557 | 0.0291 | 0.0619 | 0.0220 | 0.0181 | 0.1371 | 0.0679 | 0.0374 |
> > | STP | 0.5212 | 0.2609 | 0.0993 | 0.1307 | 0.0468 | 0.0159 | 0.0203 | 0.0067 | 0.0025 | 0.0799 | 0.0409 | 0.0222 |
> > | STP+JTP | 0.3848 | 0.2340 | 0.1114 | 0.1015 | 0.0418 | 0.0168 | 0.0163 | 0.0065 | 0.0032 | 0.0613 | 0.0318 | 0.0161 |
> > | MTP | **0.2636** | **0.1137** | **0.0485** | **0.0435** | **0.0115** | **0.0040** | **0.0066** | **0.0014** | **0.0006** | **0.0360** | **0.0132** | **0.0069** |
> >
> >
> > [Results on weather tasks, $\gamma = 0.5$ and $m = 10$.]
> >
> > | Tasks | TempMin |  | TempMax |  | Humidity |  | Precip |  | Cloud |  | Dew |  |
> > |---:|---:|---:|---:|---:|---:|---:|---:|---:|---:|---:|---:|---:|
> > | metric | MSE | NLL | MSE | NLL | MSE | NLL | MSE | NLL | MSE | NLL | MSE | NLL |
> > | MAML | 0.0067 | - | 0.0094 | - | 0.0705 | - | 0.3041 | - | 0.2987 | - | 0.0106 | - | 0.0106 | - |
> > | Reptile | 0.0060 | - | 0.0078 | - | 0.0691 | - | 0.3160 | - | 0.3047 | - | 0.0096 | - | 0.0096 | - |
> > | STP | 0.0046 | -1.1514 | 0.0069 | -1.0390 | 0.0632 | 0.1273 | 0.2607 | 1.1242 | 0.2631 | 0.8563 | 0.0086 | -0.9815 | 0.0086 | -0.9815 |
> > | STP+JTP | 0.0045 | -1.1703 | 0.0068 | -1.0681 | 0.0607 | 0.0169 | 0.2348 | 0.6792 | 0.2376 | 0.6812 | 0.0084 | -0.9946 | 0.0084 | -0.9946 |
> > | MTP | **0.0037** | **-1.1832** | **0.0054** | **-1.1049** | **0.0546** | **-0.1006** | **0.2276** | **0.6557** | **0.2215** | **0.6660** | **0.0073** | **-1.0331** | **0.0073** | **-1.0331** |
> >
> >
> > We observe that when the data is highly incomplete ($\gamma = 0.5, 0.75$), these two meta-learning baselines perform better than STP and/or JTP in synthetic tasks. This demonstrates that using inter-task correlation through multi-task data is beneficial when the effective number of observable examples is small (either small m or high gamma). Also, unlike JTP, they can inherently handle incomplete data and thus do not have to suffer from noisy imputations. However, they still perform worse than MTP in general and are comparable to STP and JTP when the data is complete ($\gamma = 0$). Also, they generally perform poorly in weather tasks. We noticed that the models fail to capture the global shape of each task, overfitting the context points, which is illustrated in qualitative results in Appendix D.1 and E of the main paper. We conjecture that the overfitting comes from the lack of global inference on the function space, which is especially important when there exist a large amount of observation noises as in weather data.
> >
> > We also notice that the gradient-based meta-learning approaches are much inefficient in the meta-testing time compared to NP baselines, because they require several gradient steps for adaptation. For example, in our synthetic experiment, they required 1000 backward steps (with minibatch-size 4) per set of tasks to converge, while NP baselines require only a single forward step per set of tasks. NPs and MTPs are also more reliable as there are no optimization hyperparameters (such as fine-tuning learning rates or steps) in their adaptation.
> >
> > We appreciate the reviewer’s suggestion and hope our response and the new experiments address the concern. Please let us know if the reviewer has any remaining concerns or suggestions for the experiments.
> >
> > [a] Kim et al. Attentive neural processes. In ICLR, 2019.
> > [b] Louizos et al. The functional neural process. In NeurIPS, 2019.
> > [c] Wang and Hoof. Doubly stochastic variational inference for neural processes with
> > hierarchical latent variables. In ICML, 2020.
> > [d] Finn et al. Model-agnostic meta-learning for fast adaptation of deep networks. In ICML, 2017.
> > [e] Nichol et al. On first-order meta-learning algorithms. arXiv preprint arXiv:1803.02999 (2018).
> >
> > &nbsp;
> >
> > ---

---

> > ### Author Response · Authors · 2021-11-23
> > **Rebuttal from Authors of Paper952 to Reviewer F6YH (1/5) (continued)**
> >
> > [Results on weather tasks, $\gamma = 0.5$ and $m = 10$.]
> >
> > | Tasks | TempMin |  | TempMax |  | Humidity |  | Precip |  | Cloud |  | Dew |  |
> > |---:|---:|---:|---:|---:|---:|---:|---:|---:|---:|---:|---:|---:|
> > | metric | MSE | NLL | MSE | NLL | MSE | NLL | MSE | NLL | MSE | NLL | MSE | NLL |
> > | MOSM (w/ pretraining) | 0.0638 | 297.5341 | 0.0107 | 4.9678 | 0.1944 | 371.3901 | 0.3906 | 35.0833 | 0.3631 | 12.3570 | 0.0168 | 13.6767 |
> > | CSM (w/ pretraining) | 0.0074 | 0.0404 | 0.0132 | 4.1934 | 0.0781 | 2.1373 | 0.2896 | 6.6982 | 0.2944 | 3.0075 | 0.0137 | 0.9086 |
> > | MOSM (w/o pretraining) | 0.0091 | -0.0194 | 0.0124 | -0.0259 | 0.0827 | 1.3831 | 0.3021 | 4.1009 | 0.3170 | 2.0663 | 0.0128 | -0.0255 |
> > | CSM (w/o pretraining) | 0.0069 | -0.8839 | 0.0123 | -0.8522 | 0.0906 | 0.6640 | 0.2895 | 3.1897 | 0.2983 | 1.2655 | 0.0118| -0.7243 |
> > | STP | 0.0046 | -1.1514 | 0.0069 | -1.0390 | 0.0632 | 0.1273 | 0.2607 | 1.1242 | 0.2631 | 0.8563 | 0.0086 | -0.9815 |
> > | STP+JTP | 0.0045 | -1.1703 | 0.0068 | -1.0681 | 0.0607 | 0.0169 | 0.2348 | 0.6792 | 0.2376 | 0.6812 | 0.0084 | -0.9946 |
> > | MTP | **0.0037** | **-1.1832** | **0.0054** | **-1.1049** | **0.0546** | **-0.1006** | **0.2276** | **0.6557** | **0.2215** | **0.6660** | **0.0073** | **-1.0331** |
> >
> > We observe that pretraining is effective in synthetic datasets, where we sample the shape parameters $a, b, c, w$ from the same distributions for training and testing sets of tasks. Thus we revise the tables for synthetic tasks with the pretrained version. However, even with pretraining, the MOGPs perform worse than the NP baselines and MTP when the context size is small ($m = 5, 10$). We observed that the MOGPs often fail to capture the general shapes of the curves (e.g., unimodal nature of Gaussian). Also, the extrapolation performance of them is poor while they interpolate context points well. These trends can be seen in Figure 8-11 that we included in Appendix D.1.
> >
> > On the other hand, pretraining deteriorates both the MSE and NLL in the weather dataset, which is likely due to the overfitting in the pretraining datasets. This is not surprising as the training and testing sets in the weather dataset contain weather attributes from different cities, which may have different distributions. We conjecture that the lack of meta-learning ability makes the pretrained GPs prone to overfitting in such real-world meta-learning scenarios.

---

### Decision · Program_Chairs · 2022-01-20

**Decision:**

Accept (Poster)

**Comment:**

Three knowledgeable referees recommend Accept. Reviewer eyrZ's concerns have been addressed by the authors in the rebuttal, in my opinion.  Therefore I recommend Accept. I ask the authors to 1) rename the title of their paper and their model to the more specific name Multi-task Neural Processes (MTNP). I agree with both reviewers F6YH and ACBa that the name "Multi-task Processes" does not make justice to the many other models out there that also provide ways to model several stochastic processes simultaneously. Make sure you propagate the name of the new model through the paper. 2) include a discussion in the main paper about the variability of the new results provided in the rebuttal. Only mean NLL and MSE are provided which can be misleading without standard deviations and potential tests for statistical significance.